# Liver X receptor unlinks intestinal regeneration and tumorigenesis

Srustidhar Das[1,2,21 ✉], S. Martina Parigi[1,2,19,21], Xinxin Luo[1,2,21], Jennifer Fransson[1,2], Bianca C. Kern[1,2], Ali Okhovat[1,2,3], Oscar E. Diaz[1,2], Chiara Sorini[1,2], Paulo Czarnewski[1,2,4], Anna T. Webb[5], Rodrigo A. Morales[1,2], Sacha Lebon[6], Gustavo Monasterio[1,2], Francisca Castillo[1,2], Kumar P. Tripathi[1,2], Ning He[1,2], Penelope Pelczar[7], Nicola Schaltenberg[7], Marjorie De la Fuente[8,9], Francisco López-Köstner[10], Susanne Nylén[11], Hjalte List Larsen[12], Raoul Kuiper[13,14], Per Antonson[15], Marcela A. Hermoso[9], Samuel Huber[7], Moshe Biton[6], Sandra Scharaw[5,20], Jan-Åke Gustafsson[15,16], Pekka Katajisto[5,17] & Eduardo J. Villablanca[1,2,18 ✉]

Uncontrolled regeneration leads to neoplastic transformation[1–3]. The intestinal epithelium requires precise regulation during continuous homeostatic and damage-induced tissue renewal to prevent neoplastic transformation, suggesting that pathways unlinking tumour growth from regenerative processes must exist. Here, by mining RNA-sequencing datasets from two intestinal damage models[4,5] and using pharmacological, transcriptomics and genetic tools, we identified liver X receptor (LXR) pathway activation as a tissue adaptation to damage that reciprocally regulates intestinal regeneration and tumorigenesis. Using single-cell RNA sequencing, intestinal organoids, and gain- and loss-of-function experiments, we demonstrate that LXR activation in intestinal epithelial cells induces amphiregulin (*Areg*), enhancing regenerative responses. This response is coordinated by the LXR-ligand-producing enzyme CYP27A1, which was upregulated in damaged intestinal crypt niches. Deletion of *Cyp27a1* impaired intestinal regeneration, which was rescued by exogenous LXR agonists. Notably, in tumour models, *Cyp27a1* deficiency led to increased tumour growth, whereas LXR activation elicited anti-tumour responses dependent on adaptive immunity. Consistently, human colorectal cancer specimens exhibited reduced levels of *CYP27A1*, LXR target genes, and B and CD8 T cell gene signatures. We therefore identify an epithelial adaptation mechanism to damage, whereby LXR functions as a rheostat, promoting tissue repair while limiting tumorigenesis.

Barrier tissues are continuously challenged by environmental stressors and must therefore adapt and establish programs designed to preserve tissue integrity. For example, after sensing damage, the intestinal barrier triggers molecular and cellular programs aimed at promoting tissue repair. To achieve this, intestinal stem cells (ISCs), located at the crypt bottom, engage in dynamic cross-talk with neighbouring cells (the ISC niche) that provides signals modulating ISC functions[6–9]. Such modulation is critical to the development of ISC-based regenerative therapies for chronic diseases that involve intestinal barrier loss, including inflammatory bowel disease and graft-versus-host disease[1,6,10,11].

However, excessive regeneration is strongly linked with neoplastic transformation[12], therefore limiting therapeutic strategies promoting regeneration without the risk of developing tumours. How ISC and their niche must adapt and trigger regenerative processes after injury while not compromising the final goal of a tumour-free system is unclear.

## LXR activation promotes intestinal regeneration

We hypothesized that pro-regenerative pathways shared between different tissues and injury models might represent a conserved regulatory

[1]Division of Immunology and Respiratory Medicine, Department of Medicine Solna, Karolinska Institutet and University Hospital, Stockholm, Sweden. [2]Center of Molecular Medicine, Stockholm, Sweden. [3]Structural Genomics Consortium, Division of Rheumatology, Department of Medicine Solna, Karolinska Institute and University Hospital, Stockholm, Sweden. [4]Science for Life Laboratory, Department of Biochemistry and Biophysics, National Bioinformatics Infrastructure Sweden, Stockholm University, Solna, Sweden. [5]Department of Cell and Molecular Biology, Karolinska Institutet, Solna, Sweden. [6]Department of Immunology and Regenerative Biology, Weizmann Institute of Science, Rehovot, Israel. [7]I. Medizinische Klinik, Universitätsklinikum Hamburg-Eppendorf, Hamburg, Germany. [8]Center of Biomedical Research (CIBMED), School of Medicine, Faculty of Medicine-Clinica Las Condes, Universidad Finis Terrae, Santiago, Chile. [9]Laboratory of Innate Immunity, Program of Immunology, Institute of Biomedical Sciences, Faculty of Medicine, Universidad de Chile, Santiago, Chile. [10]Centro de Enfermedades Digestivas, Programa Enfermedad Inflamatoria Intestinal, Clínica Universidad de Los Andes, Universidad de Los Andes, Santiago, Chile. [11]Department of Microbiology, Tumor and Cell Biology, Karolinska Institutet, Stockholm, Sweden. [12]Novo Nordisk Foundation Center for Stem Cell Medicine (reNEW), University of Copenhagen, Copenhagen, Denmark. [13]Section for Aquatic Biosecurity Research, Norwegian Veterinary Institute, Ås, Norway. [14]Department of Laboratory Medicine, Karolinska Institutet, Huddinge, Sweden. [15]Department of Biosciences and Nutrition, Karolinska Institutet, Huddinge, Sweden. [16]Center for Nuclear Receptors and Cell Signaling, Department of Biology and Biochemistry, University of Houston, Houston, TX, USA. [17]Institute of Biotechnology, University of Helsinki, Helsinki, Finland. [18]Clinical Immunology and Transfusion Medicine, Karolinska University Hospital, Stockholm, Sweden. [19]Present address: Robin Chemers Neustein Laboratory of Mammalian Cell Biology and Development, The Rockefeller University, New York, NY, USA. [20]Present address: Max Planck Institute of Molecular Cell Biology and Genetics, Dresden, Germany. [21]These authors contributed equally: Srustidhar Das, S. Martina Parigi, Xinxin Luo. ✉e-mail: srustidhar.das@ki.se; eduardo.villablanca@ki.se

mechanism to limit tumorigenesis. To identify a gut-regenerative core signature, we compared the transcriptomic landscape of tissue regeneration after irradiation- and dextran sodium sulfate (DSS)-induced damage[4,5] in the small intestine (SI) and colon with their steady-state transcriptomics signature. KEGG pathway analysis of 245 differentially expressed genes (DEGs) shared between DSS- and irradiation-induced injury models revealed conserved upregulation of cholesterol metabolism in response to damage (Extended Data Fig. 1a). Among the genes defining this term, four genes (*Abca1*, *Apoc3*, *Apoa1* and *Apoa4*) (Extended Data Fig. 1a (dashed box)) are known to be transcriptionally regulated by the nuclear receptor LXR[13–17]—a central regulator of cholesterol homeostasis. Using spatial transcriptomics, we observed a widespread induction of *Abca1* transcripts at 3 days post-irradiation (d.p.i.) compared with steady-state conditions (0 d.p.i.) in the SI (Fig. 1a). Using RNA in situ hybridization (RNAscope) analysis of *Lgr5* and *Abca1*, we observed induction of *Abca1* transcripts in SI crypts at 3 d.p.i. compared with steady-state conditions (0 d.p.i.) (Fig. 1b), which was confirmed by quantitative PCR (qPCR) in regenerating SI crypts after irradiation- and DSS-induced damage (Fig. 1c,d). To assess the functional role of LXR activation in response to damage, we fed wild-type (WT) mice with a diet either supplemented with a synthetic LXR agonist (GW3965) or a control diet (standard) and analysed intestinal tissues after damage (Extended Data Fig. 1b). As expected, the LXR target genes *Abca1* and *Abcg1* were induced in GW3965-fed mice compared with in standard-diet-fed mice, irrespective of the type of damage or region of the intestine (Extended Data Fig. 1c). We observed increased crypt area and reduced histopathological score in GW3965-fed compared with control-fed mice after irradiation- and DSS-induced damage, respectively (Fig. 1e,f and Extended Data Fig. 1d,e). To assess whether the observed protective effect was due to reduced damage or enhanced regeneration, we analysed crypt cell death and proliferation, respectively. By 1 d.p.i., we observed a comparable increase in cleaved caspase-3-positive (cCASP3[+]) cell numbers (Extended Data Fig. 1f) and a decrease in *Olfm4* (ISC marker) expression in both diet groups (Extended Data Fig. 1g), indicating analogous crypt damage after irradiation regardless of LXR activation in vivo. By contrast, GW3965-fed mice showed significant upregulation of *Olfm4* transcripts in SI crypts at 3 d.p.i. (Extended Data Fig. 1g) with a concomitant increase (around 10%) in surviving crypts (≥10 BrdU[+] cells per crypt) and an increased number of BrdU[+] cells per crypt at 5 d.p.i. compared with standard-diet-fed mice (Fig. 1g). Moreover, therapeutic administration of the GW3965 diet only during the recovery phase after DSS withdrawal led to increased colonic crypt cell proliferation (Fig. 1h) and colon length (Fig. 1i). This suggests that LXR activation promotes intestinal crypt cell proliferation rather than protection from damage. To functionally confirm the pro-regenerative role of LXR activation in response to damage, mice were treated with DSS for 7 days while being fed with standard or GW3965 diet. On day 9 or 10, distal SI crypts were cultured in vitro in the absence of additional stimuli and were assessed for their ability to form organoids. Using de novo organoid budding as a proxy for regenerative growth[18], we observed that in vivo LXR stimulation enhanced intestinal epithelial regenerative growth, as evidenced by an increased number of de novo buds per organoid and a significant relative enrichment of organoids with higher de novo buds (that is, with ≥4 buds per organoid) (Fig. 1j). Together, these results suggest that LXR activation is an adapted response to damage and is sufficient to bolster intestinal epithelial regeneration in the SI and colon.

To gain insights into the mechanism, we analysed the overall changes after 10 days of LXR activation in the absence of damage. The absolute numbers of major intestinal immune cell types were comparable between mice that were exposed for 10 days to either standard or GW3965 diet (Extended Data Fig. 2a,b). Moreover, single-cell RNA-seq (scRNA-seq) analysis of sorted intestinal epithelial (IEC, EPCAM[+]CD45[−]), immune (EPCAM[−]CD45[+]) and stromal (double-negative (DN), EPCAM[−]CD45[−]) cells did not reveal overt differences in cell composition between mice fed with standard or GW3965 diet (Extended Data Fig. 2c–f). Likewise, spatial transcriptomics analysis of the whole SI using non-negative matrix factorization[19] showed comparable spatial distribution of gene programs between GW3965- and standard-diet-fed mice (Extended Data Fig. 2g). To investigate whether LXR activation promoted immunomodulation as previously described[16,20,21], we analysed 'inflammatory response' genes (annotated using the Gene Ontology database)[22] in our scRNA-seq datasets. Despite the strong LXR activation in every cell type analysed, as seen by increased expression of cholesterol metabolism and transporter genes such as *Abca1*, *Abcg1* and *Srebf1* (Extended Data Fig. 2h), we observed comparable inflammatory tone in immune cells from standard-diet- and GW3965-treated mice (Extended Data Fig. 2i), therefore suggesting minimal effect on inflammatory immune responses after 10 days of LXR activation.

## LXR-induced epithelial AREG drives regeneration

We next focused on the epithelial cell compartment and analysed the villus and crypt length as a proxy for overall epithelial cell abundance using histological analysis (Extended Data Fig. 3a). Ten days of LXR activation in the absence of damage led to an increase in villus length (and, to a lesser extent, crypt height) (Extended Data Fig. 3a), without affecting ISC proliferation (Extended Data Fig. 3b,c). To functionally assess whether such previous LXR activation primed the epithelium to enhanced regeneration that might surface after injury, we performed intestinal organoid culture, recapitulating aspects of intestinal regeneration after damage[23]. Indeed, SI crypts from GW3965-diet fed mice plated in vitro without further treatment formed organoids with more buds compared with those from standard-diet-fed mice (Extended Data Fig. 3d), suggesting that synthetic LXR activation primes the epithelium for increased regenerative growth that surfaces after challenge (that is, when tested for organoid formation). In agreement, spatial transcriptomics datasets from SIs after damage showed increased abundance of factors associated with epithelial cells (NMF3) in GW3965-diet- compared to standard-fed mice (Extended Data Fig. 3e). To test whether LXR activation drives intestinal regeneration in an IEC-autonomous manner, we generated conditional knockout mice lacking both LXR isoforms in IECs (*Villin-cre;Nr1h3[fl/fl]Nr1h2[fl/fl]* (where *Nr1h3* and *Nr1h2* encode LXRα and LXRβ, respectively); hereafter, LXR[ΔIEC] mice)[24]. Intestines from the LXR[ΔIEC] mice did not display any phenotypic alteration at steady state, as evidenced by comparable SI and colon length (Extended Data Fig. 3f,i,j), SI crypt and villus length (Extended Data Fig. 3g) and SI crypt cell proliferation (Extended Data Fig. 3h). However, when exposed to irradiation-induced damage, LXR[ΔIEC] mice showed a significant decrease (-15%) in surviving crypts compared with the littermate controls (Fig. 2a). To mechanistically dissect how IEC-intrinsic LXR activation led to regeneration, we used organoid[25] cultures from SI crypts stimulated with GW3965 or vehicle (DMSO) in vitro. GW3965 treatment led to increased *Abca1* expression (Extended Data Fig. 4a), enhanced regenerative growth (that is, de novo buds per organoid) and a significant relative enrichment for organoids with higher de novo buds (that is, with ≥4 buds per organoid) (Fig. 2b). To rule out any effect of possible contaminating non-epithelial cells during crypt isolation and subsequent organoid culture, we established organoids using fluorescence-activated cell sorting (FACS)-sorted LGR5[+] ISC–Paneth cell co-culture and observed a similar increase in buds per organoid after LXR activation (Extended Data Fig. 4b), confirming that LXR activation in IEC is sufficient in inducing enhanced regenerative growth. Having observed an epithelial intrinsic pro-regenerative effect in vivo and in vitro, we further analysed the expression of niche factors involved in ISC maintenance, self-renewal and/or differentiation[18,26] in SI organoids and ex vivo SI crypts at different timepoints after irradiation. We observed that LXR activation led to comparable transcript levels of notch ligands (*Jag1*, *Jag2*, *Dll1* and *Hes1*) and a mild increase in epithelial *Wnt3* (Extended Data Fig. 4c,d), compared with the controls.

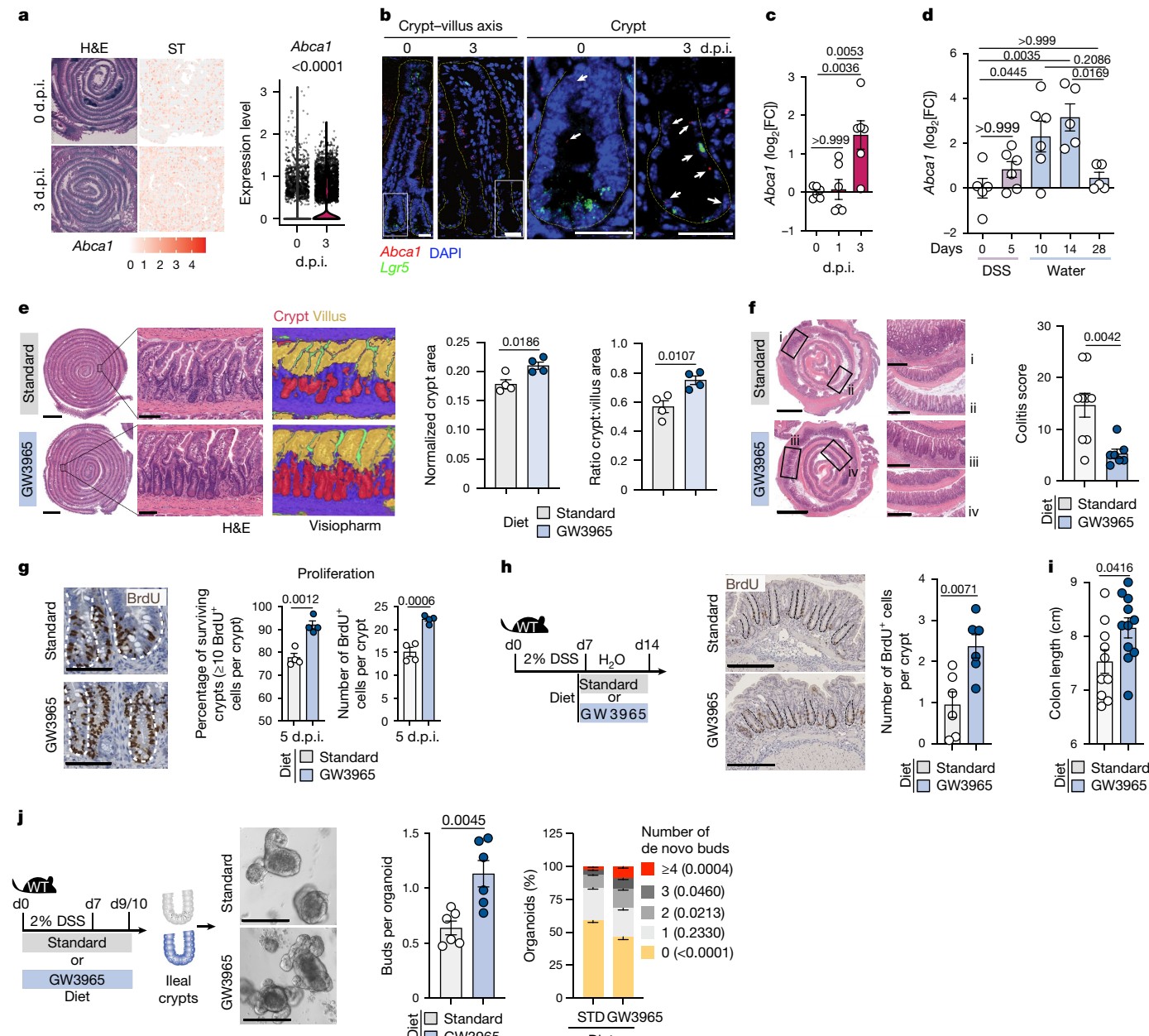

**Fig. 1 | LXR activation in vivo promotes intestinal regeneration in response to damage. a**, Haematoxylin and eosin (H&E) staining and *Abca1* transcript levels in 0 and 3 d.p.i. SI Swiss rolls. The *Abca1* levels in the tissue (left) and quantification per spot (right) are shown. ST, spatial transcriptomics. **b**, RNA scope analysis of *Abca1* and *Lgr5* transcripts at 0 and 3 d.p.i. in the SI. The white arrows in magnified crypts indicate *Abca1* transcripts. Scale bars, 25 µm. **c**,**d**, *Abca1* expression in SI crypts at 0, 1 and 3 d.p.i. (**c**) and ileal crypts of DSS-treated WT mice (**d**). **e**, H&E images and Visiopharm artificial intelligence deconvolution showing crypt (red) and villi (yellow) (left). Quantification of the normalized crypt area and crypt:villus area (right) of distal SI from standard (STD)-diet-fed and GW3965-fed WT mice at 5 d.p.i. is shown. Scale bars, 2 mm (left) and 100 µm (right). **f**, H&E images of colon Swiss rolls and histopathological assessment after recovery (day 14 (d14)) from DSS treatment in WT mice fed with standard or GW3965 diet. Scale bars, 2 mm (left) and 500 µm (right).

**g**, BrdU-stained images and quantification of the percentage of surviving crypts and BrdU+ cells per crypt in the distal SI at 5 d.p.i. Scale bars, 100 µm. **h**,**i**, Experimental schematic (left) and staining and quantification of BrdU+ cells per colonic crypt (right) (**h**) and colon length (**i**) at day 14 after DSS treatment. Scale bars, 200 µm (**h**). **j**, Schematic of the experiment (left). Representative ileal organoid (day 5) images with quantification of average buds per organoid and the percentage of organoids with the indicated number of buds. Scale bars, 200 µm. Data are representative of one (**a**,**b**), two (**d**–**h**) or three (**c**,**i**,**j**) independent experiments with 4–10 mice per condition (each dot is a biological replicate). For **c**–**j**, data are mean and s.e.m. *P* values were calculated using Wilcoxon tests (**a**), one-way analysis of variance (ANOVA) with Bonferroni's test (**c**,**d**), unpaired two-tailed *t*-tests (**e**–**j**) and two-way ANOVA with Bonferroni's test (de novo buds in **j**). The dashed lines denote crypts and villi in the same plane. The diagrams in **h** and **j** were adapted from ref. 19, CC-BY 4.0.

By contrast, we observed a consistent and significant increase in the expression of epidermal growth factor (EGF)-family ligands (that is, *Areg*, *Ereg*, *Tgfa* and *Egf*) after LXR activation in vitro and in vivo (Fig. 2c and Extended Data Fig. 4e). As the standard organoid culture medium includes EGF (a component of ENR, that is, EGF, noggin and R-spondin), we postulated that LXR activation might compensate for the lack of EGF if it were removed from the culture medium. Indeed, LXR activation resulted in enhanced regenerative growth in organoids cultured with NR medium (noggin, and R-spondin) (Extended Data Fig. 4f,g, and Fig. 2d–f) and downstream EGFR signalling[27,28] (Extended Data Fig. 4h).

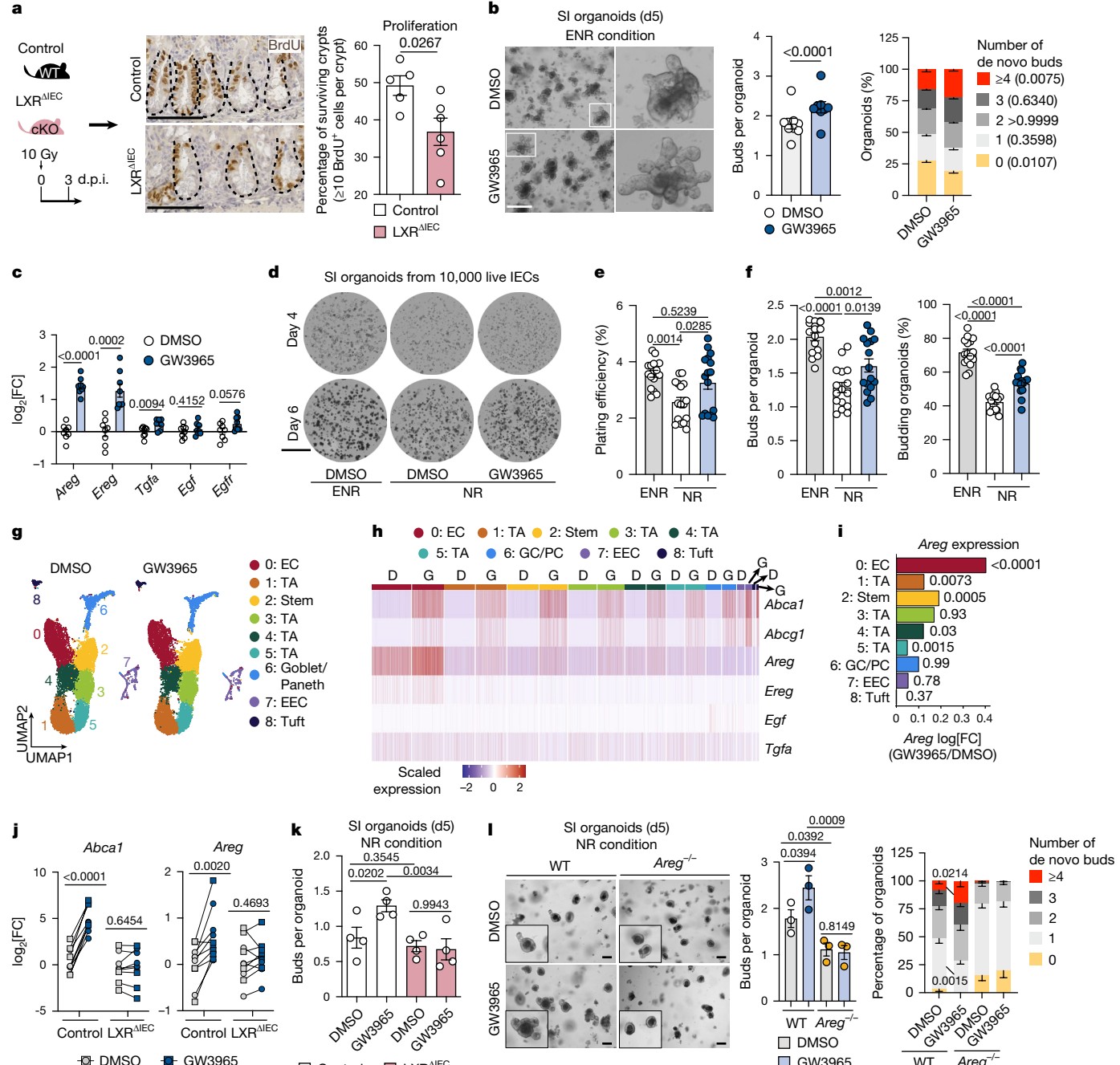

**Fig. 2 | LXR promotes intestinal regeneration by inducing epithelial amphiregulin. a**, Representative images and quantification of the surviving crypts in LXR^ΔIEC mice and littermate controls at 3 d.p.i. Scale bars, 100 μm. The diagram was adapted from ref. 19 CC-BY 4.0. **b**, Representative images and quantification of buds per organoid and organoids with the indicated number of buds. Scale bar, 400 μm. **c**, EGFR ligand expression in SI organoids cultured in ENR medium as indicated. **d**–**f**, Representative images (**d**) and quantification of the plating efficiency at day 4 (**e**) and the number of buds per organoid and the percentage of budding organoids at day 6 (**f**) in secondary organoids cultured as indicated. Scale bar, 1 mm (**d**). **g**–**i**, scRNA-seq analysis of organoids (*n* = 2 mice pooled per treatment) that were treated or not with GW3965 in NR medium. Uniform manifold approximation and projection (UMAP) cluster visualization (**g**), cluster-wise expression heatmap of the indicated genes (**h**) and cluster-wise relative fold change in *Areg* expression in GW3965 (G) over DMSO (D) (**i**). **j**, qPCR analysis of *Abca1* and *Areg* in LXR^ΔIEC and control organoids cultured in ENR (circle symbols) or NR (square symbols) medium.

**k**, The number of buds per organoid from LXR^ΔIEC and littermate controls with or without GW3965. **l**, Representative images, the number of buds per organoid and the frequencies of de novo buds of organoids from *Areg*^−/− and littermate controls cultured in NR medium. Scale bars, 100 μm. Data are from one (**g**–**i**), two (**a**), three (**d**–**f**,**l**), four (**b**,**c**,**k**) or nine (**j**) independent experiments with 3–9 mice per condition (each dot represents a biological replicate or well in **e** and **f**). For **a**–**c**,**e**,**f**,**j**–**l**, data are mean ± s.e.m. Significance was assessed using unpaired (**a**,**c**) or paired (buds per organoid in **b**,**j**) two-tailed *t*-tests; one-way ANOVA with Tukey's test (**e**,**f**) or Fisher's least significant difference (LSD) post hoc test (**k** and buds per organoid in **l**); two-way ANOVA with Bonferroni's post hoc test (de novo buds in **b**) or Fisher's LSD post hoc test (de novo buds in **l**). For **i**, significance was assessed by differential expression analysis using Wilcoxon rank-sum tests and subsampling of 150 cells per group. The dashed lines denote crypts in the same plane. E, EGF; N, Noggin; R, R-spondin; EC, enterocytes; GC, goblet cell; TA, transit amplifying cells; PC, Paneth cells; EEC, enteroendocrine cells.

This enhanced regenerative growth was observed in organoids grown from both crypts (Extended Data Fig. 4f) and single cells derived from primary organoids (Extended Data Fig. 4g and Fig. 2d–f). Notably, in the secondary culture from single cells, in addition to the increased organoid budding (Fig. 2d–f and Supplementary Videos 1–3), we also observed enhanced plating efficiency[29] when stimulated with GW3965 (Fig. 2d,e and Supplementary Videos 1–3). This further substantiates the pro-regenerative role of LXR activation in the intestinal epithelium. Together, these results show that LXR activation promotes the expression of EGFR family ligands, thereby driving intestinal epithelial regeneration.

Next, to unbiasedly investigate the mechanism(s) of LXR mediated pro-regenerative growth in organoids cultivated under NR condition, we performed scRNA-seq analysis of EPCAM[+] IECs from DMSO-treated ($n = 12,526$ cells) and GW3965-treated ($n = 11,665$ cells) SI organoids (Extended Data Fig. 5a). Unsupervised clustering revealed nine distinct clusters containing all major IEC types[4,30] (Fig. 2g and Extended Data Fig. 5b). Analysis of *Abca1* expression across clusters indicated that all IEC types responded to LXR activation, which did not promote a marked difference in cluster proportions (Fig. 2h and Extended Data Fig. 5c,d), suggesting that GW3965 does not induce bias towards specific cell types emanating from ISC in vitro. We next stratified the most highly differentially expressed genes across clusters (Extended Data Fig. 5e). As expected, GW3965 stimulation led to upregulation of genes involved in cholesterol transport (for example, ABC transporters such as *Abca1* and *Abcg1*) and lipid and fatty acid biosynthesis/metabolism (for example, *Lpcat3*, *Srebf1* and *Hmgcs2*) (Extended Data Fig. 5e). Notably, *Areg* was the most abundant and differentially expressed EGFR ligand after LXR activation (Extended Data Fig. 5e). A biased survey of the most common niche pathways (that is, WNT, notch and EGFR family ligands) in the scRNA-seq dataset also revealed *Areg* as the most prominent upregulated gene (Fig. 2h,i and Extended Data Fig. 5f), differentially expressed by a broad range of IECs (clusters 0–2 and 4–5) (Fig. 2i). In vivo, our spatial transcriptomics datasets from the irradiated intestine revealed that *Areg*[+] spots from 2 out of 12 clusters (clusters 4 and 5) showed the largest increase in *Areg* expression after LXR activation (Extended Data Fig. 6a,b) and were spatially restricted to the crypt region (Extended Data Fig. 6c). In agreement, the top genes that define clusters 4 and 5 were those that were primarily expressed by crypt-residing Paneth cells, located at the crypt bottom (Extended Data Fig. 6d). Furthermore, immunohistochemistry and RNA in situ hybridization analyses of irradiated SIs showed increased AREG protein (Extended Data Fig. 6e,f) and RNA (Extended Data Fig. 6g) expression after LXR activation in the crypt and crypt–villus junctional region, showing that LXR activation drives amphiregulin expression in pan-crypt IECs after injury. Finally, organoids from LXR[ΔIEC] mice did not upregulate *Areg* when stimulated with GW3965 (Fig. 2j) with concomitant abrogation of the regenerative growth (Fig. 2k and Extended Data Fig. 6h), therefore demonstrating that LXR activity in IECs is necessary to induce *Areg* expression and to promote intestinal regeneration.

To test whether AREG was necessary for the pro-regenerative effects of LXR, we used an AREG-neutralizing antibody (Extended Data Fig. 6i) in SI organoids from WT mice (Extended Data Fig. 6j) and generated organoids from crypts isolated from *Areg*[−/−] mice (Fig. 2l). We observed abrogation of LXR-induced pro-regenerative effect both when blocking (Extended Data Fig. 6j) or due to lack (Fig. 2l) of amphiregulin. A recent study has shown that EGFR ligands are involved in YAP1-mediated damage-induced regenerative program[31]. Although depletion of *Yap1* in SI organoids established from *Villin-creERT2+Yap1[fl/fl]Taz[fl/fl]* mice showed downregulation of YAP1-activated genes such as *Msln* and *Il33*[31], it did not abrogate LXR-induced *Abca1* or *Areg* (Extended Data Fig. 7a–f). This shows that LXR drives its regenerative program through *Areg* independently of YAP1.

Activation of LXR leads to expression of genes involved in multiple aspects of cholesterol metabolism such as efflux, biosynthesis and absorption[32]. Importantly, increasing cellular cholesterol either by adding exogenous/dietary cholesterol or by increasing cholesterol biosynthesis and uptake has been associated with enhanced IEC proliferation and consequently regeneration[33]. Thus, to test whether LXR-dependent regulation of cholesterol remodelling has a role in IEC regeneration, we treated LXR-sufficient and LXR-deficient organoids with cholesterol and assessed their regenerative potential. Consistent with a previous report[33], cholesterol treatment of SI organoids resulted in enhanced regenerative growth, with organoids exhibiting higher de novo buds (that is, with ≥4 buds per organoid) compared with the vehicle control (Extended Data Fig. 8a). However, LXR-deficient organoids did not show enhanced budding in the presence of cholesterol (Extended Data Fig. 8a). Together, these data suggest that LXR activation promotes regeneration through its role as a regulator of cellular cholesterol remodelling and through transcriptional control of pro-regenerative mediators (such as *Areg*).

Mirroring the effect of GW3965, organoids stimulated with an another LXR agonist RGX-104, currently in a multicentre phase I clinical trial in patients with advanced solid malignancies and lymphoma (NCT02922764), also led to increased budding and was accompanied by upregulation of *Areg* (Extended Data Fig. 8b,c). Moreover, single-cell replating of primary colonic epithelial organoids (Extended Data Fig. 8d) led to enhanced plating efficiency and organoid area (Extended Data Fig. 8e–g and Supplementary Videos 4 and 5) after LXR activation. This was accompanied by an increase in *Areg* expression (Extended Data Fig. 8h), indicating a regenerative growth pattern similar to that observed in SI organoids. Finally, in support of LXR activation as a more generalized mechanism of tissue regeneration, we observed a similar growth advantage and induction of Areg expression after LXR activation in vitro in salivary gland organoids[34] (Extended Data Fig. 8i). Consistent with the in vitro findings, in vivo LXR activation in a radiation-induced salivary gland injury model resulted in overall increase in the ductal area, a parameter used as a proxy to evaluate salivary gland regeneration (Extended Data Fig. 8j). Together, our data reveal a broader regenerative program driven by LXR-induced amphiregulin in epithelial cells.

## CYP27A1 acts upstream of LXR

To gain insights into the endogenous ligands that might activate LXR after damage, we examined pathways that produce oxysterols[35], which are generated from cholesterol by the members of the cytochrome P450 monooxygenase family *Cyp7a1*, *Cyp46a1*, *Cyp11a1* and *Ch25h*[35,36] (Fig. 3a). Revisiting the scRNA-seq datasets from regenerating SI crypts[4] (Fig. 3b) and our published RNA-sequencing (RNA-seq) dataset using colonic biopsies[5] (Fig. 3c) together with qPCR on ileal crypts over the course of DSS-induced injury–repair (Fig. 3d), we observed a selective increase in *Cyp27a1* expression during the regenerative phase, whereas other enzymes were either undetectable or upregulated across the whole response. Similarly, in the SI, immunohistochemical analysis showed induction of CYP27A1 after irradiation-induced intestinal injury (Fig. 3e and Extended Data Fig. 9a–d). Although the majority of CYP27A1[+] cells in the SI were found in the villi, after irradiation, CYP27A1 was initially expressed in the crypt epithelium (1 d.p.i.) and later (3–5 d.p.i.) in the lamina propria (LP) adjacent to the crypts (Extended Data Fig. 9b–d), suggesting local production of 27-hydroxycholesterol (27-HC). Consistently, analysis of *Cyp27a1* expression in FACS-sorted epithelial (EPCAM[+]CD45[−]), immune (EPCAM[−]CD45[+]) and DN (EPCAM[−]CD45[−]) cells from SIs of mice undergoing irradiation-induced injury–repair (Extended Data Fig. 9e–g) showed a significant upregulation in IECs by 1 d.p.i. (Fig. 3f) and was accompanied by a concurrent increase in LXR target gene *Abca1* (Fig. 3g). We therefore investigated whether CYP27A1 was necessary to promote LXR-dependent intestinal regeneration in vivo. Sorted IECs from whole-body-irradiated *Cyp27a1[−/−]* mice showed reduced

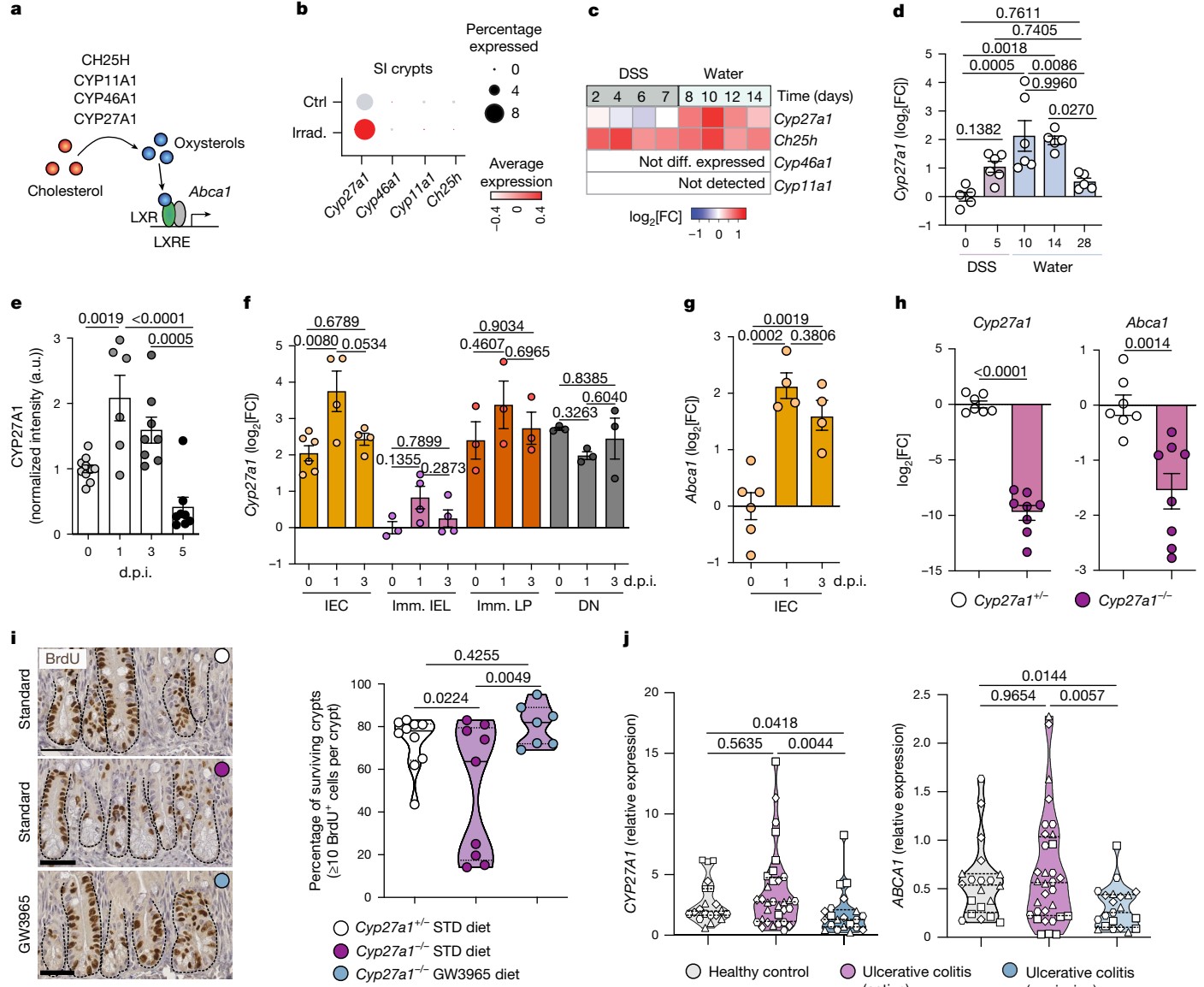

**Fig. 3 | CYP27A1 is upregulated during intestinal injury and promotes intestinal regeneration. a**, Schematic of cholesterol metabolism to oxysterols by the cytochrome P450 monooxygenase family of enzymes (CYPs) and its downstream action on LXR. **b**, The average expression (colour coded) and the percentage of cells (circle size) expressing oxysterol-producing enzymes in SI crypts at 0 (Ctrl) or 3 d.p.i. (irrad.)[4]. **c**, Heatmap of oxysterol-producing enzyme expression in whole-colonic tissue during DSS kinetics[5] (log₂-transformed fold change compared with day 0). Diff., differentially. **d**, qPCR analysis of *Cyp27a1* in ileal crypts from DSS-treated WT mice. **e**, Immunohistochemical quantification of CYP27A1 in the SI of WT mice at the indicated d.p.i. **f**, qPCR analysis of *Cyp27a1* in FACS-sorted EPCAM⁺ epithelial (IEC), CD45⁺ immune (imm. IEL and imm. LP) and EPCAM⁻ CD45⁻ DN cells from WT mouse SI collected at the indicated d.p.i. **g**, qPCR analysis of *Abca1* in FACS-sorted IECs from WT mouse SI collected at the indicated d.p.i. IEL, intraepithelial lymphocyte; LP, lamina propria. **h**, qPCR analysis of *Cyp27a1* and *Abca1* in FACS-sorted IECs from

*Cyp27a1⁻/⁻* mice and littermate control SI at 3 d.p.i. **i**, Representative images and quantification of BrdU⁺ surviving crypts from the SI at 3 d.p.i. Crypts in the same plane are marked by a dashed line. Scale bars, 50 μm. **j**, qPCR analysis of *CYP27A1* and *ABCA1* in human intestinal biopsies (collected from terminal ileum (square symbols), ascending colon (triangle symbols), transverse colon (diamond symbols) and sigmoid rectum (hexagon symbols)) from healthy control individuals (*n* = 28), and patients with active ulcerative colitis (UC) (*n* = 39) and ulcerative colitis in remission (*n* = 27). Data are representative of two (**d**,**f**–**h**) or three (**e**,**i**) independent experiments with 3–10 mice per condition (each dot represents one biological replicate). Data are mean ± s.e.m. (**d**–**h**) and median ± quartiles (**i**,**j**). Significance was assessed using unpaired two-tailed *t*-tests (**h**), one-way ANOVA with Tukey's post hoc test (**d**–**g**), Fisher's LSD post hoc test (**i**) and nonparametric Kruskal–Wallis test with uncorrected Dunnett's post hoc test (**j**).

*Cyp27a1*, with a concomitant reduction in *Abca1* compared with the littermate controls (Fig. 3h), suggesting that lack of *Cyp27a1* dampens LXR activity in IECs. After irradiation, *Cyp27a1⁻/⁻* mice displayed reduced surviving crypts compared with their littermate controls, and LXR activation by GW3965 diet administration rescued crypt cell proliferation (Fig. 3i). Furthermore, *Cyp27a1⁻/⁻* mice displayed reduced colon length (Extended Data Fig. 9h) and worsened histological score (Extended Data Fig. 9i) after DSS-induced damage compared with the

littermate controls. However, given the known role of CYP27A1 in regulating bile acids and cholesterol absorption[37], it remains possible that whole-body deletion of *Cyp27a1* compromises intestinal regeneration not solely by affecting LXR signalling. Finally, to test whether this axis was a conserved response to injury in humans, we observed increased expression of both *CYP27A1* and *ABCA1* in the terminal ileum and colon (ascending, transverse and sigmoid rectum) of patients with active ulcerative colitis[38] compared with in patients in remission (Fig. 3j).

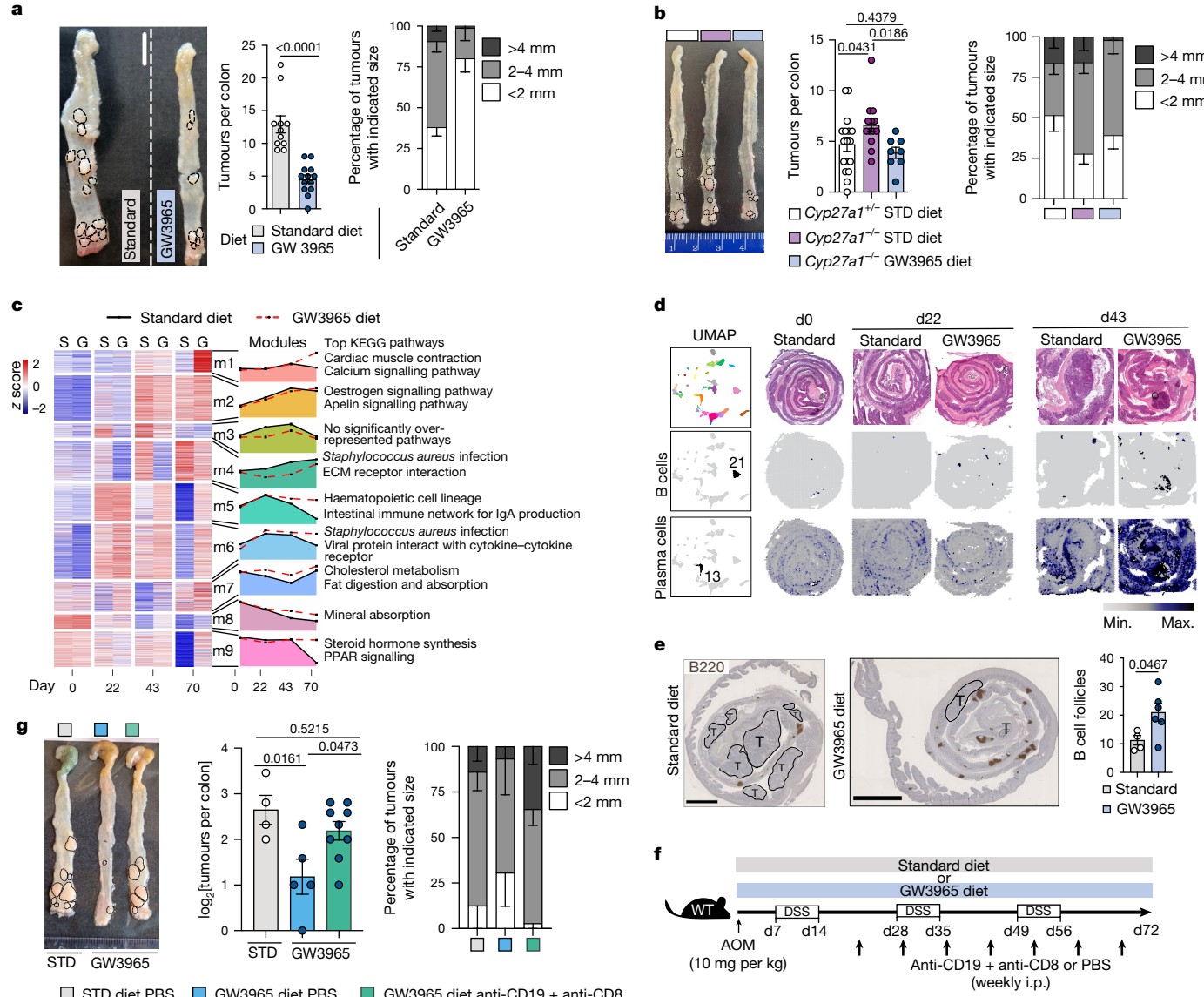

**Fig. 4 | LXR activation in vivo suppresses intestinal tumorigenesis.**
**a**, Representative images and quantification of tumour numbers and size at day 70 of AOM–DSS-induced tumorigenesis in WT mice that were fed a standard or GW3965 diet. Scale bar, 1 cm. **b**, Representative images and quantification of tumour numbers and size at day 70 of AOM–DSS-induced tumorigenesis in *Cyp27a1*$^{-/-}$ and littermate controls fed as indicated. **c**, Longitudinal bulk RNA-seq analysis of mouse colon from AOM–DSS WT mice fed with standard (S) or GW3965 diet. Clustered heatmap of DEGs for the diet or diet:time interactions (likelihood ratio test, false-discovery rate (FDR) < 0.05), divided into modules of similar gene behaviour (left). Mean-scaled log-fold-change (compared with day 0) of each gene module (m1–m9) for each timepoint/diet group and the top KEGG pathways significantly enriched (adjusted *P* < 0.05) (right). **d**, The scores of selected B cell and plasma cell clusters from quantitative deconvolution of scRNA-seq data[49] onto colon Swiss roll spatial transcriptomics from day 0 (standard diet), day 22 and day 43 (standard and GW3965 diet) AOM–DSS-treated mice.

**e**, Representative images and quantification of B220 immunohistochemical staining of colons from AOM–DSS treated WT mice that were fed with standard or GW3965 diet at day 70. Scale bars, 2.5 mm. **f**,**g**, Schematic (**f**) and representative images and quantification of tumour numbers (log$_2$ normalized) and sizes (**g**) from AOM–DSS-treated WT mice fed on a standard or GW3965 diet and treated with repeated intraperitoneal injections of anti-CD19 and anti-CD8 antibodies or PBS as shown in the schematic. Data are representative of 2–4 (**a**,**b**,**e**,**g**) independent experiments with 4–13 mice per condition (each dot represents one biological replicate); one experiment with 3–4 mice per timepoint for bulk RNA-seq (**c**); one mouse per timepoint from one experiment for spatial transcriptomics (**d**). For **a**,**b**,**e**,**g**, data are mean ± s.e.m. Significance was assessed using two-tailed unpaired *t*-tests (**a**,**e**) or one-way ANOVA with fisher's LSD (**b**) and Tukey's (**g**) post hoc test. The dashed lines denote tumours. The diagram in **f** was adapted from ref. 19, CC-BY 4.0.

Collectively, these results revealed an ISC-niche adaptation to damage whereby *Cyp27a1*, upregulated in response to injury, drives LXR activation in IEC and promotes intestinal regeneration.

## LXR activation inhibits tumorigenesis

Pathways used by ISC to self-renew and regenerate during barrier repair can be hijacked by cancer stem cells, leading to uncontrolled

proliferation[1–3]. To investigate the role of LXR in intestinal tumorigenesis, *Apc*$^{Min/+}$ mice were fed with GW3965 or standard diet and analysed for adenoma formation. Consistent with previous findings of mice engineered to express a constitutively active form of LXRα specifically in epithelial cells[39], *Apc*$^{Min/+}$ mice fed with the GW3965 diet resulted in reduced tumour numbers and size compared with the control mice (Extended Data Fig. 10a–c). Likewise, extending our findings to the azoxymethane (AOM)–DSS inflammation-driven colorectal

cancer (CRC) model, we observed reduced tumour numbers and size after LXR activation (Extended Data Fig. 10d,e and Fig. 4a). Furthermore, *Cyp27a1*[-/-] mice treated with AOM–DSS displayed more and larger tumours compared with their littermate controls and GW3965 administration in *Cyp27a1*[-/-] mice rescued this phenotype (Fig. 4b and Extended Data Fig. 10f), suggesting that LXR activation downstream to CYP27A1 restrains intestinal tumorigenesis. Extending our findings to humans, we further confirmed downregulation of CYP27A1 in the colonic epithelium from patients with CRC compared with healthy individuals (Extended Data Fig. 10g,h), suggesting that suppression of the CYP27A1–LXR axis might function as a tumour-escape mechanism. The mechanisms involved in the reciprocal regulation of CYP27A1 in the context of regeneration and tumorigenesis remain to be addressed and constitute an important area for future investigation. Enhanced expression of EGFR ligands including amphiregulin are observed in multiple human malignancies including CRC[40]. However, despite LXR-activation-induced *Areg* expression in response to damage, we did not observe induction of EGFR ligands in the context of tumorigenesis (Extended Data Fig. 10i). To further investigate LXR-mediated anti-tumour mechanism(s), we performed bulk RNA-seq analysis of the colonic tissue in the presence or absence of GW3965-mediated LXR activation at different timepoints during AOM–DSS tumorigenesis (Fig. 4c). Timepoints were chosen based on the analysis of macroscopic tumours—day 22 showed no tumours, day 43 showed tumours almost exclusively in control mice and day 70 showed tumour development in both GW3965- and standard diet-fed mice, albeit to a different extent (Extended Data Fig. 10d,e,j,k). Longitudinal analysis combining differential gene expression (DEGs in diet and/or diet:time interaction) and hierarchical clustering[5] resulted in nine distinct modules (Fig. 4c). Each module was defined by a set of genes displaying similar expression pattern across timepoints and diet groups, and we used KEGG pathway analysis to identify which pathways were enriched in each module (Fig. 4c). Among the modules, m5 (enriched for pathways annotated as haematopoietic cell lineage and intestinal immune network for IgA production) started to diverge and was upregulated in GW3965-fed mice after day 22 onwards (Fig. 4c), mirroring the difference observed in overall tumour growth between the two groups (Extended Data Fig. 10j,k). Moreover, m5 showed comparable expression between standard diet- and GW3965-diet-fed mice at day 0 (that is, mice fed either standard or GW3965 diet for 22 or 43 days without any AOM–DSS treatment), arguing that LXR-dependent activation of this gene program is fuelled by the tumorigenesis process (Fig. 4c). Organization of immune cells in tertiary lymphoid structures (TLSs) is associated with favourable prognosis in patients with CRC (as reviewed previously[41,42]). To evaluate whether the sustained expression of m5 genes in GW3965-fed mice correlated with enhanced TLS formation and plasma cell signature in the tissue, we performed spatial transcriptomics analysis of the colonic tissue of standard-diet- and GW3965-fed mice at day 22 and day 43 of AOM–DSS treatment (Extended Data Fig. 10l). LXR activation in the context of AOM–DSS treatment promoted the formation of B-cell-signature-enriched TLS at day 22 and more prominently at day 43, and the widespread expression of plasma cell signatures in the GW3965-treated colon at day 43 compared with the standard-diet-fed controls (Fig. 4d). Furthermore, immunohistochemical staining of the B cell marker B220 revealed a significant increase in B cell follicles in the colon of GW3965-fed compared with in standard-diet-fed mice (Fig. 4e) at day 70 of AOM–DSS treatment. TLSs have been described as sites of induction of adaptive anti-tumour immune responses including expansion and/or infiltration of cytotoxic CD8 T cells[43–46]. To assess whether expansion/activation of adaptive immunity functionally underlined the reduced tumour formation observed after LXR activation, we treated GW3965- and standard-diet-fed mice with anti-CD19 and anti-CD8 antibodies (together or separately) or isotype control or PBS starting at day 22 and at repeated timepoints over the course of AOM–DSS

administration (Fig. 4f,g and Extended Data Fig. 11a,b). After GW3965 treatment, simultaneous blocking of CD19 and CD8 significantly increased tumour formation compared with the control-treated mice (Fig. 4f,g), demonstrating that B and T lymphocytes contribute to the LXR-mediated anti-tumour response. To examine the potential relevance of LXR activation in human CRC, we analysed the expression of *CYP27A1*, LXR target genes and TLS/B and T cell signatures in a published dataset containing 443 CRC samples divided into six tumour subtypes (c1 to c6)[47]. Expression of the LXR ligand producing enzyme *CYP27A1*, LXR target gene *ABCG1* (but not *ABCA1*) and TLS/B and T cell signatures (*MS4A1*, *CD19*, *CR2*, *CD8a*, *GZMB* and *CD69*) was decreased in tumours compared with non-tumour samples (Extended Data Fig. 11c), suggesting that suppression of the CYP27A1–LXR–TLS/B cell axis might function as a tumour-escape mechanism in human CRC.

## Discussion

Overall, our work proposes a mechanism of disease tolerance[48] in ISCs and their niche whereby LXR activation functions as a rheostat that enables adaptation to barrier damage without compromising the host fitness by reducing the risk of tumorigenesis. We found that, whereas LXR activation in the context of intestinal injury promotes regeneration by modulating ISC-niche derived signals such as AREG, it puts a check on the uncontrolled proliferation of the otherwise transformed cells by eliciting tumour suppressive mechanisms mediated by adaptive immune cells. Notably, such responses seem to be locally choreographed by the upregulation of CYP27A1 triggered by barrier damage. The risk of neoplastic transformation in the aftermath of regeneration must be carefully considered for the development of safe therapies against chronic intestinal disorders. LXR activation may represent one such pathway conserved across tissues, injury models and species that balances these processes to preserve homeostasis. Our results showed that LXR signalling reciprocally regulates intestinal regeneration and tumorigenesis (Extended Data Fig. 12).

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

# Methods

## Mice

All of the mice used in this study were on the B6 genetic background. WT C57BL/6J mice were purchased from TACONIC. *Nr1h3*[fl/fl]*Nr1h2*[fl/fl] mice, provided by J.-Å. Gustafsson (Karolinska Institutet), were crossed with *Villin-cre* mice (*B6.SJL-Tg(vil-cre)997Gum/J*, provided by J.-Å. Gustafsson; and *B6.Cg-Tg(Vil1-Cre)1000Gum/J*, obtained from Jackson laboratory) to generate *Nr1h3*[fl/fl]*Nr1h2*[fl/fl] and *Villin-cre;Nr1h3*[fl/fl]*Nr1h2*[fl/fl] littermate controls. *Cyp27a1*[−/−] (*B6.129-Cyp27a1*[tm1Elt]*/J*) mice were purchased from Jackson Laboratory and bred locally as *Cyp27a1*[+/−] × *Cyp27a1*[+/−] or *Cyp27a1*[+/−] × *Cyp27a1*[−/−] breeding pairs. *Lgr5-eGFPIRES-creERT2* mice[50] were maintained on a C57BL/6J background. *Villin-creERT2 Yap1*[fl/fl] *Taz*[fl/fl] and *Yap1*[fl/fl]*Taz*[fl/fl] control mice were provided by H. L. Larsen (University of Copenhagen). LXR-sufficient and *Cyp27a1*-sufficient controls for deficient mice were littermates and co-housed (unless specifically indicated). Mice, both males and females, were generally used between 8 and 20 weeks of age. Mice were maintained under specific-pathogen-free conditions at Karolinska Institutet, except for some experiments with *Cyp27a1*[−/−] and littermate controls in which some of the mice were housed in an MPV-positive animal facility (the experiments with DSS administration and half of the irradiation experiments performed on *Cyp27a1*[−/−] and littermate controls). *Apc*[Min/+] mice (*C57BL/6J-Apc*[Min]*/J*), provided by S. Huber (some of the *Apc*[Min/+] mice were on a *Hmgb1*-floxed background without any endogenous Cre), were maintained under specific-pathogen-free conditions at Hamburg-Eppendorf University (Hamburg). *Areg*[−/−] mice were provided by M. Biton and were maintained under specific-pathogen-free conditions at Weizmann Institute of Science (Israel). When administered in vivo, GW3965 (Adooq Bioscience) was administered through formulated drug in diet at 50 mg per kg per day (Research Diet and SSNIFF). The same purified based diet (D11112201, Research Diet or AIG93G + Inulin, SSNIFF), used to formulate GW3965-diet, was used in standard-diet fed mice. BrdU was prepared at 10 mg ml$^{-1}$ in PBS, sterile filtered through a 0.22 µm filter and injected at 100 mg per kg body weight 2 h before euthanasia. No randomization or blinding was used. Sample sizes were determined based on pilot and preliminary experiments. All of the experimental procedures were performed in accordance with national and institutional guidelines and regulations. Animals used for experiments performed at Karolinska Institutet (Stockholm, Sweden) were maintained under specific-pathogen-free conditions at Karolinska Institutet (Stockholm, Sweden) animal facilities and were handled according to protocols approved by the Stockholm Regional Ethics Committee.

## Colitis model

Colitis was induced by administration of 2–3% (w/v) DSS (TdB Consultancy, molecular weight 40 kDa) dissolved in drinking water ad libitum for 6–7 days, followed by regular drinking water. Body weight loss and mouse health status were monitored regularly. At the indicated timepoints, the colon length was measured, 0.5 cm of distal colonic biopsies was collected for RNA extraction (when indicated) and the colonic tissue was cut open longitudinally, Swiss-rolled and fixed in 10% buffered formalin for histological analysis. For ileal organoids and gene expression analyses in ileal crypts after DSS administration, distal third of the SI (that is, ileum) was collected at the indicated timepoints and processed for crypt isolation as described below in the 'SI organoids' section. The colitis score was assessed in a blinded manner in paraffin-embedded sections stained with H&E and calculated using a TJL-based system according to the following formula: (degree of severity + degree of hyperplasia + degree of ulceration) × percentage area involved. Each parameter was scored from 0 to 3 according to the following description: (a) severity: 0, unaffected; 1, single/widely scattered, LP involved; 2, larger or involving submucosa; 3, ulcers longer than 20 crypts; (b) hyperplasia: 0, normal; 1, 1–2× normal thickness; 2, 2–3× normal thickness, hyperchromasy, reduced goblet cells, elongated crypts, increased mitoses; 3, 4× normal thickness, hyperchromasy, reduced goblet cells, elongated crypts, high mitotic index, crypt branching; (c) ulceration: 0, no ulcers; 1, 1–2 ulcers, less than 20 crypts; 2, 1–4 ulcers, 20–40 crypts; 3, >4 ulcers, >40 crypts; (d) percentage of area involved: 0, 0%; 1, 10–30%; 2, 40–70%; 3, >70%.

## Mouse tumour models

**AOM–DSS model.** Mice were intraperitoneally (i.p.) injected with 10 mg per kg of body weight AOM (Sigma Aldrich) resuspended at 10 mg ml$^{-1}$ in water, which was further diluted to 1 mg ml$^{-1}$ in PBS before injecting in mice. Administration of modified diet (standard or GW3965 diet ad libitum) was started on the same day and continued for the entire duration of the experiment. Then, 1 week later, 1.5–2% (w/v) DSS (TdB Consultancy, molecular weight 40 kDa) was administered in the drinking water for 7 days followed by 2 weeks of regular water. The same DSS treatment was repeated for two more cycles (three cycles in total), each time with an interval of 2 weeks of regular water in between. Mice were euthanized at the indicated timepoints and colon samples were collected. Anti-CD19 (300 µg per mouse per injection, BioXcell, 1D3) and anti-CD8β (300 µg per mouse per injection, BioXcell, Lyt3.2) were treated alone or in combination i.p. every 7 days starting from day 22 (that is, after the first round of DSS treatment) until the end of the experiment. Control mice were injected 300 µg per mouse per injection with isotype control (rat IgG2a, BioXcell) or PBS. The tumour number and size were evaluated manually using a digital micro-caliper. Tissue biopsies for RNA extraction were collected from the distal colon (corresponding to tumour areas). Colon tissues were then Swiss-rolled for histological analysis. Histological scoring of tumours was performed in a blinded manner using the H&E stained Swiss-rolls in which the tumour areas were categorized into either well-differentiated, well-moderately differentiated or moderately differentiated. The dysplastic regions seen in the tissues were from the non-tumour area and not the dysplasia within the tumour.

**Apc$^{Min/+}$ model.** *Apc*[Min/+] mice (aged 4–5 weeks) were given the modified diet (standard or GW3965-diet) ad libitum for the entire duration of the experiment. Mice were euthanized at around 4 months of age and macroscopic tumours were measured and counted manually.

## Total-body irradiation

Adult mice were exposed to a dose of 10 Gy from an X Radiation Unit (CIX3 X-Ray Cabinet or with an X-RAD 320 irradiation source with 20 × 20 cm irradiation field within a mouse pie) and euthanized at the indicated timepoints. At the indicated timepoints, SI tissues were either Swiss rolled for histological analyses, Visium processing or SI crypts were isolated as described below in the 'SI organoids' section. Histological analyses of H&E-stained tissues embedded in paraffin was conducted using Visiopharm (v.23.01), an artificial-intelligence-powered pathology program that uses a trainable algorithm based on a convolutional neural network. The regions of interest (ROIs) within the Swiss roll (that is, the total measured area) were manually selected to exclude damaged areas caused by tissue rolling artefacts. Subsequently, the crypt and villus area within these selected ROIs were quantified. Crypt hyperplasia was evaluated using normalized crypt area (that is, crypt area/total area) and the crypt area-to-villus area ratio.

## Radiation-induced salivary gland injury

Mice at the indicated timepoints were first anaesthetized in the isoflurane chamber (1% $O_2$, 1% air and 5% flow rate), and, once asleep, were transferred to the irradiator bed with 0.3% $O_2$, 0.3% air and 1.5% flow rate until the irradiation protocol was finished. The mice were irradiated individually using X-ray irradiator CIX3 (Xstrahl) with only the neck exposed using 1 cm focal tube and rest of the body and head shielded. An X-ray irradiation dose of 9 Gy (that is, 300 kV, 10 mA for a period

of 7.04 min) was administered. After irradiation, the mice were monitored until they were fully conscious and awake. Non-irradiated control mice were subjected only to the anaesthesia protocol without being irradiated. Mice were euthanized at 14 d.p.i. and salivary glands were collected and fixed in 10% buffered formalin for histological analysis.

## Histological quantification of salivary gland tissues

To get automated quantification of salivary gland histology, a deep learning algorithm was developed to automatically segment and classify ducts, blood and air from histological tissue samples. For dataset preparation, histological images underwent tile extraction and preprocessing, following the NoCodeSeg framework (35155486). The dataset included 43 training patients and 1 testing patient. Tiles of $512 \times 512$ pixels in size were obtained using QuPath. The training dataset comprised 35,506 objects, addressing class imbalances through augmentation. For deep learning model training, a deep learning model, based on the Xception architecture[51] (https://arxiv.org/abs/1610.02357), was trained using the MIB[52] MATLAB package. Parameters were saved for the model, and training used $512 \times 512$ pixel patches. The validation set was split randomly for training (97%) and validation (3%). For model evaluation, post-training, the model's performance was assessed using loss curves and a confusion matrix on the test set. For application of trained model, the trained model was integrated into FastPathology for practical application. This involved copying the model and pipeline files, creating a new project, adding images and selecting the pipeline for processing. Annotations in QuPath were loaded into QuPath by downloading and running the provided script for each image in the project. More details about the training and inference of the model can be found on the GitHub repository (https://github.com/BIIFSweden/Eduardo-Villablanca2023-1). Outliers were identified and removed from the imported data in R using the boxplot function. Subsequently, the combined GW0 ($n = 2$) and STD0 ($n = 2$) datasets were analysed, yielding summary statistics. Bin analysis was performed using breaks delineated at 0 to first quartile, first quartile to mean, mean to third quartile and third quartile to the maximum, labelled as noise, small, medium and large, respectively. Noise-tagged ducts were filtered out, and their distributions were visualized through violin plots. The mean and s.d. of duct areas were computed for each biological replicate, followed by the application of Welch two sample $t$-tests to compare treatments.

## SI organoids

Mouse SI organoids were obtained from purified crypts derived from the entire SI, unless specified. In brief, SI tissue was flushed with cold PBS, cleaned from attached fat tissue, cut opened longitudinally and subsequently cut into approximately 0.5 cm pieces. Tissues were incubated in 30 ml of 10 mM cold PBS-EDTA on ice for the following incubation periods: 10 min, $3 \times 15$ min and 1 h. After each incubation, tubes were gently shaken, supernatant containing tissue debris and villi fractions were discarded and fresh PBS-EDTA was added. After the last incubation, the tubes were vigorously shaken and filtered through 70 µm cell strainers and isolated SI crypts were retrieved in the flow-through. The last step was repeated until no more crypts were present in the flow-through. Alternatively, SI tissue, flushed with PBS and cut opened longitudinally, was cut into 4–5 pieces (approximately 7 cm long) and incubated for 1 h in ice cold 10 mM PBS-EDTA. Using two glass slides, intestinal villi were then removed by gentle scraping of the luminal side. SI crypts were then scraped by applying stronger pressure with the glass slides and collected in recipient tubes filled with cold PBS. Crypts were centrifuged at 4 °C, $300g$ for 5 min. The basic culture medium (ENR) contained advanced DMEM/F12, $1\times$ penicillin–streptomycin, $1\times$ GlutaMAX (Thermo Fisher Scientific), 10 mM HEPES (Thermo Fisher Scientific), $1\times$ B27 (Life Technologies), $1\times$ N2 (Life Technologies), 1 mM *N*-acetylcysteine (Sigma-Aldrich) and was supplemented with 50 ng ml⁻¹

of murine recombinant epidermal growth factor (that is, EGF, from R&D), 250–500 ng ml⁻¹ recombinant murine R-spondin (R&D) and 100 ng ml⁻¹ recombinant murine Noggin (Peprotech). For experiments with NR medium, the same medium as above with the exception of EGF was prepared. SI crypts were resuspended in 30–40% basic culture medium with 60–70% Matrigel (Corning) and 20 µl containing approximately 300–500 crypts were plated in a prewarmed flat-bottom 48-well plate. The plate was placed at 37 °C and allowed to solidify for 15 min before 200 µl of ENR or NR medium (containing the different stimuli) was overlaid. The medium was replaced every 2 days with fresh medium and the cultures were maintained at 37 °C in fully humidified chambers containing 5% $CO_2$. Only on the first 2 days of culture, 10 µM Y-27632 (Sigma-Aldrich) was added in the medium. For in vitro stimulation, LXR agonist GW3965 (1 µM, Sigma-Aldrich) or RGX-104 (1 µM, MedChemExpress) was added in the medium for the entire duration of the organoid culture. DMSO was used as vehicle. For rAREG and anti-AREG experiment, rAREG (50 ng ml⁻¹, Peprotech) and/or anti-AREG (1.5 µg ml⁻¹, R&D Systems) were added in the medium for the entire duration of the organoid culture. For experiments with cholesterol, water soluble cholesterol (50 mM, Sigma-Aldrich) was added into the medium for the entire duration of the organoid culture. For organoids from *Villin-creERT2 Yap1^{fl/fl}Taz^{fl/fl}* mice, tamoxifen (1 µM, Sigma-Aldrich) treatment was performed in secondary organoids as indicated with or without GW3965 treatment. In brief, primary organoids from *Villin-creERT2 Yap1^{fl/fl}Taz^{fl/fl}* mice grown under either ENR or NR conditions were recovered from Matrigel by two washes with 0.1% BSA PBS. The organoids were mechanically fragmented by passing (30 times) through a 200 µl pipette tip. Equal amounts of organoid fragments were plated in 20 µl Matrigel (70% Matrigel) dome in a prewarmed 48-well plate and cultured for 5–8 days as indicated. Secondary organoids were cultured using either ENR or NR medium containing DMSO or GW3965 (1 µM). Y-27632 (10 µM, Sigma-Aldrich) was added only during the first 2 days of the culture, with medium change done every 2–3 days afterwards. Tamoxifen (1 µM; Sigma-Aldrich) was added at the indicated timepoints. For ileal organoids after DSS treatment, mice were given 2% DSS in the drinking water for 7 days followed by normal drinking water. Two to three days after DSS removal, the crypts were isolated from the distal third of the SI (that is, ileum) and organoids were cultured in either ENR or NR medium as described above. Crypt domain budding was quantified under the microscope at the indicated days in a blinded manner. Each condition was plated in triplicates and 2–3 wells per condition were quantified. Each dot in the quantification plot represents one mouse and an average of 2–3 technical replicates. Furthermore, organoids were stratified based on the number of buds/organoid (0, 1, 2, 3 or ≥4) and plotted as the percentage of organoids per well. Organoids were collected at the indicated timepoints and stored in RLT-Plus buffer (Qiagen) + 2% β-mercaptoethanol (Gibco) at −80 °C until RNA extraction.

## Single-cell replating experiment for SI organoids

For replating experiments (passage 1), single cells were isolated from WT (C57BL/6) mouse primary SI organoids grown under either ENR-DMSO, NR-DMSO or NR-GW3965 (1 µM) for a period of 5 days, as described before. In brief, primary organoids from respective culture conditions were digested with TrypLE Express (Gibco) with 1,200 U ml⁻¹ of DNase I (Roche) at 37 °C for 10 min. Cells were then washed; dead cells were removed using the EasySep dead cell removal (Annexin V) kit (STEM Cells), filtered using 40 µm filters and live cells were counted using the Guava Muse cell counter (Cytek) and the Muse Count and Viability Kit (Cytek). Equal numbers (10,000) of live IECs per condition in a 20 µl Matrigel (70% Matrigel) dome were plated in prewarmed 48-well plate and cultured for 6–7 days and were longitudinally imaged using the Incucyte S3 Live-Cell Analysis Instrument (Sartorius). Cells from ENR-DMSO, NR-DMSO or NR-GW3965 (1 µM) were cultured in respective medium with WNT-surrogate Fc (1 nM) (IpA Therapeutics)

and Y-27632 (10 µM, Sigma-Aldrich) added only during the first 3 days of the culture, with medium change done every 2 days afterwards. Organoid numbers on day 4 counted using the Incucyte Organoid Analysis Software Module were used to estimate the organoid plating efficiency (that is, the number of organoids/10,000 × 100). Buds per organoid and the percentage of budding organoids were estimated from day 6/7 images which were counted manually using Fiji/ImageJ (NIH).

### Colon organoids and single-cell replating experiment
After crypt isolation from the whole colon of WT (C57BL/6) mice, crypts were resuspended in 70% Matrigel (BD Bioscience) and 30% basal medium (advanced DMEM/F12, 1× penicillin–streptomycin, 1× GlutaMAX (Thermo Fisher Scientific), 10 mM HEPES (Thermo Fisher Scientific), 1× B27 (Life Technologies), 1 mM $N$-acetylcysteine (Sigma-Aldrich)). To establish primary colon organoids, approximately 500–600 crypts embedded in 20 µl of Matrigel (70%):crypt suspension (30%) and were seeded onto each well of a 48-well plate. Once solidified, the Matrigel was incubated in 200 µl WENR stem cell medium for 3 days supplemented with 10 µM Y-27632 (R&D Systems). The stem cell culture medium (WENR) contained basal medium supplemented with 50 ng ml⁻¹ of recombinant murine EGF (R&D), 500 ng ml⁻¹ recombinant murine R-spondin (R&D), 100 ng ml⁻¹ recombinant murine noggin (Peprotech), 500 nM A83-01 (Sigma-Aldrich), 10 mM nicotinamide (Sigma-Aldrich) and 1 nM WNT surrogate Fc (IpA Therapeutics). On day 3, WENR stem cell medium was replaced with ENR differentiating medium lacking WNT surrogate Fc, nicotinamide and Y-27632, with subsequent medium change every 2 days. All throughout the primary organoid culture, DMSO or GW3965 (1 µM, Sigma-Aldrich) were added to the culture medium.

For replating experiments (passage 1), single cells were isolated from the primary colon organoids grown under either DMSO or GW3965 (1 µM) for a period of 6–7 days. In brief, primary colon organoids from the respective culture conditions were digested with TrypLE Express (Gibco) with 1,200 U ml⁻¹ of DNase I (Roche) at 37 °C for 10 min. Cells were then washed; dead cells were removed using the EasySep dead cell removal (annexin V) kit (StemCell), filtered using 40 µm filters and live cells were counted using Guava Muse cell counter (Cytek) and the Muse Count and Viability Kit (Cytek). Equal numbers (10,000) of live colonocytes per condition in a 20 µl Matrigel (70% Matrigel) dome were plated in a prewarmed 48-well plate and cultured for 6–7 days and were longitudinally imaged using the Incucyte S3 Live-Cell Analysis Instrument (Sartorius). Cells from DMSO-treated or GW3965-treated (1 µM) conditions were cultured in respective WENR stem cell medium for first 3 days, followed by differentiating ENR medium with subsequent medium change every 2 days. Organoid numbers on day 4 counted using the Incucyte Organoid Analysis Software Module were used to estimate the organoid plating efficiency (that is, the number of organoids/10,000 × 100). Furthermore, owing to non-budding nature of the colon organoids, the total organoid area was estimated using the Incucyte organoid module. Colon organoids were collected at day 6/7 after single-cell seeding for gene expression analysis using qPCR.

### Image analysis protocol using the Incucyte Organoid Analysis Software Module
A set of representative images was chosen to ensure comprehensive coverage of organoid structures. The background to cell ratio was adjusted to accurately define the boundaries of organoid objects. Subsequently, the bright-field mask was evaluated, and filter parameters were refined accordingly. The refined bright-field mask was then applied to the image set, and a verification was conducted to ensure effective masking across all representative images. The edge split parameter was adjusted to delineate between individual organoid objects and refine segmentation. The bright-field mask was re-evaluated, and filter parameters were refined to ensure optimal delineation. After confirming that the parameters appropriately masked all organoids in the representative images, the Launch Wizard analysis was completed by selecting the scan times and wells to be analysed. This final step ensured a systematic and thorough analysis of organoid structures in the selected images.

### Salivary gland organoids
Salivary gland organoids were cultured according to the protocol published before[34]. Salivary glands from WT (C57BL/6J) mice were collected in 2,880 µl of RPMI (Sigma-Aldrich) supplemented with 2 mM L-glutamine, 1× penicillin–streptomycin (Thermo Fisher Scientific), 5% FCS and 0.1 mg ml⁻¹ DNase I (Roche). The salivary glands were minced with scissors into small pieces followed by addition of 120 µl of 25 mg ml⁻¹ of collagenase D (Sigma-Aldrich) to the 2,880 µl of RPMI. Tissues were incubated at 37 °C with gentle agitation for 40 min. Digestion was stopped by adding 30 µl of 0.5 M EDTA (Invitrogen). Cell suspensions were filtered using 100 µm cell strainers (Corning), centrifuged at 300$g$ for 8 min at 4 °C. The cell pellet was further digested with TrypLE Express (Gibco) with 1,200 U ml⁻¹ of DNase I (Roche) at 37 °C for 10 min. Cells were then washed, resuspended in advanced DMEM/F12 (Gibco) with 1× HEPES (Thermo Fisher Scientific) and then plated for organoids. Cells in basic culture medium were mixed with Matrigel (Corning) at a ratio of 30 (medium):70 (Matrigel) and 20 µl of mixture was plated carefully on to the centre of prewarmed flat bottom 48-well cell culture plate. The plate was placed at 37 °C and allowed to solidify for 15 min before 200 µl of organoid medium was added to each well. The basic culture medium contained advanced DMEM/F12, 1× penicillin–streptomycin, 1× GlutaMAX (Thermo Fisher Scientific), 10 mM HEPES (Thermo Fisher Scientific), 1× B27 (Life Technologies), 1× N2 (Life Technologies) and 1 mM $N$-acetylcysteine (Sigma-Aldrich), and was supplemented with 0.2 µg ml⁻¹ Primocin (InVivoGen), 100 ng ml⁻¹ of murine noggin (Peprotech), 100 ng ml⁻¹ recombinant murine R-spondin (R&D), 5 nM FGF1 (Peprotech), 1 nM FGF7 (Peprotech), 5 nM NRG1 (R&D) and 0.5 µM A83-01 (Merck). The medium was changed every 3 days and the cultures were maintained at 37 °C in fully humidified chambers containing 5% $CO_2$. For first 7 days of culture, 10 µM Y-27632 (Sigma-Aldrich) was added in the medium. On day 7, the salivary glands were washed to clean the Matrigel and replated without breaking the organoids. On day 7, while replating, DMSO or GW3965 (1 µM, Sigma-Aldrich) was added to the organoid medium without further addition of Y-27632. Then, 3 days after (day 10), when the medium was replenished, DAPT (5 µM, Tocris) was added to the culture medium. Organoids were imaged using the Nikon SMZ25 microscope with the NIS Elements software and collected for gene expression analysis between day 11 and day 13. For quantification of organoid area, $z$ stacks of whole-well images were converted to a focused EDF document and, using Affinity Photo 2, the contrast and gamma ratio were adjusted for each image, cropped in oval mode to remove edges but take all organoids. Segmentation analysis was performed using OrganoSeg software[53] according to the instructions and the organoid area was measured for each organoid and averaged per well from four different experiments.

### Single-cell sorting for organoid co-culture
To obtain single cells, isolated SI crypts were digested with TrypLE Express (Gibco) with 1,200 U ml⁻¹ of DNase I (Roche) at 32 °C for 90 s. Cells were then washed and stained with the following antibodies: CD31-PE (BioLegend, Mec13.3), CD45-PE (eBioscience, 30-F11), Ter119-PE (BioLegend, Ter119), EPCAM-APC (eBioscience, G8.8) and CD24-Pacific Blue (BioLegend, M1/69) at 1:500 dilutions by incubating for 30 min on ice. After staining, cells were washed in SMEM (Sigma-Aldrich), filtered using 70 µm filter and resuspended in SMEM supplemented with 7-AAD (BD Bioscience, 1:500). Cells were sorted using the FACS ARIA II (BD Biosciences) system. Sorting strategies were as follows: ISCs, Lgr5eGFP^high EPCAM⁺CD24^med/−CD31⁻Ter119⁻CD45⁻7-AAD⁻; Paneth cells, CD24^high sidescatter^high Lgr5eGFP⁻EPCAM⁺CD31⁻Ter119⁻CD45⁻7-AAD⁻. For organoid co-culture, ISCs were sorted from *Lgr5-eGFP-creERT2* mice and PCs were sorted from non-Lgr5 WT mice. Equal numbers (~5,000

each) of ISCs and PCs were co-cultured using standard ENR medium further supplemented with additional 500 ng ml$^{-1}$ R-spondin (to have a final concentration of 1,000 ng ml$^{-1}$), 100 ng ml$^{-1}$ WNT3A (R&D Systems) and 10 μM jagged-1 peptide (Anaspec) for first 4 days together with 10 μM Y-27632. After 4 days, the medium was changed to regular ENR medium. For in vitro stimulation, LXR agonist (1 μM GW3965) was added in the medium for the entire duration of the organoid culture. DMSO was used as vehicle. Crypt domain budding was quantified on day 8 after starting the co-culture.

## Cell isolation and cell sorting from mouse SI

Cell isolation from the mouse intestine performed using previously published protocol[5] with a few modifications. In brief, dissected SIs were flushed with cold PBS to remove the intestinal content and mucus, then cut open longitudinally and cut into ~1 cm pieces. To isolate the intestinal epithelial cells (IECs) and the intraepithelial lymphocytes (IELs), intestinal pieces were incubated in HBSS (HyClone) supplemented with 5% FCS, 15 mM HEPES (Thermo Fisher Scientific), 5 mM EDTA (Life technologies) and 1 mM DTT for 30 min at 37 °C under agitation at 800 rpm. The supernatant (called IEL fraction) containing IECs and intraepithelial lymphocytes was filtered through 100 μm cell strainers, centrifuged at 500$g$ for 5 min at 4 °C and was enriched using a 44%/67% Percoll gradient and washed for FACS analysis. To further isolate the cells from the LP, the intestinal pieces were washed with PBS supplemented with 5% FCS and 1 mM EDTA for 15 min at 37 °C under agitation at 800 rpm. Next, the tissue pieces were digested in HBSS supplemented with 0.15 mg ml$^{-1}$ liberase TL (Roche) or 0.5 mg ml$^{-1}$ collagenase D (Sigma-Aldrich) and 0.1 mg ml$^{-1}$ DNase I (Roche) at 37 °C for 45 min under agitation at 800 rpm. The digested tissues were filtered through 100 μm cell strainers and were washed for FACS analysis (called LP fraction). Single-cell suspensions were blocked with Fc-blocking solution (1:1,000, eBioscience) and stained with the antibody mix at 4 °C for 15 min. To examine immune cell subsets, acquired single SI cells were stained with an antibody cocktail of EPCAM-FITC (BioLegend, G8.8), CD19-FITC (BioLegend, 6D5), CD45-BV650 (BioLegend, 104), CD90-APC-Cy7 (BioLegend, 30-H12), CD64-BV421 (BioLegend, X54-5/7.1), CD11b-BV786 (Invitrogen, M1/70), CD11c-PE-Cy7 (BioLegend, N418), MHC-II (BD-Biosciences, 1A8), Ly6G-PE (BD-Biosciences, 1A8), Ly6C-PcP5.5 (Invitrogen, HK1.4), CD3-AF700 (BD-Biosciences, 500A1), CD103-APC (Invitrogen, 2E7) at 1:200 dilution for 15 min followed by flow cytometry analysis. To sort SI cells, the following antibodies were used for the staining: EPCAM-FITC (BioLegend, G8.8, 1:200), CD45-PeCy7 (BioLegend, 104, 1:200), Ter119-Pacific Blue (PB) (BioLegend, Ter119, 1:200). Cells were sorted using the SONY SH800S cell sorter. Sorting strategies were as follows: epithelial cells (IECs), EPCAM$^+$ CD45$^-$Ter119$^-$DAPI$^-$; intraepithelial immune, CD45$^+$EPCAM$^-$Ter119$^-$DAPI$^-$ from the IEL fraction; LP immune, CD45$^+$EPCAM$^-$Ter119$^-$DAPI$^-$; DN, CD45$^-$EPCAM$^-$Ter119$^-$DAPI$^-$ from LP fraction. Sorted cells were collected in RPMI (Sigma-Aldrich) medium supplemented with 5% FCS, centrifuged and resuspended in Trizol for RNA extraction. Flow cytometry data and dot plots were analysed and prepared using FlowJo (v.10).

## Sample preparation for scRNA-seq in SI tissue and organoids

For scRNA-seq from the SI, WT C57BL/6J mice were fed with either a standard ($n = 3$) or a GW3965 diet ($n = 3$) for 10 days. Next, cell suspensions were prepared and the respective fractions (that is, IEL and LP) were pooled before sorting IECs, immune and DN cells as described above. For scRNA-seq analysis of SI organoids, organoids were established using SI crypts from WT C57BL/6J mice ($n = 2$) and grown with NR culture medium and were treated with either DMSO or 1 μM GW3965 for 5 days with medium change every 48 h. On day 5, organoids were removed from Matrigel, cleaned with cold PBS and digested with TrypLE Express (Gibco) with 1,200 U ml$^{-1}$ of DNase I (Roche) at 32 °C for 5 min. After digestion, cells were washed and filtered using 70 μm filter. Single-cell suspensions were blocked with Fc-blocking solution

(eBioscience, 1:1,000) and stained with the antibody mix (1:200) at 4 °C for 15 min. The following antibodies were used for the staining: EPCAM-FITC (BioLegend, G8.8), CD45-PeCy7 (BioLegend, 104), Ter119-Pacific Blue (BioLegend, Ter119). EPCAM$^+$CD45$^-$Ter119$^-$DAPI$^-$ epithelial cells (IECs) were sorted using the SONY SH800S cell sorter. Approximately 30,000 cells from each condition were loaded onto the 10x chip and the samples were processed using the Chromium Next GEM single cell 3′ reagents kit v3.1 (dual index).

## scRNA-seq analysis

Libraries prepared for scRNA-seq were sequenced by Novogene at a sequencing depth of 300 million reads per sample using paired-end 150 bp reads on the NovaSeq 6000 sequencer and using the v1.5 reagent kit. Raw sequences were quantified and annotated using CellRanger (10x Genomics) v.6.0.2 (organoids) or v.6.1.2 (SI). Annotated gene counts were processed with Seurat[54] v.3.6.3 (organoids) or v.4.1.1 (SI tissue). Cells were filtered to include only cells with >500 (organoids) or 400 (SI) unique genes expressed, >5% ribosomal RNA and <10% mitochondrial RNA. SI samples were also filtered for cells expressing <25% Hb genes. Only genes expressed at >3 counts across all cells were included in the dataset. Mitochondrial genes were also excluded from further analyses. Doublet cells were excluded using DoubletFinder[55].

For the organoid dataset, integration of the two samples was performed using the first 30 dimensions of CCA in Seurat. Clustering was performed using FindNeighbors (dims = 1:30, k.param = 200, prune.SNN = 1/15) and FindClusters (original Louvain algorithm at resolution 0.5). Differential expression was analysed using FindAllMarkers (Wilcoxon rank-sum test, log[fold change] > 0.2, $P$ < 0.01) for cluster markers (subsampling 50 cells per cluster, difference in percentage > 0.2) and diet groups within clusters (subsampling 150 cells).

For the SI dataset, a primary UMAP was performed using all samples (based on the first 20 components of a principal component analysis (PCA) based on the 2,000 most highly variable genes). The two epithelial samples were integrated as for the organoid dataset and clustering was performed (FindNeighbors parameters: dims = 1:30, k.param = 200, prune.SNN = 1/15). The four immune and double negative samples were integrated as two samples, considering the two samples from the same animals as one, and clustering was performed as for the epithelial samples. Clusters were annotated manually using differential gene expression (FindAllMarkers as above). Differential expression analysis was performed between the standard-food and GW3965 samples for each sorting fraction separately using FindMarkers (Wilcoxon rank-sum test, subsampling 500 cells per group). The expression of inflammatory response genes was calculated using the Seurat function AddModuleScore using the genes included in GO term inflammatory response (GO: 0006954).

## Tissue processing and spatial transcriptomics of SI and colon

SI tissues from standard- and GW3965-diet fed mice at 0 and 3 d.p.i. were used for spatial transcriptomics. Colonic tissues from standard- and GW3965-diet fed mice at day 22 and day 43 of AOM–DSS treatment and untreated mice fed with standard diet only (day 0) were processed for spatial transcriptomics. In brief, SI and colon were cleaned from adipose tissue and cut longitudinally to remove the luminal content by washing in ice-cold PBS. Starting from the most distal portion, that is, ileum for SI and rectum for colon, the tissues were rolled, resulting in a Swiss roll with the most distal part in the centre and the proximal part in the outer portion of the roll. The Swiss rolls were snap-frozen for 1 min in a liquid nitrogen-cooled isopentane bath within a plastic tissue cassette. The frozen tissues were then embedded in optimal cutting temperature compound (OCT, Sakura Tissue-TEK) on dry ice for slow cooling and stored at −80 °C. Tissue sections (thickness, 10 μm) were cut using a pre-cooled cryostat and the tissue sections were transferred onto the oligo-barcoded capture areas (6.5 mm$^2$) on the 10x Visium Genomics slide and stored at −80 °C until further processing. The Visium spatial

transcriptomics protocol was performed according to previously published studies[19] and the manufacturer's instructions (10x Genomics, Visium Spatial Transcriptomic). H&E stained images were captured using a Leica DM5500 B microscope (Leica Microsystems) at 5X magnification. The Leica Application Suite X (LAS X) was used to acquire tile scans of the entire array and merge images. Sequence libraries were then processed according to the manufacturer's instructions (10x Genomics, Visium Spatial Transcriptomic). After the second cDNA strand synthesis, cDNA was quantified using the RT-qPCR ABI 7500 Fast RealTime PCR System and analysed using ABI 7500 v.2.3.

## Visium sequencing and data processing

Visium libraries were sequenced on the NovaSeq S1 flow cell (Illumina), at a depth of around 200 million reads per sample, with 28 bases from read 1 and 120 bases from read 2. FASTQ files were processed with SpaceRanger (10x Genomics) v.1.1.0 (colon) or v.1.2.0 (SI) mapped to the pre-built mm10 reference genome (GRCm38).

## Spatial transcriptomic analyses of SI tissues

The SI datasets was analysed using Seurat v.4.1.1 in R v.4.0.5 as follows. SpaceRanger output was imported and spots with fewer than 20 unique expressed genes were removed, as were genes expressed in fewer than 5 spots. PCA was performed using the 4,000 most highly variable genes. Harmony[56] was used to integrate the four samples. Graph construction was performed with the RcppHNSW[57] package using the first 50 dimensions in harmony, $k = 20$ and cosine distance, followed by Louvain clustering using the igraph[58] package. Differential expression between clusters was computed using Seurat function FindAllMarkers, with genes considered to be significant at FDR < 0.05 and $\log_2[FC] > 0.25$. Differential expression between standard diet samples at 0 d.p.i. and 3 d.p.i. was calculated using the Seurat function FindMarkers. Changes in *Areg* expression between conditions were analysed by identifying *Areg*+ spots (read counts > 0) and performing differential expression between *Areg*+ spots in standard-diet and GW3965 samples at 3 d.p.i., rerunning the analysis for each cluster separately. Non-negative matrix factorization was performed using the cNMF package in Python (v.3.9)[59].

## Spatial quantification of single cells during CRC progression

Visium sequencing data were processed using SpaceRanger software v.1.1. Six colonic sections were used for analysis, spanning from healthy control (that is, 43 days of standard diet without any AOM–DSS treatment, defined as day 0), AOM–DSS-induced CRC on standard diet at two timepoints (day 22 and day 43) and AOM–DSS-induced CRC on a diet containing GW3965 at two timepoints (GW day 22 and GW day 43). We also used a public single-cell dataset (GSE148794)[49] containing the major immune, stromal and epithelial cells during the time course of DSS colitis, both at steady state and during inflammation. Single-cell and spatial transcriptomics data were analysed using Seurat[54]. Genes related to mitochondrial and Malat1 as well as those detected in at least 5 cells were omitted from the downstream analysis. The top 4,000 highly variable genes were used for denoising using PCA (50 PCs), which in turn was used for visualization using UMAP (https://arxiv.org/abs/1802.03426), $k$-nearest neighbour graph construction on cosine distance and finally clustering using Louvain (resolution = 1). Cell clusters were annotated based on differentially expressed markers genes[49]. Projection of the aforementioned single-cell clusters onto the CRC Visium dataset was done using weighted non-negative least squares implemented in the SCDC package[60], a robust and fast method for cell type deconvolution[61]. Cell type abundances were compared pair-wise between conditions and their respective control (either day 0 or day 22/day 43) using Wilcoxon tests. Alterations in cell–cell co-detection in spatial transcriptomics spots was quantified and compared between GW3965 and standard diets, using the approach previously described[62]. Significant alterations were projected back to the single-cell UMAP embeddings for illustration.

## Bulk RNA-seq analysis of AOM–DSS tumour kinetics

cDNA libraries were constructed and sequenced by Novogene at a sequencing depth of 20 million reads per sample using paired-end 150 bp reads on the NovaSeq 6000 sequencer and the v1.0 reagent kit. Raw sequences were quantified and annotated using Kallisto[63] and GRCm38 (mm10) cDNA assembly[64]. Resulting gene counts were used for the following analyses, which were performed in R v.3.6.3 (R Core Team, 2020, https://www.R-project.org/). Owing to important differences in RNA degradation and the ribosomal RNA percentage between samples, analyses included corrections for the percentage of ribosomal RNA as described below. DEGs were identified with edgeR[65], according to the linear model $y \sim$ diet × time × percentage of ribosomal RNA. Genes were included in the module analysis if FDR < 0.05 in a likelihood ratio test including the diet and diet:interaction terms of the model ($H_0$: coefficients for all included terms equal 0). For all DEGs, log[FC] values were calculated for each group by combining log[FC] values from the likelihood ratio test for each relevant term (for example, $\log[FC]_{d70\text{-}GW3965} = 0 + \log[FC]_{d70} + \log[FC]_{GW3965} + \log[FC]_{d70:GW39765}$). These log[FC] estimates were scaled by the mean and s.d., and scaled log[FC] values of DEGs were subsequently clustered into modules using hierarchical clustering (Euclidean distance, Ward's method, tree cut at height 20). Over-represented KEGG pathways[66] and GO biological process terms[22] were identified using Enrichr (FDR < 0.05). To evaluate intragroup variability, module expression for each sample was calculated as the mean expression of all scaled log-transformed normalized counts (given by the edgeR cpm function, corrected for the percentage of ribosomal RNA using the limma function removeBatchEffect after log-transformation).

## RNA in situ hybridization (RNA scope)

RNA in situ hybridization was used to detect the expression of *Lgr5*, *Abca1* and *Areg* (Advanced Cell Diagnostics), in intestinal tissues at steady state and after irradiation. Formalin-fixed paraffin-embedded SI Swiss rolls were sectioned at 5 µm depth and were captured in MiliQ water using Superfrost gold slides. The RNAscope Multiplex Fluorescent v2 Assay (Advanced Cell Diagnostics) was used according to the manufacturer's protocol. In brief, the paraffin sections were baked for 1 h at 60 °C in the HybEZ Oven, then deparaffinized in xylene and washed in 100% ethanol. RNAscope hydrogen peroxide was used to block endogenous peroxidase activity by incubating slides at room temperature for 10 min and RNAscope 1× target retrieval was performed at 100 °C for 15 min. The slides were then incubated with RNAscope Protease Plus for 30 min at 40 °C in the HybEZ Oven. *Lgr5* (C2) and *Areg* (C3) probes were diluted in *Abca1* (C1) probe at a 1:50 ratio, the mix was added to the slides and baked for 2 h at 40 °C in the HybEZ Oven. The hybridization was done for AMP1, AMP2 and AMP3 according to the manufacturer's protocol. The *Abca1* signal was developed using TSA Vivid Fluorophore kit 570, *Lgr5* using Fluorophore kit 520 and *Areg* using Fluorophore kit 650 (Tocris). Finally, the counterstaining with DAPI was performed for 30 s at room temperature and the slides were mounted using ProLong Gold Antifade mounting medium (ProLong Antifade reagent; Invitrogen) and scanned with the Zeiss LSM 880 with Airyscan confocal laser-scanning microscope (Carl Zeiss) using a ×20 air objective. Images were processed using Fiji/ImageJ (NIH).

## Immunohistochemistry on paraffin embedded murine tissues

Tissues were fixed in 10% buffered formalin, paraffin embedded and sectioned (5 µm). For immunohistochemical analysis, the sections were deparaffinized in xylene and then rehydrated with graded alcohols. Endogenous peroxidase was blocked using 3% $H_2O_2$ (Scharlau) in methanol for 1 h. Antigen retrieval was performed using either 1 mM EDTA buffer (pH 8.0) or 10 mM Na-citrate buffer (pH 6.0) at 121 °C for 20 min using 2100 Antigen Retriever. The slides were then washed with PBS and blocked using BLOXALL solution (BLOXALL Blocking solution;

Vector Labs) for 2 h at room temperature. The sections were incubated overnight at 4 °C with primary antibodies diluted in PBS. Antibodies used were as follows: rat anti-BrdU 1:300 (Abcam), rabbit anti-cleaved caspase 3 1:200 (Cell Signaling Technology), rabbit anti-CYP27A1 1:200 (Abcam), mouse anti-AREG 1:500 (Santa Cruz) and rat anti-B220 1:200 (BioLegend). The slides were washed four times with PBS and incubated with biotinylated goat anti-rabbit (Vector Labs, 1:300), biotinylated goat anti-rat (Vector Labs, 1:300) or biotinylated goat anti-mouse (Vector Labs, 1:300) for 1 h at room temperature. The slides were washed four times in PBS and were incubated with the VECTASTAIN Elite ABC HRP Kit (Vector Labs) for 30 min. Staining was developed using DAB staining (DAB peroxidase staining kit, Vector Labs). The sections were counterstained with haematoxylin (Harris hematoxylin, Leica) and washed in tap water, dehydrated in increasing grades of alcohol and then with xylene and dried. The slides were mounted using permanent mounting medium (VectaMount, Vector labs). The slides were scanned using the Hamamatsu NanoZoomer Slide Scanner. Sectioning, H&E staining and DSS colitis and AOM−DSS tumour grading was performed in a blinded manner by the FENO (Morphological Phenotype Analysis) facility (Karolinska Institutet).

### Quantification of immunohistochemistry staining

Quantification of cCASP3+ and BrdU+ cells in the SI crypts was performed in a blinded manner using QuPath software. To isolate and quantify the distal SI, the first 10 cm of the proximal SI (starting from the pylorus) was removed and the rest of the distal SI was Swiss-rolled. Crypts (approximately 100 crypts per section) were delineated manually using the polygon or brush tool in QuPath. Crypts were selected if sectioned whole length (that is, not transversally cut). The numbers of DAB+ cells per crypt were quantified automatically using the 'positive cell detection' command in QuPath. The percentage of surviving crypts was calculated as the percentage of crypts with more than or equal to 10 BrdU+ cells out of all of the crypts quantified. The average number of cCASP3+ or BrdU+ cells per crypt was calculated as the average number of DAB+ cells in all of the crypts analysed in each section. The normalized percentage of surviving crypts (where indicated), was calculated by dividing the percentage of surviving crypts in each experiment by the average of the percentage of surviving crypts in the control group of each experiment.

For quantification of LP CYP27A1+ cells in the crypt niche, an average of 48 intact crypts within Swiss rolls stained for CYP27A1+ at different d.p.i. were first identified using NDP.View 2 (NanoZoomer Digital Pathology) software. DAB+ cells in the epithelial crypt base, transit-amplifying zone and LP surrounding each crypt were manually counted.

Quantification of total CYP27A1 staining in SI and colon Swiss-rolls was quantified using ImageJ. In brief, scanned sections were converted to black and white using the 'make binary' command in ImageJ and the total mean grey value of the stained area was calculated. In the colour threshold tab, hue, saturation and brightness were adjusted to highlight only the DAB+ signal. Using the make binary command, the mean grey value of the DAB intensity was then calculated. The CYP27A1 intensity was calculated as the ratio of the mean grey value of the DAB intensity by the mean grey value of the total stained area (that is, area of the total Swiss-roll). The normalized CYP27A1 intensity was calculated by dividing the intensity of each timepoint (for example, post-irradiation) by the average of the intensity of the untreated control (for example, day 0).

AREG quantification in the crypt and villus LP of the distal SI was performed in a blinded manner. For crypt AREG quantification, around 10 high-powered field (×40) images per mouse were exported and analysed using QuPath. In each image, crypts (4–15 crypts per field) were delineated manually using the polygon or brush tool as described previously. Overall around 100–125 crypts per mouse were quantified. Using the automated quantification tool, mean DAB intensity and the crypt area were quantified for each crypt. Mean DAB intensity for each crypt was normalized to its area, which was then used to calculate the average DAB intensity (AREG staining) per mouse. For AREG staining in villus LP, 10X images of the complete distal SI was exported and analysed using QuPath. In brief, the crypt regions were selected out and the analysis was focused only on the villus region. Using the cell detection tool, the mean DAB intensity and the area of individual selected cells was quantified. Using this, we quantified around 300–1,500 cells in the villus region/mouse. The mean DAB intensity for each cell in the villus LP was normalized to its area, which was then used to calculate the average DAB intensity (AREG staining) per mouse.

Quantification of B220+ B cell follicles in AOM−DSS tumour samples was done blinded using QuPath software. In brief, the anti-B220 stained follicles irrespective of their sizes or staining intensity were selected and outlined using a polygon or brush tool in QuPath. The number of annotated B220+ follicles was counted and plotted.

### Immunofluorescence of mouse paraffin tissues

Tissue fixation, paraffin embedding, sectioning and staining protocols were performed as described above for immunohistochemistry with the following changes. For primary antibody detection, slides were incubated in PBS with rabbit anti-OLFM4 (1:300, Cell Signalling), rat anti-BrdU (1:300, Abcam) and wheat germ agglutinin (WGA) AF555 (1:5,000, Invitrogen) overnight at 4 °C in darkness. After primary antibody incubation, the slides were washed with PBS and stained with goat anti-rabbit AF488 (1:500, Invitrogen), donkey anti-rat AF647 (1:500, Jackson ImmunoResearch) for 90 min at room temperature in darkness. After secondary antibody staining, the slides were washed with PBS and nuclei were stained with DAPI (1 μg ml⁻¹) for 30 min and subsequently washed with PBS. Mounting was performed using Vibrance Antifade Mounting Medium (Vectashield). Fluorescent-stained slides were imaged using the LSM880 confocal microscope (Zeiss). Quantification of BrdU+Olfm4+ and Olfm4+ cells from 20–30 ileal crypts per mouse was performed using the Cell Counter plugin in Fiji/ImageJ (NIH). WGA staining was used to differentiate the crypt niche and the TA zone.

### Immunoblotting

SI organoids were grown in ENR or NR medium supplemented with DMSO, GW3965 (1 μM), with the medium changed every second day. On day 5, organoids were lysed in RIPA buffer with 1× protease inhibitor cocktail (Roche) and 1× PhosStop (Roche) phosphatase inhibitors. Cells were sonicated, centrifuged for 5 min at 10,000 rpm, and the cleared lysates were measured using the DC Protein Assay (Bio-Rad). The samples were run on 4–12% Bis-Tris protein gels (Thermo Fisher Scientific) with 20 ng μl⁻¹ protein loaded per well and the Precision Plus Protein All Blue Standards (Bio-Rad, 1610373EDU) as a molecular mass marker. Gels were blotted onto PVDF membranes, followed by blocking in 5% non-fat dry milk and incubated overnight at 4 °C with the following primary antibodies: rabbit polyclonal ERK1/2 (Cell Signaling Technology; 1:1,000), rabbit polyclonal phospho-ERK1/2 (Cell Signaling Technology; 1:1,000), and mouse monoclonal Vinculin (Sigma-Aldrich, V9131, 1:1,000). HRP-conjugated anti-rabbit (Sigma-Aldrich; 1:5,000) or anti-mouse (Cell Signaling Technology; 1:1,000) were used as secondary antibodies for 1 h at room temperature. Signal was detected using the Pierce SuperSignal West Pico Plus reagent (Thermo Fisher Scientific) and visualized on the BioRad luminescence detector. Densitometry was performed using the software Image Lab.

### RNA extraction and RT−qPCR

Tissue biopsies collected for RNA extraction were preserved in RNAlater (Invitrogen) at −20 °C. Tissues were transferred to RLT Plus buffer (Qiagen) + 2% β-mercaptoethanol and lysed by bead-beating (Precellys) and stored at −80 °C till RNA extraction. Intestinal crypts and organoid cell suspensions were incubated in RLT Plus buffer (Qiagen) + 2%

β-mercaptoethanol and stored at −80 °C before RNA extraction. RNA isolation was performed using the RNAeasy Mini Kit (Qiagen) according to the manufacturer instructions and was quantified using the Nanodrop. For RNA extraction from cells sorted directly into Trizol (Life technologies) or resuspended in Trizol after sorting, sorted cells were mixed well in Trizol and incubated for 5 min at room temperature and stored at −80 °C until further processing. For RNA extraction, samples were thawed at room temperature and an appropriate amount of chloroform (Fisher Scientific) was added and mixed by vortexing it well followed by 2–3 min incubation at room temperature and then centrifuging at 12,000$g$ for 15 min at 4 °C. The aqueous phase containing the RNA was carefully transferred to new Eppendorf tubes and 1 μl of Glycoblue (Thermo Fisher Scientific) was added to each of the samples, which were then mixed by flicking; an appropriate amount to isopropanol was then added to each sample and the samples were mixed well. The samples were then incubated for 1 h at −80 °C followed by centrifuging at 12,000$g$ for 20 min at 4 °C. The samples were then washed in 70% cold ethanol by centrifuging at 12,000$g$ for 15 min at 4 °C. Excess ethanol was removed and the samples were air dried until all of the residual ethanol was removed. The samples were resuspended in an appropriate amount of nuclease-free water (Life Technologies) and were quantified using the Nanodrop. Reverse transcription was performed using iScript RT Supermix (Bio-Rad). Gene expression was analysed by qPCR using the iTaq Universal SYBR Green Supermix (Bio-Rad) and log$_2$-transformed fold changes were calculated relative to either *Hprt* (when using tissues) or *B2m* (when using cells) or *Actb* (when indicated) mRNA using the $2^{-\Delta\Delta C_t}$ method[67]. For mouse *Cyp27a1* expression, primer pair 1 was used when expression was analysed in WT mice, and primer pair 2 was used when expression was analysed in *Cyp27a1*$^{-/-}$ or control littermates. Primers used for qPCR were as follows: *b2m*, ACCGTCTACTGGGATCGAGA, TGCTATTTCTTTCTGCGTGCAT; *Hprt*, TCAGTCAACGGGGGACATAAA, GGGGCTGTACTGCTTAACCAG; *Actb*, CACTGTCGAGTCGCGTCC, GTCATCCATGGCGAACTGGT; *Abca1*, TGGGCTCCTCCCTGTTTTTG, TCTGAGAAACACTGTCCTCCTTTT; *Abcg1*, AGGCAGACGAGAGATGGTCA, AAGAACATGACAGGCGGGTT; *Olfm4*, TGCTCCTGGAAGCTGTAGTCA, TGTATTCAAAGGTGCCACCCA; *Areg*, CAGTGCACCTTTGGAAACGA, ATGTCATTTCCGGTGTGGCT; *Ereg*, TGCTTTGTCTAGGTTCCCACC, CGGGGATCGTCTTCCATCTG; *Tgfa*, CAAACACTGTGAGTGGTGCC, GGGATCTTCAGACCACTGTCTC; *Egf*, AGGATCCTGACCCCGAACTT, ACAGCCGTGATTCTGAGTGG; *Egfr*, CGCC AACTGTACCTATGGATGT, GGGCCACCACCACTATGAAG; *Wnt3*, TGGA ACTGTACCACCATAGATGAC, ACACCAGCCGAGGCGATG; *Jag1*, GAGC CAAGGTGTGCGG, GCGGGACTGATACTCCTTGA; *Jag2*, GCCTCGTCGT CATTCCCTTT, AGCTCCTCATCTGGAGTGGT; *Dll1*, GCGACTGAG GTGTAAGATGGA, GCAGCATTCATCGGGGCTAT; *Hes1*, GAAAAATTC CTCCTCCCCGGT, GGCTTTGATGACTTTCTGTGCT; *Cyp27a1* (pair 1), CCCACTCTTGGAGCAAGTGA, CCATTGCTCTCCTTGTGCGA; *Cyp27a1* (pair 2), CCCAAGAATACACAGTTTG, GCCTCTTTCTTCCTCAGC; *Il33*, GGGCTCACTGCAGGAAAGTA, TGGGATCTTCTTATTTTGCAAGGC; *Clu*, ATTCTCCGGCATTCTCTGGG, CCTTGGAATCTGGAGTCCGGT; *Msln*, GCCTAGTAGACACTACTGCAGAC, AGCAGTAGGAAGCTTCGGC; *Yap1*, ATTTCGGCAGGCAATACGGA, CACTGCATTCGGAGTCCCTC; *Wnt9b*, CCAGAGAGGCTTTAAGGAGACG, GGGGAGTCGTCACAAGTACA.

For gene expression analysis in human biopsies, the following Taqman probes from Thermo Fisher Scientific were used: *HPRT1*: Hs02800695_m1_4453320; *CYP27A1*: Hs01017992_g1_4448892; *ABCA1*: Hs01059137_m1_4448892.

## Cohort of patients with ulcerative colitis

Paired endoscopic biopsy specimens were obtained from the ascending colon and sigma/rectum from patients with IBD or suspicion of intestinal disease. One of the paired biopsies was used for histological assessments and the other for RNA extraction. Human studies were approved by the local ethical committee (EthikKommission der Ärztekammer Hamburg PV4444).

## Human CRC tissue microarray

**Patients.** Samples from patients with CRC from the Coloproctology Department, Clínica Las Condes, were included between 2015 and 2017. All of the patients signed informed consent forms approved by the institution (Cómite de ética de la investigación de CLC, O22019AA) and the procedures were performed according to human experimental and clinical guidelines. Patients undergoing surgery for tumour resection had to be older than 18 years old and not have received chemotherapy or neoadjuvant therapy before total or partial colectomy. Tumour staging was classified according to the TNM classification (The Union for International Cancer Control; UICC). Immediately after surgery, samples of fresh tumour and healthy intestinal mucosa (at least 10 cm away from tumour) were macroscopically selected. Biopsy-size samples of tumour and healthy tissue were fixed in 2% paraformaldehyde and paraffin embedded for a tissue microarray (TMA) construction and immunohistochemistry analysis.

**TMA generation.** For TMA generation, TMAs were assembled from formalin-fixed paraffin-embedded tissues using a 0.6-mm-diameter punch (Beecher Instruments). The arrays encompass 14 tissue cores from colonic tumours and healthy tissue derived from 11 patients. Moreover, two cores from kidney tissue were used (for orientation purposes). Using a tape-transfer system (Instrumedics), 2 μm sections were transferred to glass slides and analysed using immunohistochemistry (IHC).

**IHC.** Conventional IHC on TMA sections was performed on 2 μm sections using antibodies against CYP27A1 (Abcam, dilution 1:200). Immunohistochemistry was performed using the R.T.U. VECTASTAIN Kit (Vector Labs), according to the manufacturer's instructions. Sections were deparaffinized and rehydrated with deionized water. The sections were then heated in EDTA buffer, pH 8.0, for 20 min, and cooled for 10 min before immunostaining. All of the samples were blocked by exposure to R.T.U. normal horse serum for 15 min, then incubated in the following sequential order: primary antibody for 1 h, 3% $H_2O_2$ (blockade of endogenous peroxidases) for 15 min, R.T.U biotinylated universal antibody anti/rabbit/mouse IgG for 30 min, R.T.U. ABC reagent, 3'3-diaminobenzidine as a chromogen for 5 min and finally counterstained with haematoxylin for 5 min. The above was carried out at room temperature and, between incubations, the sections were washed with Tris-buffered saline. Coverslips were placed using the Tissue-Tek SCA (Sakura Finetek). Images were captured using the Aperio ScanScope equipment, analysed by the Aperio ImageScope Software and evaluated using the Positive Pixel Count 9 algorithm. The proportion of positive pixels with respect to the total pixels (positive and negative) per area was evaluated in crypts and tumour region, excluding stroma (positivity/area).

**Indirect immunofluorescence.** Paraffin histological sections derived from a primary CRC tumour and healthy tissue were evaluated for the co-expression of CYP27A1 and Vimentin by immunofluorescence. In brief, the sections were subjected to deparaffinization (NeoClear, Merck), then rehydrated with a battery of alcohols from absolute ethanol to 70% ethanol. The antigen retrieval was performed with EDTA buffer (pH 8.0). The sections then were incubated with 100 mM glycine and 2% BSA (Sigma–Aldrich) + 1% normal donkey serum in 1× PBS (Sigma–Aldrich) (for autofluorescence and non-specific protein blocking, respectively). The sections were incubated at room temperature for 1 h with the following primary antibodies: rabbit anti-CYP27A1 (1:200, Abcam) in conjunction with mouse anti-vimentin (1:1,000, Abcam). After a rinse with PBS, the tissue sections were incubated for 1 h at room temperature with the following secondary antibodies (Invitrogen): goat anti-rabbit IgG conjugated with Alexa Fluor 488 (1:500) and goat anti-mouse IgG conjugated with Alexa Fluor

546 (1:200). Hoechst 33342 (1:500) was used as a nuclear counterstain. Finally, the slides were covered with a coverslip plus mounting solution (Dako, Agilent Technologies). The slides were visualized using the C2+ confocal microscope with the ×20 objective (Nikon Instruments).

**Bulk and single-cell RNA-seq reanalyses of published datasets**

**Re-analysis of deposited scRNA-seq datasets.** The following publicly available datasets were downloaded from the Gene Expression Omnibus (GEO): GSE117783 (ref. 4), scRNA-seq data from both normal and irradiated SI crypts. The standard Seurat v.3.1.3 protocol was used to analyse these datasets. Cells with a number of expressed genes of <200 and >10,000 were first filtered out. Genes expressed in at least one cell were retained. Preprocessing, normalization and scaling of the data were carried out using inbuilt Seurat functions. Later, a graph-based clustering approach was used to identify subpopulations of cells and also to optimize the clustering approach to get finalized clusters at resolution 0.5. A list of markers genes for these clusters was obtained and used to further characterize the subpopulations. Furthermore, for the GSE117783 dataset, we analysed the crypts and whole epithelial cells separately. In the crypts sub-dataset, we merged all of the irradiated and normal cells from different clusters together and carried out differential expression analysis using Wilcox tests between the irradiated (cells in bulk) versus normal (cells in bulk) condition. For all the graphs we used the inbuilt functions in Seurat v.3.1.3.

**Comparison of the bulk DSS kinetics dataset and scRNA-seq data from irradiated crypts.** Time-series bulk RNA-seq data GSE131032 (ref. 5) were downloaded from the GEO. In this dataset, published by our group, we determined in an unbiased manner which genes and pathways are differentially regulated during mouse colonic inflammation followed by tissue regeneration. From this publication, we obtained supplementary dataset 1, containing all the DEGs, which were compared with the upregulated genes in irradiated crypts cells in bulk using jvenn program[68]. Functional enrichment analysis using enrichR (v.2.1)[69] was performed on the genes shared between these two datasets (that is, DEGs in DSS kinetics and upregulated genes in irradiated crypts compared to control). Graphical plotting was performed using inbuilt functions in R.

**Analysis of the CRC microarray dataset GSE39582.** Analysis of the CRC microarray dataset GSE39582 (ref. 47) was done in R. Probes with a $\log_2$ signal below 5 in at least 3 samples were filtered out. Data were quantile normalized and processed for differential gene expression using limma package[70] by testing each CRC subgroup to the healthy control group. Genes with FDR < 0.01 and fold change > 1.5 were considered to be significant. Samples were stratified into six tumour subtypes (c1 to c6) based on clinicopathological and molecular differences used in the published study.

**Statistical analysis**

All statistical analysis, unless otherwise indicated, were performed using GraphPad Prism 9, v.9.5.0. No statistical methods were used to predetermine sample size. Details of statistical tests are provided in the figure legends.

**Reporting summary**

Further information on research design is available in the Nature Portfolio Reporting Summary linked to this article.

**Data availability**

RNA-seq data generated in this study have been deposited at the NCBI GEO under the following accession codes. Bulk RNA-seq data in AOM–DSS CRC (GSE180078), scRNA-seq in organoids (GSE180079),

spatial transcriptomics in AOM–DSS CRC (GSE227598), spatial transcriptomics in irradiation induced injury–repair in the SI (GSE227742) and scRNA-seq of SI cells after LXR activation (GSE227726). All of these datasets are now publicly available. All other publicly available datasets used herein are available at the GEO under accession codes GSE148794 (ref. 49), GSE117783 (ref. 4), GSE131032 (ref. 5) and GSE39582 (ref. 47). Source data are provided with this paper.

**Code availability**

All of the custom codes generated in this study have been deposited in a GitHub public repository (https://github.com/ejvillablancaLab/LXR_in_regeneration_and_tumorigenesis), which can be accessed at Zenodo[71] (https://doi.org/10.5281/zenodo.13133717).

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

**Acknowledgements** We thank the Villablanca laboratory members for discussions; N. Nedelsky for editorial assistance; S. Inampudi and M. Shaik for their help with experiments; the staff at the FENO facility for tissue processing. The computations/data handling was enabled by resources provided by the Swedish National Infrastructure for Computing (SNIC), under project number SNIC 2017/7-445 and partially funded by the Swedish Research Council through grant agreement no. 2018-05973. We also thank the staff at the National Bioinformatics Infrastructure Sweden (NBIS) long-term support. The analysis pipeline for salivary glands has been made possible in part by BioImage Informatics Facility, a unit of the National Bioinformatics Infrastructure Sweden NBIS, with funding from SciLifeLab, National Microscopy Infrastructure NMI (VR-RFI 2019-00217) and Chan Zuckerberg Initiative DAF (DAF2021-225261, https://doi.org/10.37921/644085ggkbos, an advised fund of Silicon Valley Community Foundation, https://doi.org/10.13039/100014989). The Novo Nordisk Foundation Centre for Stem Cell Medicine (reNEW) are supported by a Novo Nordisk Foundation grant number

NNF21CC0073729. This project has received funding from the Innovative Medicines Initiative 2 Joint Undertaking (JU) under grant agreement No 875510. The JU receives support from the European Union's Horizon 2020 research and innovation programme and EFPIA and Ontario Institute for Cancer Research, Royal Institution for the Advancement of Learning McGill University, Kungliga Tekniska Hoegskolan, Diamond Light Source Limited. This communication reflects the views of the authors and the JU is not liable for any use that may be made of the information contained herein. X.L. was supported by grants from China Scholarship Council (nr. 201907930012). S.D. was supported by Cancerfonden (CAN 2016/1206), Åke Weibergs Stiftelse (M21-0074, M22-0048, M23-0179); and E.J.V. by grants from the Swedish Research Council VR (K2015-68X-22765-01-6, 2018-02533 and 2021-01277), FORMAS (FR-2016/0005 and 2022-01066), Cancerfonden (19 0395 Pj and 22 2060 Pj), European Research Council (ERC) Synergy Grant 101118531, and Wallenberg Academy Fellow (WAF) program (2019.0315 and 2024.0135).

**Author contributions** S.D., S.M.P. and X.L. designed, performed and analysed most of the experiments. S.S., B.C.K., A.O., O.E.D., C.S., A.T.W., R.A.M, S.L., G.M., F.C., P.P., N.S. and M.D.l.F. performed experiments and analysed the data. J.F., K.P.T., P.C. and N.H. analysed bioinformatics data. S.N., H.L.L., P.A., S.H., M.B., J.-A.G. and P.K. provided mice, reagents, samples and feedback. F.L.-K. and M.A.H. provided human material. R.K. performed pathological evaluation of mouse DSS and tumour samples. S.D., S.M.P., X.L. and E.J.V. wrote the manuscript. S.D., S.M.P. and E.J.V. conceived the study. E.J.V. provided supervision and funding acquisition.

**Funding** Open access funding provided by Karolinska Institute.

**Competing interests** E.J.V. has received research grants from F. Hoffmann-La Roche and is a founder of PaperVids. S.D. works as a consultant for Cellphi Biotechnology. The other authors declare no competing interests.

**Additional information**
**Correspondence and requests for materials** should be addressed to Srustidhar Das or Eduardo J. Villablanca.

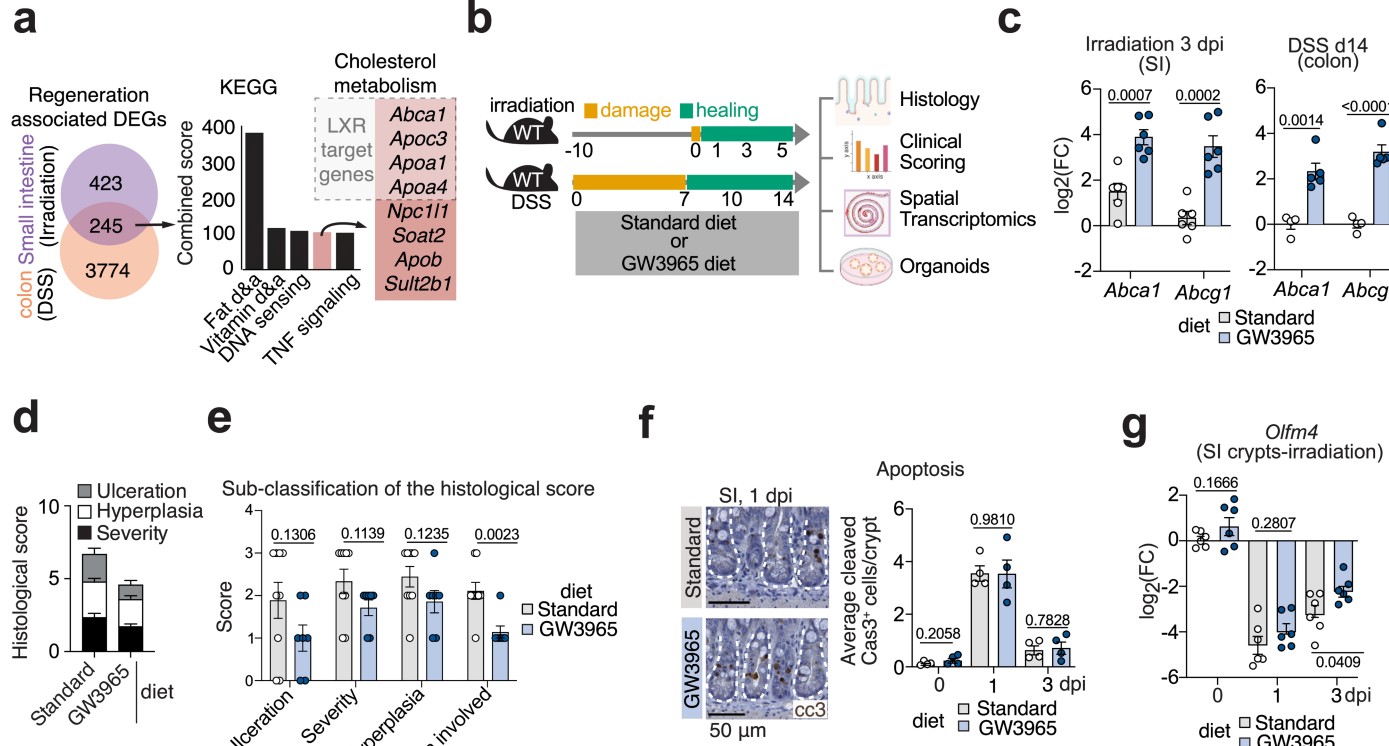

**Extended Data Fig. 1 | LXR activation promotes intestinal regeneration without affecting tissue damage. (a)** Venn diagram showing differentially expressed genes (DEG) overlap between regenerative and steady state colonic tissue and small intestine (SI) crypts following DSS-induced colitis and irradiation, respectively. Kyoto Encyclopedia of Genes and Genomes (KEGG) pathway analysis from shared differentially expressed genes (DEGs) between colon and SI. d&a: digestion and absorption. **(b)** Scheme of the experimental procedures shown in Extended Data Figs. 1–3 and Fig. 1. Briefly, WT mice were fed with standard (STD) or GW3965-containing diet for 10 days and then subjected to 10 Gy total body irradiation (TBI, i.e., damage) and allowed to heal (for 1, 3 or 5 days) while still on modified diet. Alternatively, mice were fed STD or GW3965 diet and simultaneously administered DSS in drinking water for 7 days (i.e., damage) followed by 2–7 days of regular water (i.e., healing). As a readout for mucosal healing the intestinal tissue was harvested for histology,

blind clinical scoring, spatial transcriptomics and/or organoid culture. **(c)** *Abca1* and *Abcg1* expression by qPCR from SI crypts of mice at 3 days post irradiation (3dpi) or from colonic tissue of mice at day 14 of DSS-induced colitis. Datasets for irradiation are normalized to crypts from 0dpi. **(d-e)** Cumulative **(d)** and stratified **(e)** histological score of colonic tissue at day14 after DSS treatment of mice fed STD or GW3965 diet. **(f)** Representative images (1dpi) and quantification of cleaved caspase 3$^+$ (cCASP3$^+$) cells/crypt in the distal SI at 0, 1 and 3 dpi. **(g)** *Olfm4* expression by qPCR in SI crypt at 0, 1 and 3 dpi from STD or GW3965 diet fed mice. In panels **c**, **e-g** each dot represents one mouse, and the data are representative of two **(d-f)** or three **(c** and **g)** independent experiments. Data are shown as bar plots with mean ± s.e.m. in **(c-g)**. Significance was assessed by unpaired two-sided *t* test between diet groups **(c**, **e-g)**. Part of the schematic in panel **b** was drawn using BioRender.com.

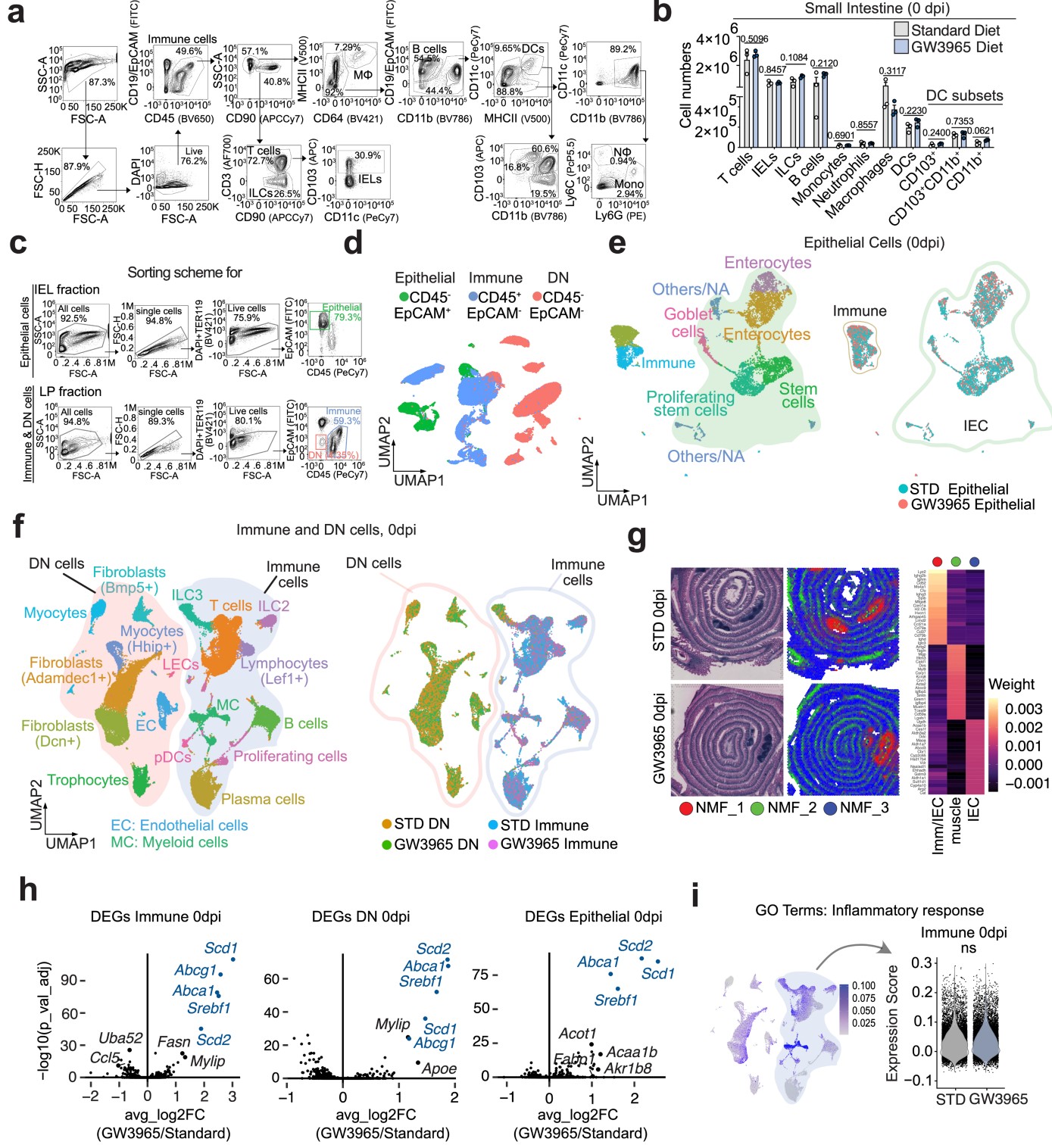

**Extended Data Fig. 2** | See next page for caption.

**Extended Data Fig. 2 | Immune, epithelial, and stromal cell profiling upon 10 days of standard or GW3965-diet.** (a-b) Flow cytometry dot plots showing the gating strategy (**a**) and quantification (**b**) of the number of major immune cell subsets in the SI tissue of mice exposed to 10 days of standard or GW3965 diet. Cell numbers are not significantly different between diet groups in any immune cell subsets analysed. Each dot represents one mouse and data are representative of one experiment. Data are shown as bar plots with mean ± s.e.m. Unpaired two-sided *t* test between diet groups. The diagram was adapted from ref. 19, CC-BY 4.0. (**c**) Gating strategy and sorting scheme for epithelial (EpCAM$^+$CD45$^-$), immune (EpCAM$^-$CD45$^+$) and stromal (i.e., DN, double negative, EpCAM$^-$CD45$^-$) cells processed for single cell RNA sequencing (scRNAseq). (**d**) UMAP distribution of scRNAseq datasets obtained from epithelial, immune and DN cells from SI of mice fed for 10 days with STD or GW3965 diet. (**e-f**) Left: unsupervised clustering and annotation of epithelial cells (**e**) and immune and stromal cells (**f**) in UMAP. Right: cells in the clusters shown to the left are colour-coded based on the diet treatment (**e**) and the combined cell compartment and diet (**f**). (**g**) Left: SI Swiss rolls shown in hematoxylin and eosin (H&E) staining and colour coded spatial transcriptomic (ST) map based on non-negative matrix factorization (NMF). Right: heatmap showing top 20 genes defining each NMF and respective annotations (i.e., NMF1: immune and intestinal epithelial cells (IEC), NMF2: muscle and NMF3: IEC). (**h**) Volcano plots showing the log2FC and adjusted p-values from differential expression analysis between SI scRNAseq samples from GW3965 diet- vs. STD-diet fed mice in immune, DN and epithelial cells. Genes known to be regulated by LXR are marked blue. (**i**) UMAP and violin plot showing expression of module scores based on genes belonging to the GO term "Inflammatory response" (GO:0006954) and their expression score in scRNAseq datasets of immune cells from STD and GW3965 samples. For scRNAseq experiment, epithelial, immune and DN cells were sorted from a pool of 3 mice. DEG were identified with Wilcoxon rank-sum test, subsampling 500 cells per group. For ST, one mouse per diet treatment was processed. MΦ, Macrophages; DC, Dendritic Cells; IEL, intra epithelial leucocytes; ILC, innate lymphoid cells; Mono, Monocytes; NΦ, Neutrophils.

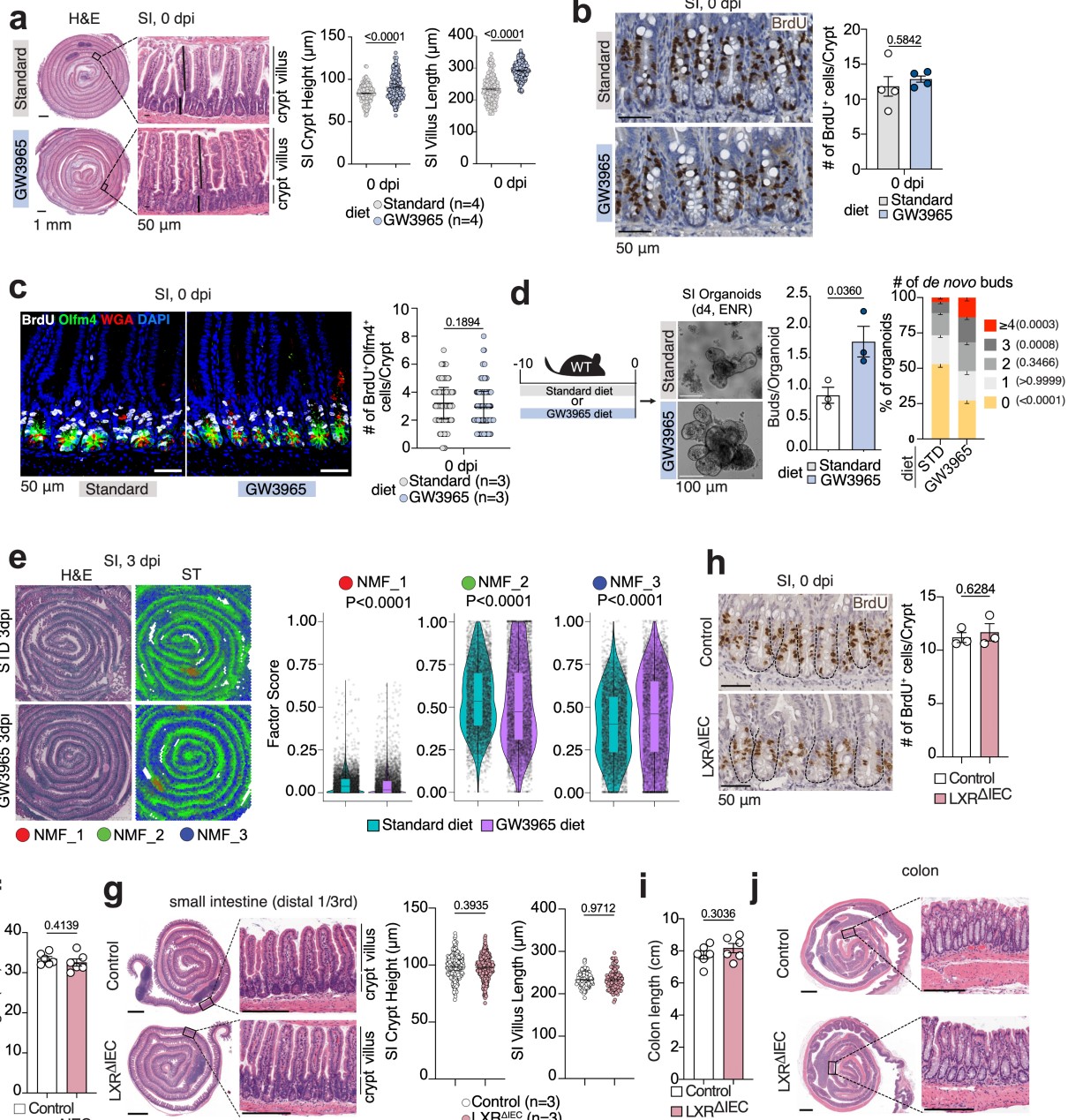

**Extended Data Fig. 3 | LXR activation in steady state primes the tissue for improved regeneration upon challenge.** (**a**) Left: representative Hematoxylin & Eosin (H&E) stained Swiss rolls and zoomed inset of distal small intestine (SI) tissue from mice fed with STD or GW3965 diet for 10 days. Right: quantification of SI crypt height and villus length. Each dot represents an individual crypt or villus (n = 4 mice). (**b**) Representative pictures and quantification of the number BrdU⁺ cells/crypt after 2 h pulse in the distal SI of mice fed for 10 days with STD or GW3965 diet. Each dot represents one mouse. (**c**) Left: representative pictures of immunofluorescence staining of SI tissue from STD or GW3965-diet fed mice after a 2 h BrdU pulse. Right: quantification of proliferating intestinal stem cells/crypt. Proliferating intestinal stem cells (ISCs) were defined as BrdU⁺Olfm4⁺ cells located underneath the uppermost wheat germ agglutinin (WGA)⁺ Paneth cells. Each dot represents one crypt and data are representative of n = 3 mice/diet group. (**d**) WT mice were fed with STD or GW3965-diet for 10 days and SI crypts were extracted and plated for organoid culture in ENR media in absence of additional stimuli *in vitro*. Representative pictures of organoids at day4 and quantification of average number of buds/organoid and % of organoids with the indicated number of buds are shown on the right. Each dot represents one mouse. The diagram was

adapted from ref. 19, CC-BY 4.0. (**e**) Left: SI Swiss rolls shown in H&E staining and colour coded spatial transcriptomic (ST) map based on non-negative matrix factorization (NMF). Right: Quantification showing relative factor score of each of the NMF obtained from the ST map. Samples are from WT mice fed with STD or GW3965-diet for 10 days and harvested at 3dpi after 10 Gy total body irradiation (TBI). (**f**) Quantification of SI length from Villin-Cre:LXRα^f/fβ^f/f (LXR^ΔIEC) mice and their littermate LXRαβ^flox/flox controls. Each dot represents one mouse. (**g**) Representative H&E pictures and quantification of distal SI crypt and villus length from LXR^ΔIEC and littermate controls. Each dot represents one crypt or villus from n = 3 mice/group. (**h**) Representative pictures and quantification of the number BrdU⁺ cells/crypt after 2 h pulse from the distal SI of LXR^ΔIEC and littermate controls. Each dot represents one mouse. (**i-j**) Representative H&E picture and quantification of colon length from LXR^ΔIEC and littermate controls. Each dot represents one mouse. Data are representative of one (**c**, **e**, **g**, **h** and **j**), two (**a**, **b**, **f** and **i**) or three (**d**) independent experiments. Data are shown as bar plots with mean ± s.e.m. in (**a-d**, **f-i**). Significance was assessed by unpaired two-sided *t* test in **a-c**, **d** (buds/organoid) and **f-i**, two-way ANOVA with Bonferroni test in **d** (*de novo* buds) and Mann-Whitney test in **e**.

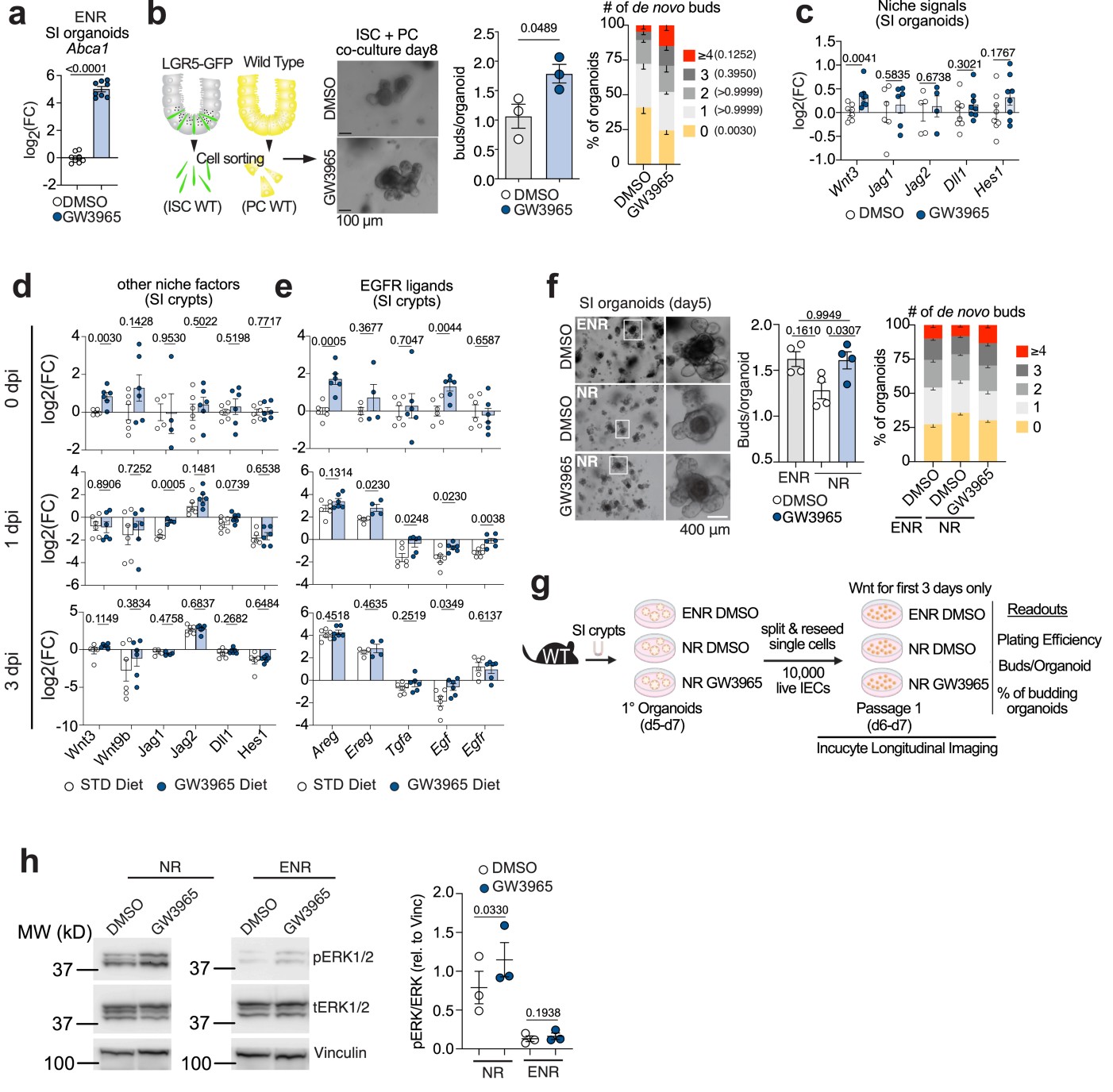

**Extended Data Fig. 4** | See next page for caption.

**Extended Data Fig. 4 | LXR activation upregulates EGFR ligands in IECs.**
(**a**) qPCR analysis of *Abca1* expression in organoids treated with either DMSO or GW3965 for 5 days in ENR culture medium. (**b**) Left: Scheme showing sorting of GFP⁺ intestinal stem cells (ISCs) from *Lgr5-eGFP-creERT2* mice and Paneth cells (PCs) from WT (non-Lgr5) mice for organoid co-culture. Right: representative pictures and quantification of average number of buds/organoid and % of organoids with the indicated number of buds from ISC-PC organoid co-culture treated with either DMSO or GW3965. The diagram was adapted from ref. 72. (**c**) qPCR analysis of niche signals in SI organoids from WT mice treated with either DMSO or GW3965 in ENR culture medium. (**d-e**) qPCR analysis of niche factors (**d**) or EGFR ligands (**e**) in SI crypts isolated on day0, day1 and day3 post-irradiation from WT mice fed with either STD or GW3965 diet. Datasets for 1 and 3dpi are normalized to their corresponding 0dpi. (**f**) Representative pictures and quantification of average number of buds/organoid and percentage of organoids with the indicated number of buds in SI organoids cultured in ENR or NR (Noggin and R-spondin only) medium and treated with DMSO or GW3965. (**g**) Scheme of the experiment shown in Main Fig. 2d-f. Briefly, SI crypts were isolated from WT mice and cultured in ENR or NR media +/− GW3965 for 5–7 days (primary organoids). Organoids were then digested to single cell suspension and 10,000 live cells were re-seeded for secondary

organoid culture. Secondary organoids were cultured in ENR or NR medium and stimulated with DMSO or GW3965 according to their treatment protocol in primary cultures. Secondary cultures were imaged longitudinally using Incucyte live imaging and plating efficiency, average number of buds/organoid and % of budding organoids was assessed at day 4–7 of culture. (**h**) Representative immunoblot of pERK1/2, tERK1/2 and vinculin from organoids cultured in ENR or NR and treated with either DMSO or GW3965. Western blot quantification of the ratio between phospho-ERK (pERK) and ERK (normalized to Vinculin) from organoids cultured for 5 days with DMSO or GW3965 in ENR or NR media. Vinculin was used as a loading control in the same gel as phospho- and total- ERK1/2 and was cut out to probe with anti-vinculin antibody due to molecular weight difference with tERK1/2 or pERK1/2. For gel source data, see Supplementary Fig. 3. Data are representative of three (**b, d, e** and **h**), four (**a, c**) independent experiments with 3–8 mice/group (each dot represents one mouse). Data are shown as mean ± s.e.m. (**a-f, h**). Significance was assessed by unpaired two-sided *t* test in **a, b** (buds/organoid) and **c-e**), paired two-sided *t* test (**h**), two-way ANOVA with Bonferroni test (**b**, *de novo* buds) and one-way ANOVA with Tukey's post hoc test (**f**). Part of the schematic in panel **g** was drawn using BioRender.com.

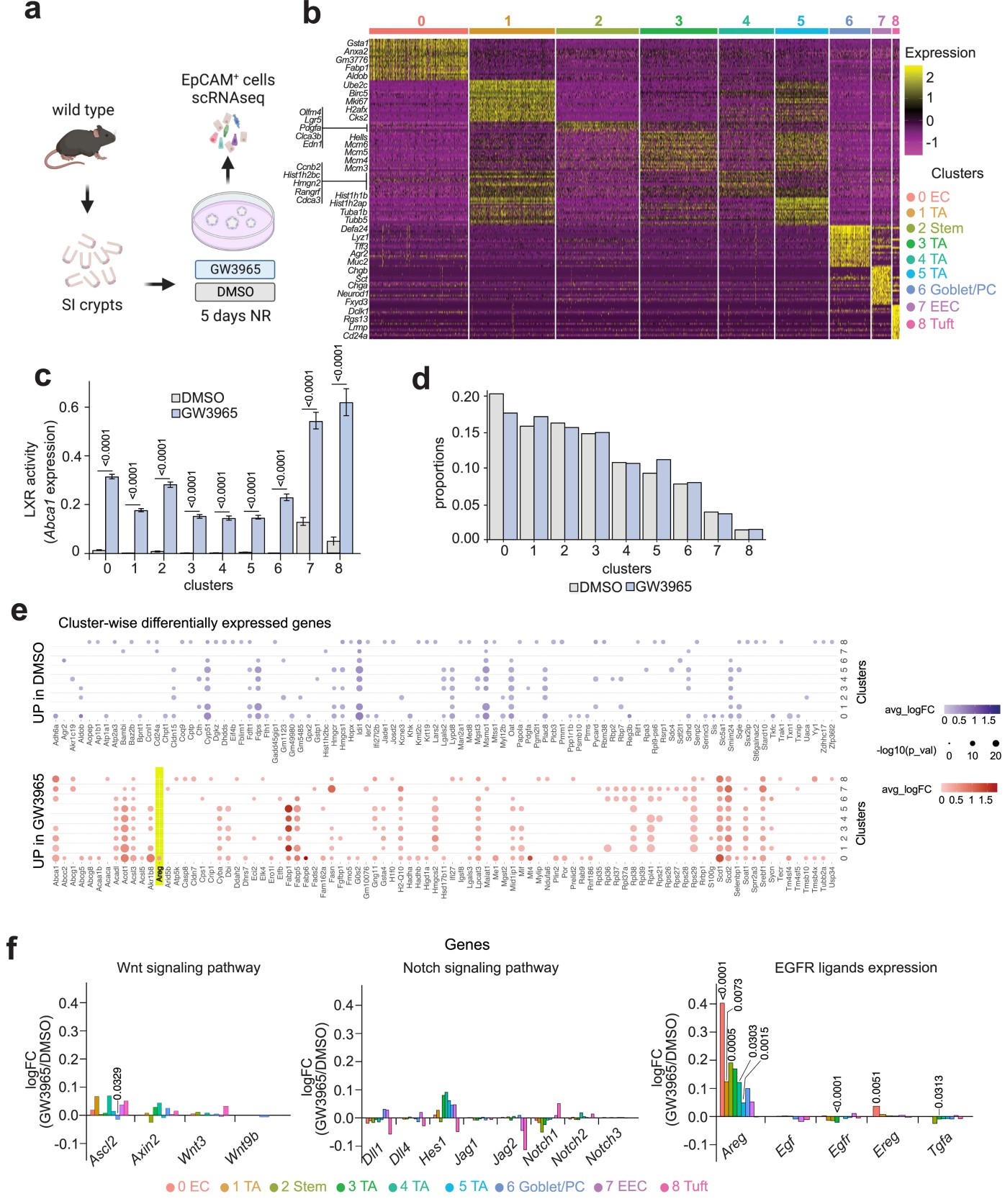

**a** wild type → SI crypts → GW3965 / DMSO → EpCAM⁺ cells scRNAseq, 5 days NR

**b**

**c** LXR activity (*Abca1* expression)

**d** proportions

**e** Cluster-wise differentially expressed genes
UP in DMSO
UP in GW3965

**f** Genes
Wnt signaling pathway
Notch signaling pathway
EGFR ligands expression

Clusters
0 EC
1 TA
2 Stem
3 TA
4 TA
5 TA
6 Goblet/PC
7 EEC
8 Tuft

**Extended Data Fig. 5** | See next page for caption.

**Extended Data Fig. 5 | scRNA sequencing of SI organoids treated with DMSO or GW3965.** (**a**) Schematic of experimental strategy for scRNAseq. (**b**) Heatmap showing expression of top 25 differentially expressed marker genes for each cluster, with 5 DEG symbols noted per cluster. (**c**) Log-transformed expression of LXR target gene *Abca1* in each cluster comparing DMSO and GW3965 treated organoids. (**d**) Cluster-wise cellular proportions comparing DMSO and GW3965 treated organoids. (**e**) Dot plot showing cluster-wise differentially expressed genes between organoids treated with either DMSO or GW3965. The colour intensity indicates the average logFC induction (top: genes upregulated in DMSO treated organoids; bottom: genes upregulated in GW3965-treated organoids) and the size of the dot indicates the p-value. Differences were considered significant at logFC>0.2, p < 0.01 and difference in proportion of expressing cells>0.2. (**f**) Relative fold-change of genes in Wnt, Notch and EGFR signalling modules in GW3965 treated organoids over DMSO in each cluster of the scRNAseq dataset. Data are produced from one experiment (**a-f**). Data are presented as mean ± s.e.m. (**c**). Significance was assessed by expression analysis with Wilcoxon rank-sum test (**c, e, f**). The schematic in panel **a** was drawn using BioRender.com.

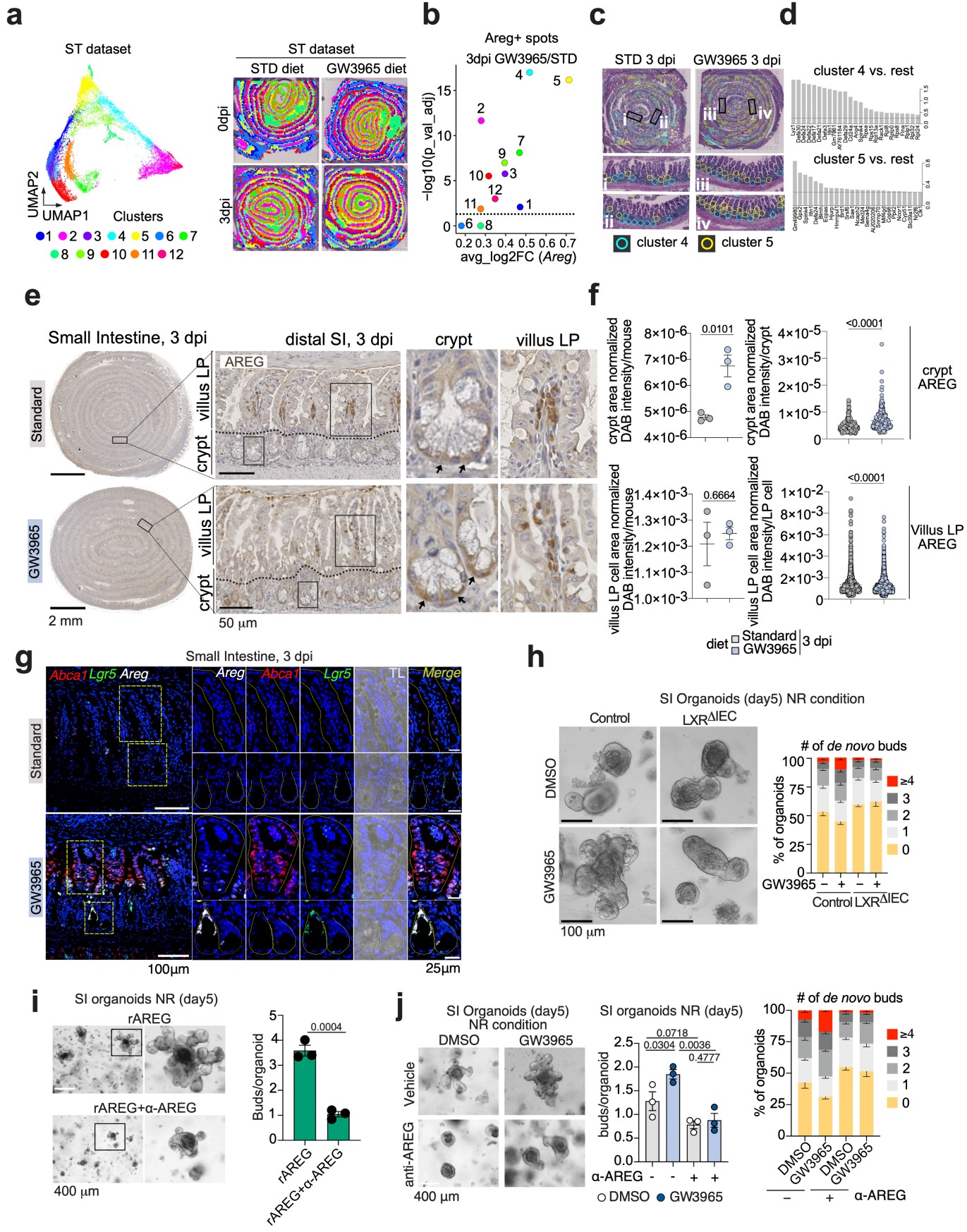

**Extended Data Fig. 6** | See next page for caption.

**Extended Data Fig. 6 | LXR activation induces Areg expression in the SI crypts in response to damage.** (**a**) Unsupervised clustering of the SI spatial transcriptomics (ST) datasets (UMAP on the left and spatial distribution on the right) from mice treated with STD or GW3965 diet at day0 or day3 post irradiation (dpi). (**b**) Differential expression (log2FC and -log10(p-value) shown) of *Areg* in *Areg*⁺ spots, calculated for each cluster, in GW3965-treated mice vs STD fed mice at 3dpi. Dashed line indicates P = 0.05. (**c**) Spatial distribution of top two *Areg*⁺ clusters (i.e., clusters 4 and 5) superimposed onto the H&E SI Swiss rolls from 3dpi STD and GW3965-diet fed mice. Marked with i-iv are insets magnified at the bottom. (**d**) Top 25 genes upregulated in ST clusters 4 and 5 compared to all other clusters. (**e**) Representative images of AREG immunohistochemistry of 3 dpi mouse SI highlighting AREG expression in the crypt epithelial cells (black arrows) and villus lamina propria (LP) cells. (**f**) Quantification of average DAB intensity (AREG expression) in the crypt (normalized to the respective crypt area) and in the Villus-LP (normalized to each DAB⁺ cell). In the plots on the left, each dot represents one mouse. In the plots on the right, each dot represents one crypt or villus LP cell. (**g**) RNA scope analysis of *Abca1* (red), *Lgr5* (green) and *Areg* (white) in the SI of WT mice fed with STD or GW3965-diet at 3 dpi. Crypt-villus view of the merged staining is shown on the left. Zoomed-in insets in the crypt and villus region with single or merged staining are shown on the right. (**h**) Representative pictures and quantification of % of organoids with indicated number of buds in SI organoids treated with DMSO or GW3965 from Villin-Cre:LXRα^{f/f}β^{f/f} (LXR^{ΔIEC}) mice and their littermate LXRαβ^{flox/flox} controls. (**i**) WT SI organoids were cultured for 5 days in NR medium with either recombinant AREG (rAREG) or rAREG together with anti-AREG antibody (α-AREG). Representative pictures and quantification of average number of buds/organoid. The boxed organoid is zoomed in on the right. (**j**) Representative pictures and quantification of average number of buds/organoid and % of organoids with the indicated number of buds from SI organoids treated with DMSO or GW3965 ± α-AREG in NR media for 5 days. Data are representative of one (**a-d, g**), two (**e-f**), three (**i-j**) or four (**h**) independent experiments and each dot represents one mouse (except for plots on the right in panel **f**). Data are shown as mean ± s.e.m. (**f-j**). Significance was assessed by unpaired two-sided *t* test (**f, i**) and by one-way ANOVA with Fisher's LSD post hoc test (**j**, buds/organoid).

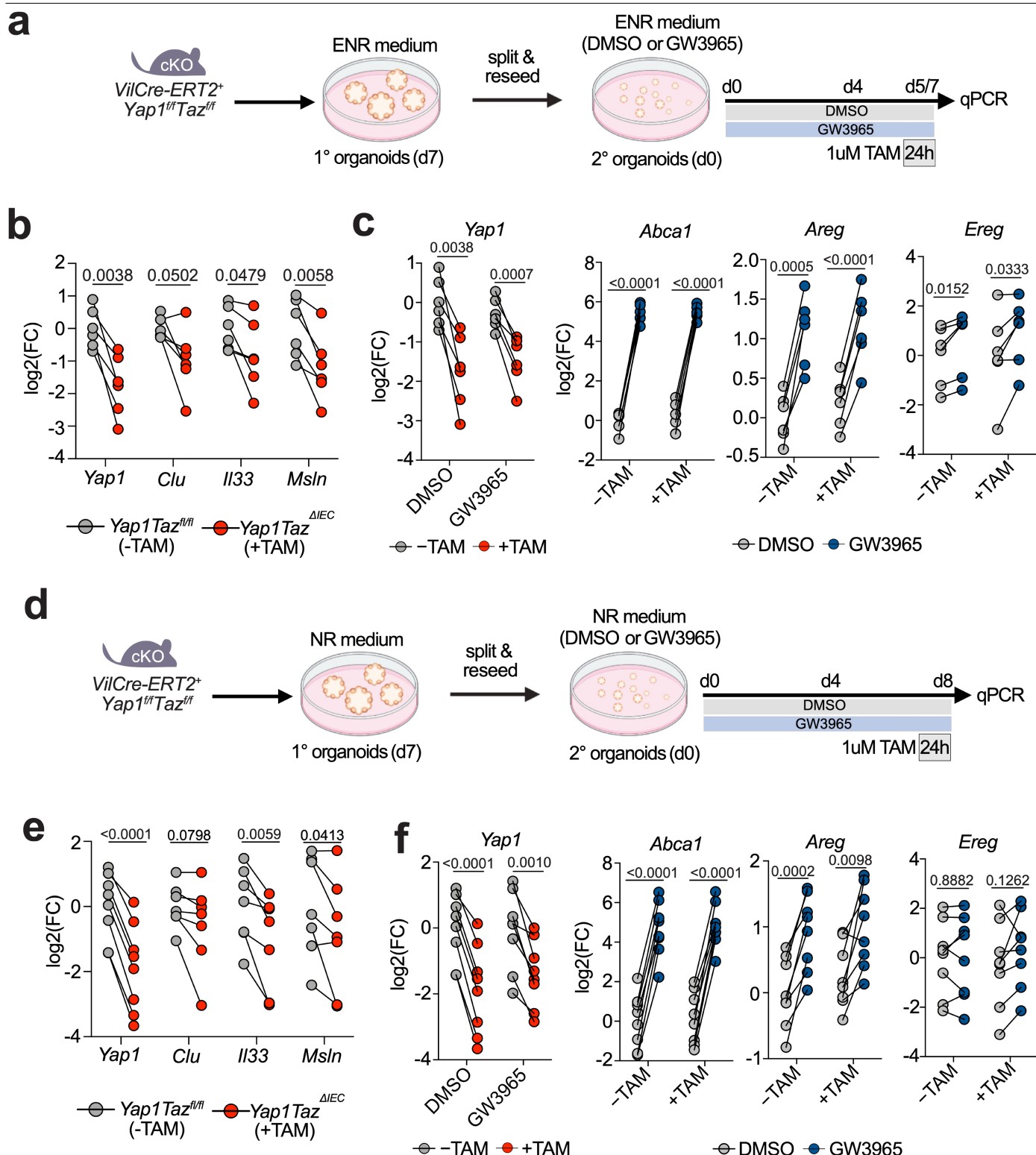

**Extended Data Fig. 7 | Yap-Taz is not necessary for LXR-dependent upregulation of Areg. (a, d)** Schematic representation of the experiment. Briefly, SI crypts from *Villin-CreERT2 Yap^flox/flox Taz^flox/flox* were cultured in ENR (**a-c**) or NR (**d-f**) media for 7 days and then split and re-plated for secondary organoid cultures. Secondary organoids were stimulated with DMSO or GW3965 and treated with tamoxifen (TAM) for the last 24 h of culture.

(**b, e**) qPCR analysis of *Yap1* and YAP-target genes in organoids treated or not with tamoxifen. (**c, f**) qPCR analysis of *Yap1, Abca1, Areg* and *Ereg* in organoids treated with DMSO or GW3965 ± tamoxifen. Data are representative of 4–6 independent experiments with 5–8 mice per group (each dot represents one mouse). Significance was assessed by paired two-sided *t* test. Part of the schematics in panel **a** and **d** was drawn using BioRender.com.

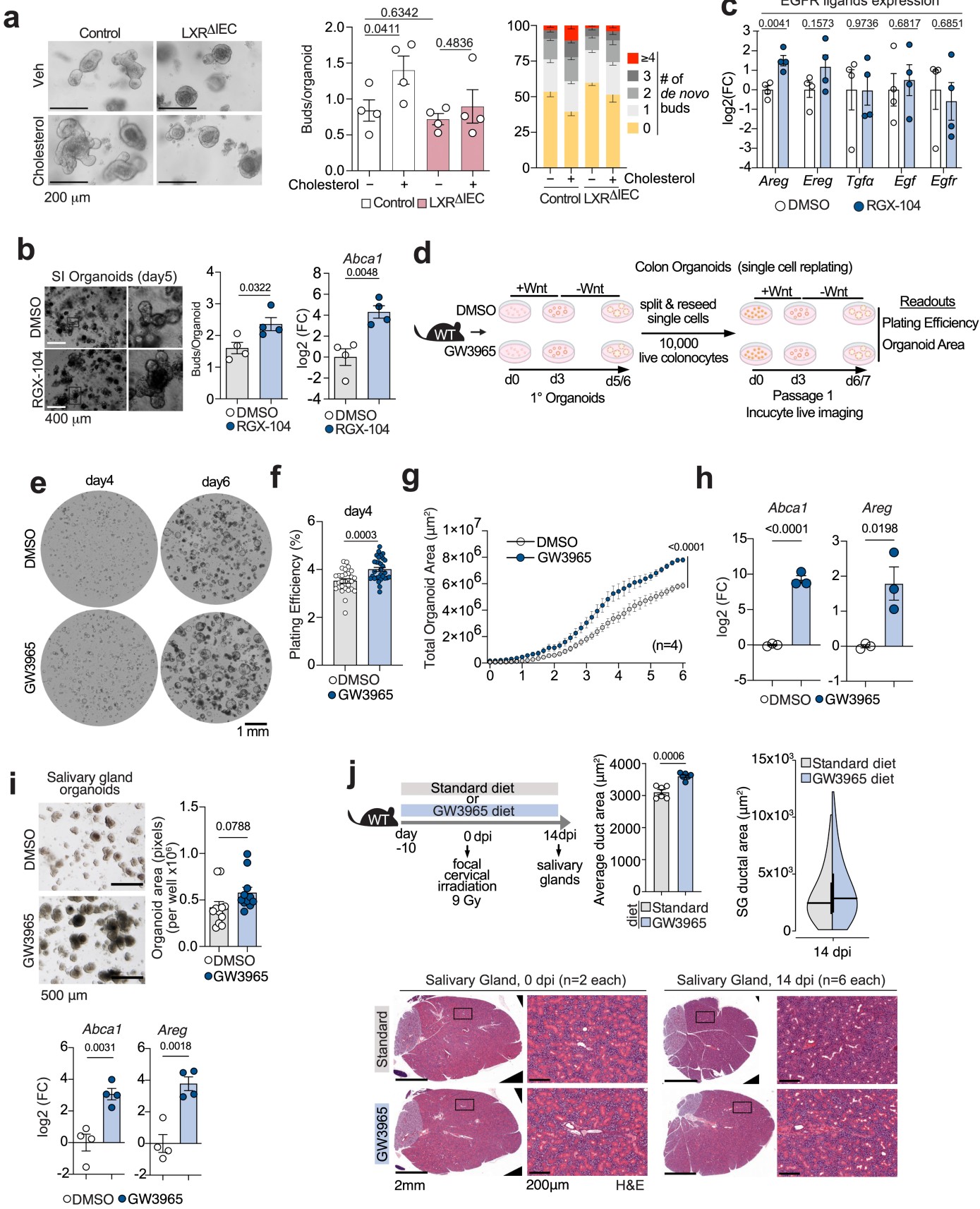

**Extended Data Fig. 8** | See next page for caption.

**Extended Data Fig. 8 | Effect of cholesterol and LXR signalling in organoids from multiple tissue types.** (**a**) Representative pictures, quantification of average number of buds/organoid and quantification of % of organoids with indicated number of buds in SI organoids treated with vehicle or cholesterol from Villin-Cre:LXRα$^{f/f}$β$^{f/f}$ (LXR$^{ΔIEC}$) mice and their littermate LXRαβ$^{flox/flox}$ controls. (**b**) WT SI organoids were cultured for 5 days in NR medium with either DMSO or RGX-104. Left: Representative pictures and quantification of average number of buds/organoid. Right: qPCR analysis of *Abca1* expression. (**c**) qPCR analysis of the expression of EGFR ligands in organoids treated with either DMSO or RGX-104. (**d**) Scheme of the experiment shown in (**e-g**): WT colon primary organoids were cultured for 5-6 days with either DMSO or GW3965 (+exogenous Wnt for the first three days). Organoids were then split and 10,000 live colonocytes were re-seeded for secondary organoids, treated with DMSO or GW3965 (as in the primary cultures) for 6-7 days. Organoids were imaged longitudinally with Incucyte live imaging and plating efficiency, and organoid area was measured at the end of the experiment. (**e-g**) Representative pictures (**e**), quantification of plating efficiency (**f**) and total organoid area over time (**g**) of WT organoids as described in **d**. For organoid area analyses over time, same wells were imaged every 4 h for a peiod of 6-7 days. (**h**) qPCR analysis

expression of *Abca1* and *Areg* in colonic organoids treated with DMSO or GW3965. (**i**) WT salivary gland organoids were cultured for 4-6 days with either DMSO or GW3965. Representative pictures, quantification of the organoids area/well and qPCR analysis of *Abca1* and *Areg*. (**j**) Scheme showing WT mice fed with either standard (STD) or GW3965 diet for 10 days followed by focal irradiation of the neck region targeting salivary glands. Representative H&E and zoomed inset of salivary glands from mice fed with STD or GW3965 diet at 0 dpi (left, 10 days after the diet) and at 14 dpi (right, 14 days after the irradiation). Measurement of salivary gland duct area was used as a proxy for salivary gland regeneration. Data are representative of 2–4 independent experiments and each dot represents one mouse, except for **f** where each dot represents one well. Data are shown as mean ± s.e.m. (**a-i**). In **j**, data are shown as mean ± s.e.m. for the bar plot and median ± quartile for the violin plot. Significance was assessed by unpaired two-sided *t* test (**b, c, f, h** and **i**), unpaired two-sided t-test between the slopes for DMSO and GW3965 treated organoids obtained by linear regression analysis (**g**), and one-way ANOVA with fisher's LSD post hoc test (**a**). Part of the schematic in panel **d** was drawn using BioRender.com and part of the schematic in panel **j** was adapted from ref. 19, CC-BY 4.0.

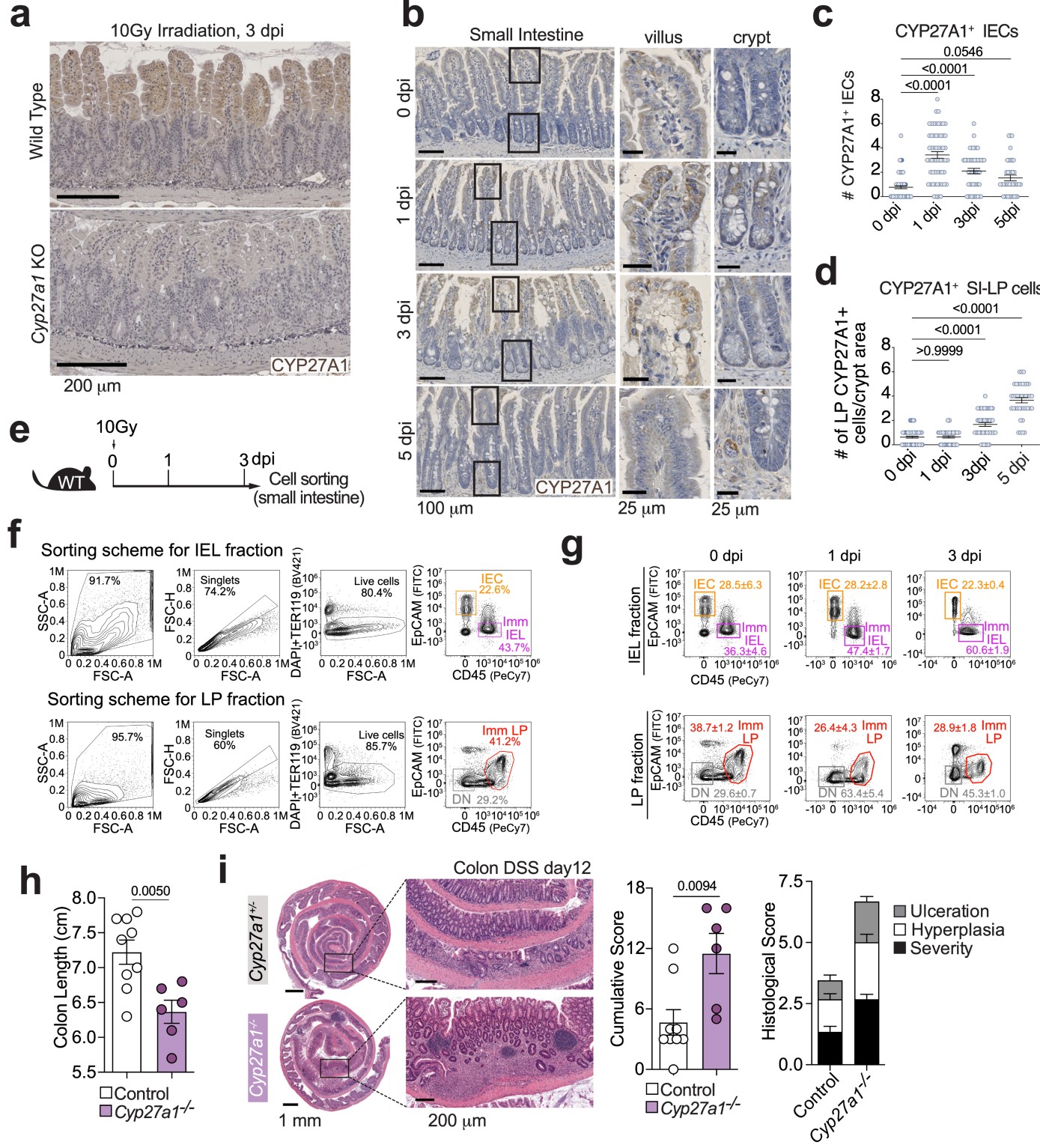

**Extended Data Fig. 9** | See next page for caption.

**Extended Data Fig. 9 | Characterization of Cyp27a1 expression and function in intestinal regeneration.** (**a**) Validation of CYP27A1 antibody to check expression of CYP27A1 in the mouse small intestine. Small intestinal tissue from *Cyp27a1*[-/-] mice was used to demonstrate the specificity of the antibody. (**b**) Representative images showing CYP27A1 expression in mouse SI during different days post-irradiation. On the right are the zoom-in of the boxed images in the villus and crypt region. (**c, d**) Quantification of number of CYP27A1[+] intestinal epithelial cells (IECs) (**c**) and small intestinal lamina propria (SI-LP) (**d**) cells/crypt area (quantified using immunohistochemical staining). (**e**) Scheme showing the experiment for sorting different cell types following irradiation (dpi) to determine the cellular source of *Cyp27a1*. (**f**) Representative dot plot showing gating strategy for sorting EpCAM[+] epithelial cells (IEC) and CD45[+] immune cells from the intraepithelial compartment (Imm IEL); CD45[+] immune cells from the lamina propria compartment (Imm LP) and Epcam[-]CD45[-] double negative (DN) cells from the LP compartment of mouse small intestine. (**g**) Representative dot plots showing Epi, Imm IEL, Imm LP and DN cells from 0-1-3 dpi mouse SI. (**h-i**) *Cyp27a1*[-/-] mice and control littermates (*Cyp27a1*[+/-] or *Cyp27a*[+/+]) were treated with 2% DSS for 7 days followed by 7 days of recovery. *Cyp27a1*[-/-] and controls were housed separately at the start of the experiment. (**h**) Graph shows the colon length in control and *Cyp27a1*[-/-] mice at the end of the DSS colitis experiment. (**i**) Representative H&E images of colon Swiss rolls and histological score from *Cyp27a1*[-/-] and littermate control at the end of the DSS colitis experiment. Data are representative of one (**a-d**), two (**e-g**) or three (**h-i**) independent experiments. Each dot represents one mouse except in (**c-d**), where each dot represents one crypt area. Data are shown as mean ± s.e.m. in (**c-d, g-i**). Significance was assessed by one-way ANOVA with Tukey's post hoc test in (**c, d**), unpaired two-sided *t* test in (**h, i**). Part of the schematic in panel **e** was adapted from ref. 19, CC-BY 4.0.

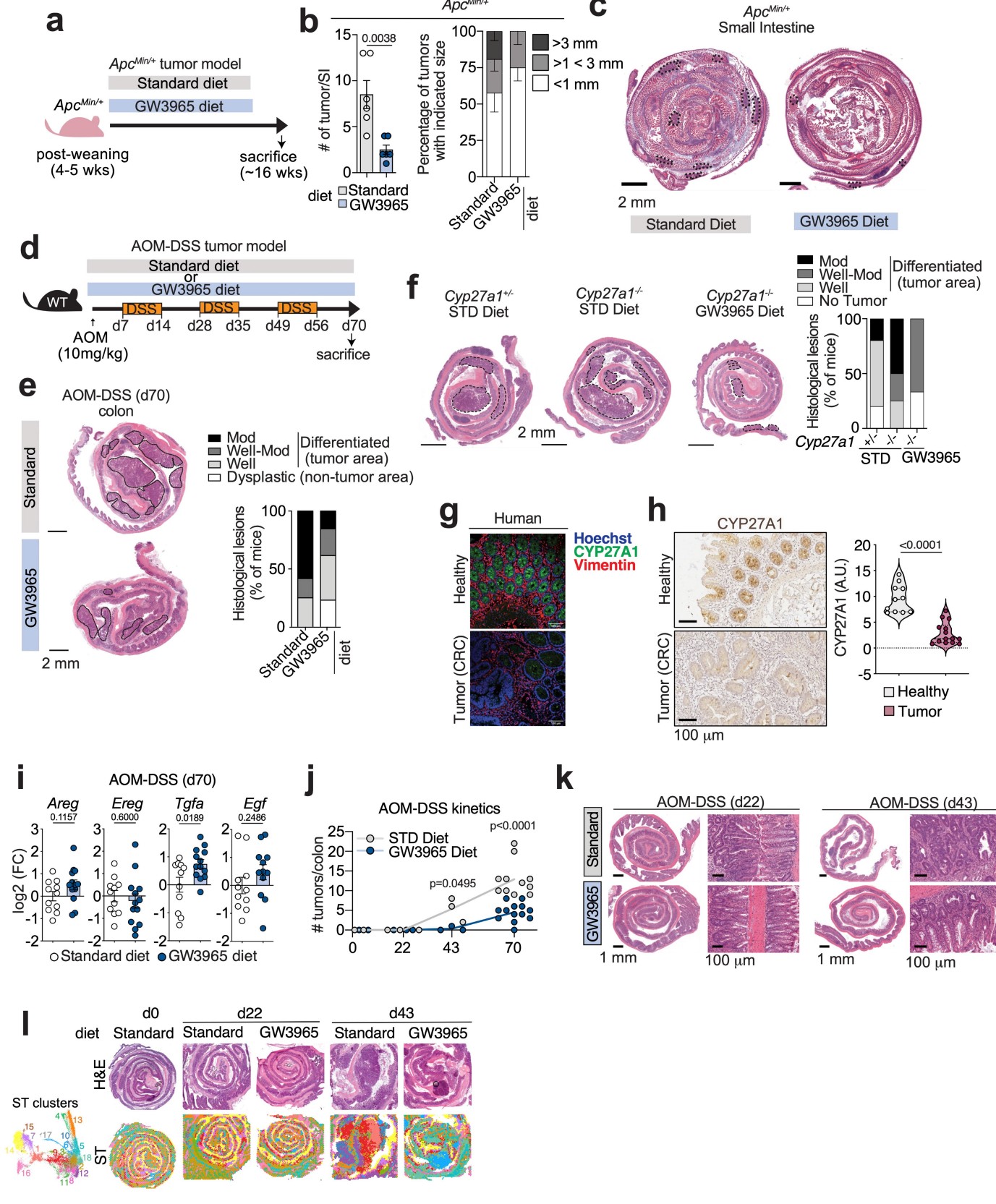

**Extended Data Fig. 10** | See next page for caption.

**Extended Data Fig. 10 | LXR activation suppresses intestinal tumorigenesis.** (**a**) Scheme of the experiment showing *Apc*^*Min/+*^ mice fed with either standard or GW3965 diet post-weaning until approximately 16 weeks of age when mice were sacrificed and evaluated for tumour development. (**b**) Quantification of tumour numbers and size in the SI of *Apc*^*Min/+*^ mice fed with standard or GW3965 diet. (**c**) Representative H&E stained SI Swiss rolls of *Apc*^*Min/+*^ mice fed with either the standard or GW3965 diet. Areas of tumour are outlined by black dotted lines. (**d**) Scheme of the experiment showing WT mice fed with standard or GW3965 diet and injected with AOM followed by 3 cycles of DSS. Mice were sacrificed at day 70. (**e**) Representative H&E stained colon Swiss rolls and quantification of histological lesions of mice undergoing AOM-DSS tumorigenesis and fed with either the standard or GW3965 diet. Areas of tumour are outlined by black dotted lines. (**f**) Representative H&E stained colon Swiss rolls and quantification of histological lesions of *Cyp27a1*^*−/−*^ and littermate control mice undergoing AOM-DSS tumorigenesis and fed with either the standard or GW3965 diet. Areas of tumour are outlined by black dotted lines. (**g**) Representative images of immunofluorescence staining of CYP27A1 (green), vimentin (red) and nuclei (Hoechst) in human colonic tissue. (**h**) Representative pictures and quantification of CYP27A1 immunohistochemical staining in human tissue microarray with healthy and CRC tumour samples. (**i**) qPCR analysis of expression of EGFR ligands in the colonic tumour biopsies at the end (day 70) of AOM-DSS experiment. (**j-k**) WT mice fed with standard or GW3965 diet were injected with AOM followed by 3 cycles of DSS and samples were collected after each DSS cycle at indicated time points. Quantification of grossly visible tumours during the course of tumour development (**j**). Mice fed with either the standard or GW3965 diet (for 22, 43 or 70 days), but not challenged with AOM-DSS were used as day0 samples. (**k**) Representative H&E images of mice colon fed with standard or GW3965 diet and undergoing AOM-DSS tumorigenesis at day 22 and day 43 post-AOM injection. (**l**) H&E stained colonic Swiss rolls and spatial transcriptomics (ST) representation of unsupervised clustering of colonic tissue from WT mice fed with STD or GW3965-diet at day 0, day 22 or day 43 of AOM-DSS. Data are representative of one (**l**) or 2–4 (**a-f**, and **i**) independent experiments with 6–13 mice per condition (each dot represents one mouse). For d0, 22 and 43 of AOM-DSS tumour kinetics (**j-k**), one experiment with 3-4 mice/time point were used, and for d70 two experiments with 11–13 mice were used (as shown in Fig. 4a). In (**h**) each dot represents one spot in the CRC tumour tissue microarray. Data are shown as mean ± s.e.m. (**b, i**), median and quartiles (**h**) or staggered replicates with lines connecting the mean (**j**). Significance was assessed by unpaired two-sided *t* test in (**b, h, i, j**). Part of the schematics in panels **a** and **d** were adapted from ref. 19, CC-BY 4.0.

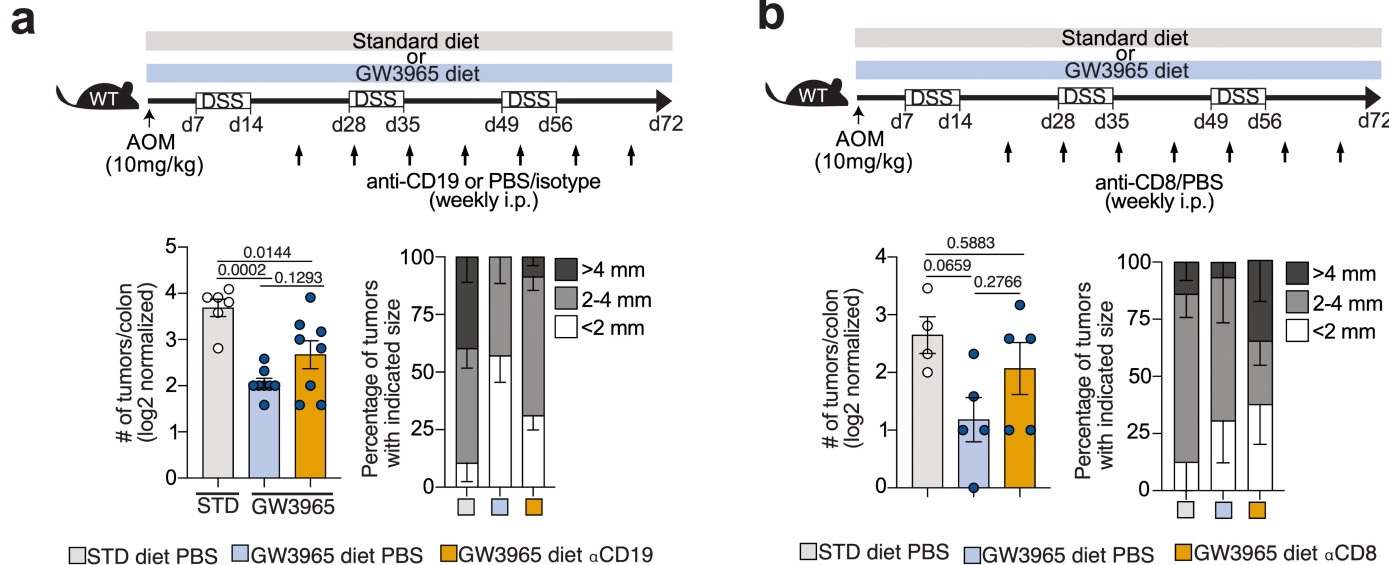

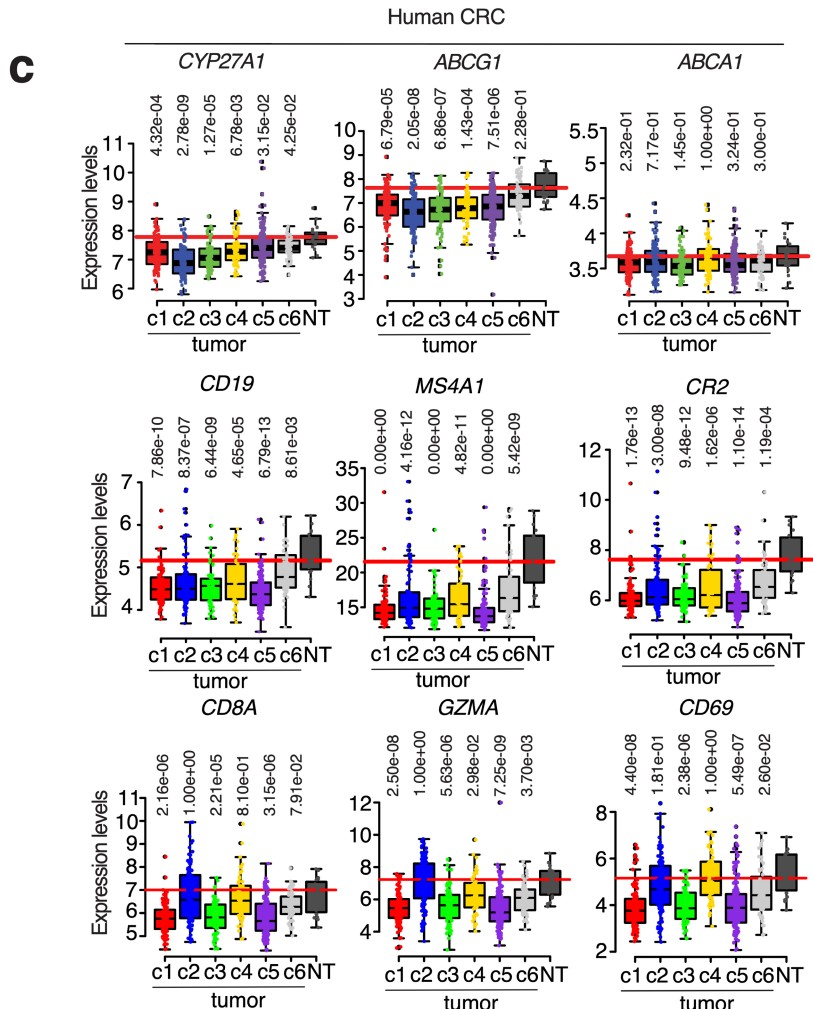

**Extended Data Fig. 11** | See next page for caption.

**Extended Data Fig. 11 | Mechanism of LXR mediated anti-tumour effects.**
(**a, b**) WT mice were fed with either standard or GW3965 diet and exposed to
AOM-DSS tumour model. Starting from d22 after AOM injection, mice were
treated with repeated (weekly) intraperitoneal injections of either 300 μg
anti-CD19 (αCD19) (**a**) or anti-CD8 (αCD8) (**b**) antibody. Control mice were
injected with PBS. Mice were sacrificed at day 70 and macroscopic tumour
numbers were counted. Quantification of tumour numbers (log2 normalized)
and sizes are shown. (**c**) Expression levels of *CYP27A1*, *ABCG1*, *ABCA1*, *CD19*,
*MS4A1*, *CR2*, *CD8A*, *GZMA*, and *CD69* in a human CRC whole genome
transcriptome array dataset stratified in six tumour subtypes (c1 to c6) and
non-tumour control (NT). Data are representative of two independent
experiments (**a,b**) with 4–8 mice per condition (each dot represents one
mouse). Data are shown as mean ± s.e.m. (**a,b**) and quartiles (**c**). Significance
was assessed by One-way ANOVA with Tukey's post-hoc test in (**a,b**) or two-way
ANOVA with Dunnett's post hoc test in (**c**). Part of the schematics in panels **a** and
**b** were adapted from ref. 19, CC-BY 4.0.

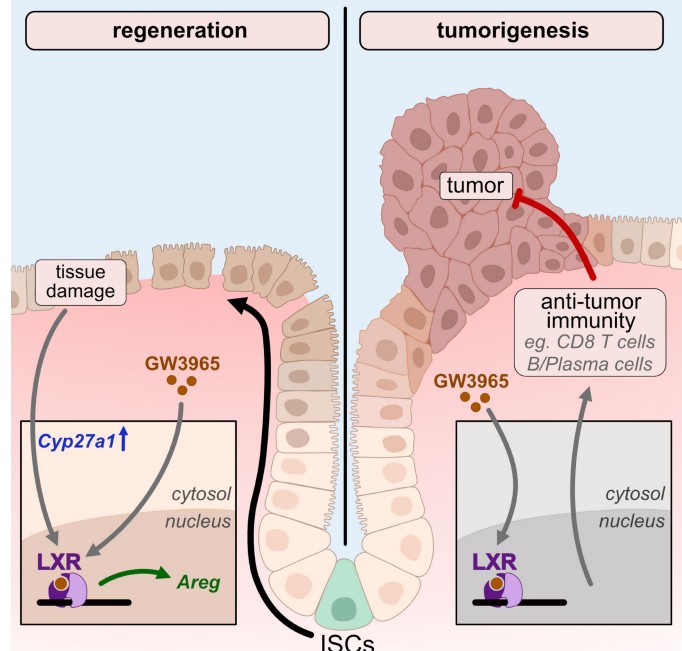

**Extended Data Fig. 12 | LXR unlinks intestinal regeneration and tumorigenesis.** Schematics showing the proposed model. Left side: intestinal tissue damage leads to increased expression of the enzyme *Cyp27a1*, which metabolizes oxysterol ligands for LXR. Either endogenous or synthetic (GW3965) LXR ligands act in epithelial cells to induce the EGFR ligand amphiregulin (*Areg*) promoting intestinal epithelial regeneration. Right side: In the context of tumour development, LXR activation amplifies anti-tumour immunity (dependent on B cells and CD8 T cells) controlling intestinal tumour growth.

# Reporting Summary

## Statistics

For all statistical analyses, confirm that the following items are present in the figure legend, table legend, main text, or Methods section.

| n/a | Confirmed | |
|---|---|---|
| ☐ | ☒ | The exact sample size (*n*) for each experimental group/condition, given as a discrete number and unit of measurement |
| ☐ | ☒ | A statement on whether measurements were taken from distinct samples or whether the same sample was measured repeatedly |
| ☐ | ☒ | The statistical test(s) used AND whether they are one- or two-sided *Only common tests should be described solely by name; describe more complex techniques in the Methods section.* |
| ☒ | ☐ | A description of all covariates tested |
| ☐ | ☒ | A description of any assumptions or corrections, such as tests of normality and adjustment for multiple comparisons |
| ☐ | ☒ | A full description of the statistical parameters including central tendency (e.g. means) or other basic estimates (e.g. regression coefficient) AND variation (e.g. standard deviation) or associated estimates of uncertainty (e.g. confidence intervals) |
| ☐ | ☒ | For null hypothesis testing, the test statistic (e.g. *F*, *t*, *r*) with confidence intervals, effect sizes, degrees of freedom and *P* value noted *Give P values as exact values whenever suitable.* |
| ☒ | ☐ | For Bayesian analysis, information on the choice of priors and Markov chain Monte Carlo settings |
| ☒ | ☐ | For hierarchical and complex designs, identification of the appropriate level for tests and full reporting of outcomes |
| ☒ | ☐ | Estimates of effect sizes (e.g. Cohen's *d*, Pearson's *r*), indicating how they were calculated |

*Our web collection on statistics for biologists contains articles on many of the points above.*

## Software and code

Policy information about availability of computer code

| Data collection | Flow cytometry: FACSDiva (BD), LSR Fortessa(BD) Floresence activated cell sorting: SH800S(SONY), FACS ARIA II (BD) Confocal microscope: LSM880 (Zeiss) Cell culture imaging: Incucyte S3 Live-Cell Analysis Instrument (Sartorius) (v2023A) qRT-PCR: CFX 384 Realtime C1000 Touch (BioRad) Next Generation Sequencing: 10X chromium Controller, NovaSeq S1 flow cell (Illumina) |
|---|---|
| Data analysis | GraphPad Prism 9, version 9.5.0 Microsoft Excel 16.51 FlowJo v10 QuPath 0.2.3 Fiji (ImageJ2, version:2.14.0/1.54f) R (versions stated in the code) Visiopharm (version 23.01) NDP.View2 Incucyte® Software (v2023A) Incucyte® Organoid Analysis Software Module (Cat. No. 9600-0034) |

For further package details please see in methods section and source code repository.

For manuscripts utilizing custom algorithms or software that are central to the research but not yet described in published literature, software must be made available to editors and reviewers. We strongly encourage code deposition in a community repository (e.g. GitHub). See the Nature Portfolio guidelines for submitting code & software for further information.

## Data

Policy information about availability of data

All manuscripts must include a data availability statement. This statement should provide the following information, where applicable:

- Accession codes, unique identifiers, or web links for publicly available datasets
- A description of any restrictions on data availability
- For clinical datasets or third party data, please ensure that the statement adheres to our policy

RNAseq data generated in this study has been deposited in the NCBI Gene Expression Omnibus (GEO) repository with the following accession codes. Bulk RNAseq in AOM-DSS CRC experiment (GSE180078), scRNA-seq in organoids (GSE180079), spatial transcriptomics in AOM-DSS CRC experiment (GSE227598), spatial transcriptomics in irradiation induced injury-repair in small intestine (GSE227742) and scRNA-seq of small intestinal cells following LXR activation (GSE227726). All these datasets are now publicly available. All other publicly available datasets used herein are available on GEO with accession codes: GSE14879449, GSE1177834, GSE1310325, GSE3958247. Source data are provided with this paper.

All the custom codes generated in this study have been deposited in a GitHub public repository
(https://github.com/ejvillablancaLab/LXR_in_regeneration_and_tumorigenesis),
which can be accessed via a DOI at Zenodo: https://doi.org/10.5281/zenodo.13133717.

For further details please see methods section.

# Field-specific reporting

Please select the one below that is the best fit for your research. If you are not sure, read the appropriate sections before making your selection.

☒ Life sciences          ☐ Behavioural & social sciences          ☐ Ecological, evolutionary & environmental sciences

For a reference copy of the document with all sections, see nature.com/documents/nr-reporting-summary-flat.pdf

# Life sciences study design

All studies must disclose on these points even when the disclosure is negative.

| | |
|---|---|
| Sample size | No statistical methods were used to predetermine sample size. To determine the sample size, we conducted pilot experiments to estimate an appropriate number that would ensure significant results in statistical tests. The exact sample sizes used in this study are specified in each figure legend. |
| Data exclusions | One data point in the control (Cyp27a1+/+ or +/-) group in Fig. 3h, and one data point in the Standard diet group in Extended Data Fig. 10b were excluded which were calculated using the GraphPad outlier calculator for significant outlier in the group by ROUT method. No other data or animals were excluded from analysis, except for clear technical failure. |
| Replication | 10X scRNAseq, spatial transcriptomics in irradiation and CRC, and CRC bulk RNAseq experiments were performed once due to the high costs and intensive resource demands. These techniques capture significant sample heterogeneity, providing insights that can partially substitute for biological replicates. However, for scRNAseq and bulk RNASeq, we used a pool of 2-3 biological replicates which were then treated as single sample for sequencing. For spatial transcriptomics, the findings were further validated in multiple biological replicates using alternative methods.<br><br>All other experiments were repeated as individual experiments as stated in each figure legend, and reproducibility was confirmed. |
| Randomization | Mice were randomly assigned to experimental groups and received the designated treatments as outlined in the experimental protocol. |
| Blinding | All the organoids crypt domain quantification was done blindly, meaning the investigator quantifying the organoids was unaware of the genotype and the treatment conditions. Likewise, histology quantification (BrdU, cleaved Cas3, CYP27A1, AREG staining, scoring of DSS colitis and AOM-DSS tumor grading) were done with code-labeled samples.<br><br>For all other experiments, the researchers were not blinded to the treatment or genotypes of the mice to avoid handling errors due to the involvement of multiple scientists. However, the experimental design and use of appropriate controls ensured the accuracy and reproducibility of all measurements and analyses. |

# Reporting for specific materials, systems and methods

We require information from authors about some types of materials, experimental systems and methods used in many studies. Here, indicate whether each material, system or method listed is relevant to your study. If you are not sure if a list item applies to your research, read the appropriate section before selecting a response.

## Materials & experimental systems

| n/a | Involved in the study |
|---|---|
| ☐ | ☒ Antibodies |
| ☒ | ☐ Eukaryotic cell lines |
| ☒ | ☐ Palaeontology and archaeology |
| ☐ | ☒ Animals and other organisms |
| ☐ | ☒ Human research participants |
| ☒ | ☐ Clinical data |
| ☒ | ☐ Dual use research of concern |

## Methods

| n/a | Involved in the study |
|---|---|
| ☒ | ☐ ChIP-seq |
| ☐ | ☒ Flow cytometry |
| ☒ | ☐ MRI-based neuroimaging |

## Antibodies

**Antibodies used**

The following antibodies (supplier, clone, catalogue number, dilution) were used for staining and IHC:

CD19-FITC(BioLegend, clone 6D5, Cat#115506, 1:200)
CD45.2-BV650(BioLegend, clone 104, Cat#109836, 1:200)
CD90.2-APC-Cy7(BioLegend, clone 30-H12, Cat#105328, 1:200)
CD64-BV421(BioLegend, clone X54-5/7.1, Cat# 139309, 1:200)
CD11b-BV786(Invitrogen, M1/70, Catalog # 417-0112-82, 1:200)
CD11c-PE-Cy7(BioLegend, clone N418, Cat# 117317, 1:200)
MHC-II-V500(BD-Biosciences, clone M5/114.15.2 , Cat# 562366, 1:200)
Ly6G-PE (BD-Biosciences, clone 1A8, Cat# 551461, 1:200)
Ly6C-PcP5.5(Invitrogen, clone HK1.4, Cat# 45-5932-82, 1:200)
CD3-AF700(BD-Biosciences, clone 500A2, Cat# 557984, 1:200)
CD103-APC(Invitrogen, clone 2E7, Cat# 17-1031-80, 1:200)
CD31-PE, Biolegend, clone: Mec13.3, cat# 102507, 1:500
CD45.2-PE, eBioscience, clone: 30-F11, cat# 12-0451-82, 1:500
Ter-119-PE, Biolegend, clone: Ter119, cat# 116207, 1:500
CD24-PacificBlue, Biolegend, clone: M1/69, cat# 101819, 1:500
EpCAM-APC, eBioscience, clone G8.8, cat# 17-5791-82, 1:500
EpCAM-FITC, Biolegend, clone: G8.8, cat# 118207, 1:200
CD45.2-PeCy7, Biolegend, clone: 104, cat#109829, 1:200
Ter119-Pacific Blue, Biolegend , clone: Ter119, cat# 116231, 1:200
Rat anti-BrdU, Abcam, clone: BU1/75 ICR1, cat# ab6326, 1:200-300
Rabbit anti-cleaved caspase3, Cell Signaling, clone: 5A1E, Cat# 9664S, 1:200
Rabbit anti-CYP27A1, Abcam, clone: EPR7529, Cat# ab126785, 1:200
Mouse anti-AREG, Santacruz Biotechnology, clone: G4, Cat# sc-74501, 1:500
Mouse anti-Vimentin, Abcam, clone: RV202, cat# ab8978, 1:1000
Purified anti-mouse/human CD45R/B220 antibody, Biolegend, Clone: RA3-6B2, cat# 103201, 1:200
Rabbit monoclonal anti-OLFM4, Cell Signaling Technology, Clone: D6Y5A, cat# 39141, 1:300
Mouse monoclonal anti-Vinculin, Sigma, clone: hVIN-1, cat# V9131, 1:1000
Goat anti-rabbit AF488, Invitrogen, Cat# A32731, 1:500
Donkey anti-rat AF647, Jackson ImmunoResearch, Cat# 712-605-153, 1:500
Rabbit polyclonal anti-ERK1/2, Cell Signaling Technology, cat# 9102S; 1:1000
Rabbit polyclonal Phospho-ERK1/2, Cell Signaling Technology, cat# 9101S; 1:1000
Biotinylated goat anti-mouse (Vector Labs, Cat #BA-9200, 1:300)
Biotinylated goat anti-rabbit (Vector Labs, Cat# is BA-1000, 1:300)
Goat anti-mouse IgG conjugated with Alexa Fluor 546 (Invitrogen, Cat# A-11003, 1:200)

Antibodies used in vivo:
InVivoMAb anti-mouse CD19, Bioxcell, clone: 1D3, cat# BE0150, injected i.p. at 300μg/mouse each time.
InVivoMAb rat IgG2a isotype control, anti-trinitrophenol, Bioxcell, clone: 2A3, catlog# BE0089,  injected i.p. at 300 μg /mouse each time.
InVivoMAb anti-mouse CD8b, BioXcell, Clone: Lyt3.2, cat# BE0223,  injected i.p. at 300 μg /mouse each time

Antibodies used in organoid culture:
Polyclonal Goat IgG Mouse Amphiregulin antibody, R&D systems, AF-989SP (1.5μg/ml)

**Validation**

For Rabbit anti-CYP27A1 antibody (obtained fromAbcam, clone: EPR7529, Cat# ab126785), we validated the antibody using our WT and Cyp27a1 KO mouse small intestine samples (dilution 1:200).

For all other antibodies listed above were validated by the manufacturer and/or by peer-reviewed article.

Antibody (Mouse anti-AREG) from Santacruz Biotechnology validation information could be found in the link: https://www.scbt.com/p/amphiregulin-antibody-g-4#citations

Mouse monoclonal anti-Vinculin from Sigma, the validation information and specification sheet can be found in this link: https://www.sigmaaldrich.com/SE/en/product/sigma/v9264

Donkey anti-rat AF647 from Jackson ImmunoResearch, the validation information and specification sheet can be found in this link: https://www.jacksonimmuno.com/catalog/products/712-605-153

Polyclonal Goat IgG Mouse Amphiregulin antibody from R&D system, the validation information and specification sheet can be found in this link: https://www.rndsystems.com/products/mouse-amphiregulin-antibody_af989

Biolegend antibodies validation information can be found in:
https://www.biolegend.com/en-us/quality/quality-control
For all flow cytometry antibodies, specificity testing of 1-3 target cell types with either single- or multi-color analysis (including positive and negative cell types). Once specificity is confirmed, each new lot must perform with similar intensity to the in-date reference lot. Brightness (MFI) is evaluated from both positive and negative populations. Each lot product is validated by QC testing with a series of titration dilutions. For all IHC antibodies, purified antibodies are tested for purity by SDS-PAGE gel electrophoresis. IgG antibodies are required to have purity >95%. Fluorophore and enzyme-conjugated antibodies follow strict manufacturing specifications to ensure performance. Each lot is validated by QC testing as stated on the TDS to confirm specificity and lot-to-lot consistency.

ThermoFisher (eBioscience) antibodies validation information can be found in:
https://www.thermofisher.com/se/en/home/life-science/antibodies/invitrogen-antibody-validation.html
Thermo Fisher Scientific's Invitrogen antibody validation process typically includes various methods to ensure antibody specificity and reliability, including:
Knockout Validation: Using knockout cell lines or tissues lacking the target protein to confirm antibody specificity by demonstrating the lack of signal in these samples; Transfection Validation: Demonstration of staining pattern changes when the target protein is transfected into cells that do not typically express the protein; Immunoprecipitation: Verifying antibody performance through immunoprecipitation techniques to confirm binding specificity; Peptide Array: Validating antibody binding specificity using peptide arrays containing the antigenic sequence; Overexpression Studies: Utilizing overexpression studies to verify antibody specificity and sensitivity; Western Blotting: Checking for specific band detection at the expected molecular weight. These validation methods help ensure that Thermo Fisher Scientific's antibodies provide accurate and reliable results for researchers.

BD biosciences antibodies validation information can be found in:
https://www.bdbiosciences.com/en-us/products/reagents/flow-cytometry-reagents/research-reagents/quality-and-reproducibility
The specificity is confirmed using multiple methodologies that may include a combination of flow cytometry, immunofluorescence, immunohistochemistry or western blot to test staining on a combination of primary cells, cell lines or transfectant models. All flow cytometry reagents are titrated on the relevant positive or negative cells. To save time and cell samples for researchers, test size reagents are bottled at an optimal concentration with the best signal-to-noise ratio on relevant models during the product development. To ensure consistent performance from lot-to-lot, each reagent is bottled to match the previous lot MFI. BD Biosciences ensures high-quality reagents through a meticulous quality control process, outlined as the following: ISO 9001 Compliance, R&D to Manufacturing Transfer, Standard Operating Procedures (SOPs) and Quality Control Testing. Newly manufactured lots undergo stringent quality control tests alongside accepted lots, ensuring that the performance of each new lot is reliable and consistent, providing researchers with assurance and confidence in their research outcomes.

Cell signaling antibodies validation information can be found in:
IHC antibodies: https://www.cellsignal.com/about-us/our-approach-process/antibody-validation-immunohistochemistry
Cell Signaling Technology (CST) offers over 800 antibodies validated for IHC, as well as IHC diluents, detection reagents, substrates and controls, to ensure that your IHC studies yield accurate and reproducible results. The determination of target specificity in immunohistochemical analysis requires multiple validation steps. CST scientists use a variety of methods, as appropriate, to validate each IHC-recommended antibody, ensuring that the staining you observe with each CST antibody is specific and believable.

Western blot antibodies:
https://www.cellsignal.com/about-us/our-approach-process/antibody-validation-western-blotting
The accuracy of western blot results relies heavily of the quality of the primary antibody employed in the immunoblotting. Cell Signaling Technology (CST) provides the highest quality primary and secondary antibodies available for western blotting. CST antibodies are produced in-house and validated extensively according to a rigorous protocol.

Abcam antibodies validation information can be found in:
https://go.myabcam.com/BiophysicalQuality
Abcam company uses a toolkit of tests established by the biopharma industry for therapeutic antibodies to assess sequence identity, sequence integrity, aggregation, purity, and concentration. These include liquid chromatography-mass spectrometry (LC-MS), dynamic light scattering (DLS) and high-performance liquid chromatography (HPLC). No other reagent supplier carries out this level of analysis, providing the highest level of assurance to researchers. The Biophysical QC includes: Recombinant technology – for exceptional batch-to-batch consistency, sensitivity and specificity; Extensive application testing – standard for all our antibodies, including immunohistochemistry (IHC) on tissue microarrays; Advanced validation – antibodies validated in techniques specifically of interest for their target, including mass cytometry, ChIC/CUT&RUN and biological activity, identified by our 'Advanced Validation' tag; Knock-out validation – performed using an extensive library of human knock-out cell lines.

R&D systems antibodies validation:
https://www.rndsystems.com/products/rd-systems-approach-antibody-quality
R&D Systems carefully tests every antibody we produce to ensure antibodies performance. Each antibody is manufactured under controlled conditions, undergoing rigorous quality control testing to ensure lot-to-lot consistency and outstanding performance in all applications listed on R&D systems datasheets. All antibodies are tested for cross-reactivity with closely related molecules using a variety of applications, including direct ELISA, to ensure specificity.

# Animals and other organisms

Policy information about studies involving animals; ARRIVE guidelines recommended for reporting animal research

| | |
|---|---|
| Laboratory animals | Mouse (Mus musculus)<br>All mice used were in C57BL/6J background. In all experiments animals used were between 8-20 weeks of age. Both males and female mice were used in the study, except for the AOM-DSS tumor experiments in Cyp27a1 KO mouse line. Specifically, C57BL/6J wilde type mice were purchased from TACONIC and maintained in specific pathogen free conditions at Karolinska Institutet (Sweden). Cyp27a1 line, Villin-Cre Yap1f/fTazf/f, Villin-Cre LXRαf/fβf/f line and Lgr5-eGFPIRES-creERT2 line mice were obtained as described in method section and maintained under specific pathogen free conditions at Karolinska Institutet (Sweden). Additionally, some experiments with Cyp27a1-/- and littermate controls where some mice were housed in an MPV-positive animal facility (the experiments with DSS administration and half of the irradiation experiments performed on Cyp27a1-/- and littermate controls. ApcMin/+ line mice were maintained and being used in experiments in Hamburg-Eppendorf University (Germany), Areg mice line were maintained and being used in experiments in Weizmann Institute of Science (Israel), all were bred and housed under specific pathogen free conditions.<br>Wild type or floxed control littermates were used as controls as indicated. |
| Wild animals | No wild animals were used in the study |
| Field-collected samples | Study did not involve field collected samples |
| Ethics oversight | Experiments using laboratory animals were approved and carried out in accordance to local the guidelines.<br>Experiments conducted at Karolinska Institutet (Sweden) were approved by Stockholm Regional Ethics Committee.<br><br>Experiments conducted in Hamburg-Eppendorf University (Germany) were approved by, the Institutional Review Board "Behörde für Justiz und Verbraucherschutz, Lebensmittelsicherheit und Veterinärwesen" (Hamburg, Germany).<br><br>Experiments conducted in Weizmann Institute of Science (Israel) were approved by the institutional guidelines for animal care and all experimental protocols were approved by the Institutional Animal Care and Use Committee (IACUC) of the Weizmann Institute. |

Note that full information on the approval of the study protocol must also be provided in the manuscript.

# Human research participants

Policy information about studies involving human research participants

| | |
|---|---|
| Population characteristics | For human samples, samples were collected from healthy control (n= 28), ulcerative colitis (UC) active (n=39) and UC remission (n=27) from multiple sites such as terminal ileum, ascending colon and sigma/rectum from patients with IBD or suspicion of intestinal disease..<br><br>For human CRC samples, patients undergoing surgery for tumor resection had to be older than 18 years old and not have received chemotherapy or neoadjuvant therapy prior to total or partial colectomy. Tumor staging was classified according to the TNM classification (The Union for International Cancer Control; UICC). |
| Recruitment | For human samples, paired endoscopic biopsy specimens were obtained from the terminal ileum, ascending colon and sigma/rectum from patients with IBD or suspicion of intestinal disease.<br><br>For human CRC samples, samples from CRC patients from Colorectology Department, Clínica Las Condes, were included between 2015 and 2017. All patients signed informed consent forms approved by the institution (Cómite de ética de la investigación de CLC, O22019AA) and procedures were performed according to human experimental and clinical guidelines.<br><br>Potential biases include the recruitment source bias, where samples were gathered from specialized medical departments (from Germany and Chile), which may not represent the general population. Temporal bias could arise due to the specific period of sample collection from 2015 to 2017, impacting the applicability of results to other timeframes. Despite these biases, transparent reporting and comparative analyses are employed to mitigate their impact. |
| Ethics oversight | For human samples, the study was approved by the local ethical committee (EthikKommission der Ärztekammer Hamburg PV4444).<br><br>For human CRC samples, the study was approved by the institution (Cómite de ética de la investigación de CLC, O22019AA) |

Note that full information on the approval of the study protocol must also be provided in the manuscript.

# Flow Cytometry

## Plots

Confirm that:

☒ The axis labels state the marker and fluorochrome used (e.g. CD4-FITC).

☒ The axis scales are clearly visible. Include numbers along axes only for bottom left plot of group (a 'group' is an analysis of identical markers).

☒ All plots are contour plots with outliers or pseudocolor plots.

☒ A numerical value for number of cells or percentage (with statistics) is provided.

## Methodology

| | |
|---|---|
| Sample preparation | For sorting of mouse primary intestinal epithelial cells, cells were first isolated by using EDTA treatment of minced tissues followed by mechanical dissociation and enzymatic digestion with TrpLE express to obtain single cell suspensions. |
| | For sorting cells from organoids (for 10X scRNAseq), SI organoids on day5 were removed from the matrigel, cleaned with cold PBS and then digested with TrpLE Express to obtain the single cell suspensions. |
| | For sorting intestinal epithelial, immune and double negative cells from mouse SI from steady state and irradiated mice, dissected small intestines were flushed with cold PBS and cut in to ~ 1cm pieces. To isolate the epithelial cells (IECs) and the intraepithelial lymphocytes (IELs), intestinal pieces were incubated in HBSS supplemented with 5% FCS, 15mM HEPES, 5mM EDTA and 1mM DTT for 30 minutes at 37°C under agitation at 800 rpm. The supernatant (called IEL fraction) containing IECs and IELs were filtered through 100µm cell strainers, centrifuged at 500x g for 5 minutes at 4°C and was enriched using 44%/67% Percoll gradient and washed for FACS analysis. To further isolate the cells from the lamina propria, the intestinal pieces were washed with PBS supplemented with 5% FCS and 1mM EDTA for 15 minutes at 37°C under agitation at 800 rpm. Next, the tissue pieces were digested in HBSS supplemented with 0.15mg/ml Liberase TL (Roche) and 0.1mg/ml DNaseI (Roche) at 37°C for 45 minutes under agitation at 800 rpm. The digested tissues were filtered through 100µm cell strainers and were washed for FACS analysis (called LP fraction). |
| Instrument | Cells were sorted either using FACS ARIA II (BD) or SONY SH800S cell sorter. |
| Software | FACS Diva and SH800S software. Cells were analyzed using FlowJo v10 |
| Cell population abundance | The purity of sorted populations was dertermined by flow cytometry anaysis of sorted cells and frequencies for the gated population were above 90%. Sorted of intestinal epithelial cells from organoid culture was ~95-99% pure as it was obtained after 5 days of organoid culture. Purity of ex-vivo sorted intestinal epithelial cells, immune cells and double negative cells were ~90-95% pure based on the purity obtained from the FACS machine. |
| Gating strategy | Cells were gated using FSC-A, SSC-A followed by FSC-A and FSC-H to gate on the single cells. Gating strategy are provided in respective figures (Extended Data Figs. 2a, 2c, 9f) and in Supplementary Figs. 1 and 2. |

☒ Tick this box to confirm that a figure exemplifying the gating strategy is provided in the Supplementary Information.

