## [Peer Review File · Nature]

Manuscript Title: Liver X receptor unlinks intestinal regeneration and tumorigenesis

Reviewer Comments & Author Rebuttals

Reviewer Reports on the Initial Version:

Referee #1 (Remarks to the Author):

The manuscript reports on the role of LXR activation in the intestinal epithelium during repair following damage in both the small intestine (irradiation) and the colon (DSS administration). The authors reveal that a number of genes associated with cholesterol metabolism regulated by the nuclear receptor LXR are elevated during regeneration. They go on to show that administration of the LXR agonist GW3965 in the diet during damage enhances regeneration and reduces the severity of damage. This is associated with elevated levels of proliferating cells in the regenerating areas 5 days post injury. This is however not associated with changes in the expression of inflammatory markers suggesting that the effect of GW3965 is not linked with general changes in inflammation. Using organoid cultures, the authors show that GW3965 induces small increases in the level of budding and ascribe this to EGFR signaling. Based on single cell expression analysis the authors propose that Areg produced in the epithelium is responsible for the observed effect. In line with the observations that GW3965 enhance repair deletion of CYP27A1 impairs/stalls tissue regeneration, and show that levels of cholesterol response genes are reduced indicating that this is the key gene responsible for oxysterols production and cholesterol signaling. Interestingly, treatment of animals with GW3965 reduces tumor development, whereas CYP27A1 KO animal exhibit a slight increase in tumour development.

Overall, this is a very interesting and well written manuscript providing new insights into mechanisms controlling tissue regeneration. These are extremely timely findings, but a number of major concerns remain affecting the strength of the conclusions.

Major concerns:

It remains speculations that GW3965 is not affecting the immune system. The authors assess tissue 2 weeks after DSS administration. Yet, the manuscript is completely lacking analysis of the tissue including stroma following GW3965 administration. Furthermore, the authors argue that GW3965/LXR stimulation provides an adaptive mechanism fueling tissue regeneration, however, the drug is administered during injury, and it is unclear how the cellular steady state is affected by the treatment. The authors need to provide a detailed molecular characterization of how GW3965 affects the steady state (both epithelium and stroma), as it remains a possibility that the observed changes in regeneration is not cell autonomous to the epithelium, and also that the administration of GW3965 can alter the steady state and therefore the response to damage rather than regeneration. This should also be accompanied by additional analysis beyond proliferation and apoptosis during tissue regeneration, as evidence that the increase in proliferation is the reason for enhanced regeneration. Are more cells capable of contributing or are cells dividing more rapidly? Furthermore, do the authors also observe a beneficial effect if GW3965 is administered following the induction of injury?

The link between EGFR activation downstream of GW3965 needs to be substantiated. All the evidence is so far circumstantial. The authors argue that elevated levels of endogenous expression of Areg promotes the growth promoting effects of GW3965. However, the pictures included in figure 2 are clearly not representative (based on the data in the graph), and it will be appropriate to assess this using a genetic approach. Also, although internalisation of EGFR is linked with active signalling, there appears to be ligand dependent effects with Areg being a low affinity ligand. To assess pathway activation the authors need to assess levels of active receptor (phosphorylation levels). Moreover, are there additional differences upon stimulation with ligand e.g. growth rates, differentiation markers?

The authors argue for an autonomous effect in the epithelium upon GW3965, and perform a scRNAseq experiment to identify the source. However, it remains unclear why this experiment is performed in vitro and not in vivo. This would also enhance the strength of the conclusions related to cluster formation during regeneration, which is currently rather artificial given that these are based on organoids. Moreover, the authors argue based on the scRNAseq studies that Areg is upregulated by reserve stem cell population, this should be substantiated by in vivo studies e.g. colocalization studies of e.g. Clu, Ly6a and Areg (e.g. Ayyaz et al., 2019, Nature; Yui et al., 2018, Cell Stem Cell). The latter could be done using e.g. RNAscope analysis.

LXR mutant animals should be further characterized. The text currently states that they “did not display any significant phenotypic alteration at steady state (data not shown).” What does this mean? The authors provide 16 extended data figures and 4 main figures and leave no place for the characterization of the LXR mutant animals, and to what degree it has been analysed.

If CYP27a1 is the enzyme responsible for activation of LXR signalling, the authors should be able to rescue the phenotype by treatment with GW3965 to WT levels.

It is currently unclear how the authors in the last part of the manuscript can distinguish between the level of damage and inflammation, which has been demonstrated to drive tumor development, and the proposed PPAR signaling. The authors will need to provide experimental evidence to support that the proposed model dependent on PPAR signaling is responsible for the observed effect. Currently, it remains unclear how the tumor development data links with the rest of the manuscript.

Additional concerns:

Can organoids grown in NR supplemented with GW3965 be passaged as efficiently as cell cultured in ENR?

Extended data figure 3: Refers to ligand expression at the RNA level as EGFR signalling, however, this is not synonymous with EGFR signaling, but rather EGFR or rather ErbB ligand expression.

The data in extended data 7 is not very convincing. It is unclear how this is normalized. Moreover, how much variation is observed between experiments.

Referee #2 (Remarks to the Author):

In the manuscript Das, Parigi and coworkers explored how LXR activation impacts ISC proliferation and tumorigenesis. The authors propose Areg as the responsible mediator of improved regeneration after damage observed upon LXR activation. The authors suggest that LXR is induced during regeneration thanks to upregulation of Cyp27a1, which produces oxysterols that activate LXR. Deletion of Cyp27a1 led to more severe DSS induced colitis. Also, the authors report that LXR activation is protective against intestinal tumorigenesis, and they claim this is the result of PPAR α activation. They report that Cyp27a1^{-/-} mice showed higher tumorigenesis, and human tumors had lower level of Cyp27a1^{-/-}.

Although some of the findings are potentially interesting, many important aspects of intestinal metabolism and physiology have not been considered in the analysis. The reported results raise many questions, but unfortunately none of these have been rigorously addressed in the present work.

Major comments

- 1) What happens in terms of proliferation of ISCs in response to LXR activation under basal conditions (not during damage recovery)?
- 2) The authors claim that Abca1 and Abcg1 expression is increased after irradiation and DSS in response to LXR. However, they did not use intestinal specific LXR knock-out mice to show that Abca1 and Abcg1 expression is decreased compared to control after irradiation and DSS. Abca1 and Abcg1 could be altered independent of LXR as these genes are known to be regulated by other factors.
- 3) It appears that LXR agonism is protective after irradiation and DSS; however, some of the results are difficult to understand. GW3965 fed mice have much less immune cell infiltration in the lamina propria after DSS. It is known that LXR signaling is abundant in macrophages, which could explain this finding. The paper does not determine whether the normal crypt-villus architecture is due to LXR's effect on suppressing inflammation. On the other hand, pro-inflammatory cytokine expression is not different in DSS-induced colitis on GW3965 diet, which indicates that LXR treated mice have just as much ongoing inflammation than the standard diet. This data on the cytokines does not match the absence of inflammation on histology in the LXR treated mice.
- 4) Many of these findings appear to be unrelated. The authors do not do a good job of tying the findings together and providing a clear mechanism to understand them. For example, the authors state that inhibition of EGFR abolishes the pro-regenerative effect of LXR activation in organoids, but does LXR inhibition change EGFR levels? In extended data Fig 3h, there is no change in EGFR with LXR treatment. The authors also do not show that Cyp27A1 deficiency leads to a decrease in PPAR α signaling in the setting of increased tumor size.
- 5) Cholesterol metabolism has been shown to be important for intestinal proliferation but the authors do not appear to have taken this into consideration. It is important to address whether known mechanisms linking lipid metabolism to proliferation could be relevant to the authors findings.

6) The authors focused solely on Areg to explain the improved intestinal regeneration promoted by LXR activation. The data demonstrate that GW3965 has an effect on EGFR signaling. However, is hard to believe that all the metabolic remodeling happening in response to LXR activation plays no role, especially given the extensive literature on links between metabolism and proliferation. The fact that the authors identify cholesterol metabolism genes as differentially regulated during regeneration (Fig 1A) further suggests cholesterol metabolism and homeostasis play an important role. In the authors' investigation of the role of LXR activation in tumorigenesis, their RNAseq data suggested modulation of cholesterol metabolism and PPAR signaling, yet the authors decided to only investigate PPAR targets.

7) It is known that Cyp27a1^{-/-} mice have decreased formation of bile acids and excretion of fecal bile acids, so it is possible that this impacts cholesterol absorption in intestine. Have the authors considered this might play a role in the observed phenotype?

8) The authors should test what happens if Cyp27a1^{-/-} mice are treated with PPAR α agonist. Is this protective? This will support the existence of a Cyp27a1-LXR-PPAR link, which is otherwise not demonstrated by current results.

Referee #3 (Remarks to the Author):

In the manuscript Das et al. the authors describe how LXR signaling regulates intestinal regeneration and tumorigenesis. By using different injury models (i.e., irradiation and dextran sodium sulfate (DSS)) on intestinal organoids. This setup in combination with scRNA seq generated different transcriptomic landscapes. Based on their analysis, several genes (i.e., Abca1, Apoc3, Apoa1, and APOA4) are regulated by LXR. To assess whether LXR increases regeneration, Das and colleagues supplement an LXR agonist in the diets of mice. Both in vivo and in vitro studies demonstrate increased regeneration, which is driven by the intestinal niche cells. The regenerative response is mainly driven in an EGFR-pathway dependent manner, amphiregulin (Areg) seems responsible for this effect in intestinal stem cells.

In addition, the authors found that the LXR ligand-producing enzyme CYP27A1 was upregulated in the niche of the damaged intestinal crypts. Here, perturbations studies demonstrated either an increase or decrease of tumorigenesis upon LXR activation or inhibition, respectively.

All together the authors hypothesize that LXR activation could function as a rheostat. Where LXR regulates regeneration and preserves homeostasis and prevents neoplastic transformation. Further evidence comes from LXR activation in two different mouse tumor models where the activation results in restrained cancer progression.

In general, the manuscript is written well, but some parts miss depth and clearer explanation, moreover it is not really tailored for a broad audience and the structure is not always linear. Moreover, they have not been using state-of-the-art techniques (see Major points) to present better how significant their study and hypothesis is. The use of the different organs (small intestine and colon) and the different injury methods is unclear (DSS and irradiation). We recommend paying some extra attention in the text to why you use the different strategies in addition to our major comments. Finally the conclusion on the role of LXR as rheostat are not really justified by the data.

Major points:

- Could you untangle Figure 1A even more:

DSS irradiation

small intestine

colon

Figure 1A could serve as supplemental. Will the authors find back the same results by using the representation? Here, it stresses the difference between the two organs small intestine and colon. Especially, in the mouse intestinal track there is regional specificity, which is rather important (Beumer et al., 2020 Cell and Gebert et al., 2019 Cell Reports).

In addition, irradiation and DSS treatment are different injury methods and result in a different response. In general, it would be important to study the same parts of the mouse, even though DSS is mainly affecting the mouse colon. Nevertheless, the authors could make use of the ileal part of the DSS treated mouse and compare it to the ileal part of the whole-body irradiated mouse.

This will make their findings more concise and maybe they will also find a more profound effect for a specific region of the intestine. Also did the author have similar results in colon organoids? At this moment the colon data in the manuscript is relatively little. Probably more in vitro experiments and

analysis on the colon would be necessary.

- Ayyaz et al., 2019 demonstrated the role of the reverse stem (RS) cell after damage and the signaling cascade initiated by YAP to repopulate the crypt niche. By perturbing or activating YAP, AREG and EREG should also go down or up (Sera et al., 2019 Nature). First it is important for the author to describe properly the literature on intestinal regeneration and the role of YAP. Second some staining and perturbation of YAP would be quite important. Third checking the RS stem cells and comparing the scRNA clusters also with other data would be important. Finally, the authors should elaborate on how YAP, CYP27A1, and AREG behave on a transcriptomic timeline during organoid development from stem a non-stem cells. And maybe compare by using different datasets combined with a time course for GW3965 (pre)treated organoids.

- o Thinking about this transcriptomic timeline could the authors explain/comment on why in Figure 1C and 3C the levels of *Abca1* and *Cyp27a1* are not going back to a homeostatic level after stopping initial damage. We see a downwards trend in Figure 3C but it takes rather long and for Figure 1C there is constant high level.

- o In line with the previous comment, we find it difficult to understand that both LXR-target genes of interest, *Abca1* and *Abcg1*, do not show any upregulation during induced damage. This happens without specifically activating LXR by feeding GW39656 in neither the small intestine nor the colon as shown in Extended Fig1? This contradicts heavily with Figure 1B-C.

- For the Figures 2b-d, 3e, 4i and Extended Data Figure 2b, 7c-j it would be nice to see spatial transcriptomics. Even in vivo spatial transcriptomics would make things clearer in their natural microenvironment for the different timepoints.

- Organoid quantifications are not powerful. Think in terms of seeding density and Matrigel-dome (i.e., volume). The average per mouse is not specific enough.

Therefore, one could think of whole well overviews and organoid count per well. Crypt domain/organoid is still a good measure but should be done at the last step of the previous quantifications. Maybe also think about a dead-cell marker.

- In the lines 62-63 is the hypothesis about different tissues. This is highlighted for the intestinal track and mainly small intestine. However, it would make this study broader by demonstrating that LXR signaling regulates also tissue regeneration in for example liver, lung, or skin.

- The scRNAseq clusters are really not clear and not interpretable in line of already published datasets. For example cluster 0 has enterocytes and regenerative cells? And the author then mention "potential involvement in intestinal regeneration and enterocyte differentiation". This part is not clear and having the scRNA sequencing in the middle also disrupt the flow as most of the other data is qPCR.

- In figure 1 the authors only look at few genes (*Areg*, *Ereg*, *EGFR*). Having the scRNAseq why the authors don't start with a more LXR comprehensive analysis?

- Did the authors try to add *Areg* or *Ereg* to the organoids or the diet?

Minor points

- The bridge towards the human data is promising. It would be even more powerful if one could show, in addition to the expression data in Figure 3m and n, an example of a UCs staining with these markers (e.g., protein and spatial transcriptomics).

- Is there an explanation for the cytoskeletal remodeling in Extended data Fig 14.d?

- Figure 4j where were the non-tumor (NT) biopsies taken?

- When comparing the overlap between the two previously published scRNAseq datasets, general fat digestion and absorption comes up as the top hit of DEG instead of the LXR-regulated cholesterol metabolism. Why did the authors choose to not follow these hits and instead focus on the LXR pathway?
- In figure 3c, Cyp27a1 seems to get upregulated only after the DSS treatment stops (day 8). This is contrary to the previous result, where LXR target gene Abca1 already starts to be upregulated after 5 days of DSS treatment (Fig 1c) even though it should be downstream of Cyp27a1. Can the authors comment on this delay?
- There seems to be a problem with Figures hiding text. Line 79, 111, 195 and 221 seem to be hidden by the text.
- Ext. Data Fig 10 and 14 have a low resolution.

Author Rebuttals to Initial comments:

We sincerely thank the reviewers for their critical feedback and the challenges posed by the review process, we have rigorously addressed the reviewers' comments, and we believe that the resulting manuscript is significantly improved and provides a significant contribution to our understanding of tissue regeneration and tumorigenesis. Please find below a detailed point-by-point response to the reviewers' comments.

Point-by-point response to the Reviewers

Referee #1 (Remarks to the Author):

The manuscript reports on the role of LXR activation in the intestinal epithelium during repair following damage in both the small intestine (irradiation) and the colon (DSS administration). The authors reveal that a number of genes associated with cholesterol metabolism regulated by the nuclear receptor LXR are elevated during regeneration. They go on to show that administration of the LXR agonist GW3965 in the diet during damage enhances regeneration and reduces the severity of damage. This is associated with elevated levels of proliferating cells in the regenerating areas 5 days post injury. This is however not associated with changes in the expression of inflammatory markers suggesting that the effect of GW3965 is not linked with general changes in inflammation. Using organoid cultures, the authors show that GW3965 induces small increases in the level of budding and ascribe this to EGFR signaling. Based on single cell expression analysis the authors propose that Areg produced in the epithelium is responsible for the observed effect. In line with the observations that GW3965 enhance repair deletion of CYP27A1 impairs/stalls tissue regeneration, and show that levels of cholesterol response genes are reduced indicating that this is the key gene responsible for oxysterols production and cholesterol signaling. Interestingly, treatment of animals with GW3965 reduces tumor development, whereas CYP27A1 KO animal exhibit a slight increase in tumour development.

Overall, this is a very interesting and well written manuscript providing new insights into mechanisms controlling tissue regeneration. These are extremely timely findings, but a number of major concerns remain affecting the strength of the conclusions.

Authors' response: We are very pleased to read these laudatory remarks and we highly value the concerns raised by the reviewer, as in addressing those we have improved the quality of our study.

Major concerns:

It remains speculations that GW3965 is not affecting the immune system. The authors assess tissue 2 weeks after DSS administration. Yet, the manuscript is completely lacking analysis of the tissue including stroma following GW3965 administration. Furthermore, the authors argue that GW3965/LXR stimulation provides an adaptive mechanism fueling tissue regeneration, however, the drug is administered during injury, and it is unclear how the cellular steady state is affected by the treatment. The authors need to provide a detailed molecular characterization of how GW3965 affects the steady state (both epithelium and stroma), as it remains a possibility that the observed changes in regeneration is not cell autonomous to the epithelium, and also that the administration of GW3965 can alter the steady state and therefore the response to damage rather than regeneration. This should also be accompanied by additional analysis beyond proliferation and apoptosis during tissue regeneration, as evidence that the increase in proliferation is the reason for enhanced regeneration. Are more cells capable of contributing or are cells dividing more rapidly? Furthermore, do the

authors also observe a beneficial effect if GW3965 is administered following the induction of injury?

Authors' response: The reviewer raises several key points:

- 1) **“LXR stimulation fuels regeneration by a non-cell autonomous mechanism to the epithelium”**. This concern is very well-founded, as LXR is virtually expressed in every cell of our body and has been reported by us and by others to play a key immunomodulatory role in several immune cell types. As a result, systemic administration of GW3965 might impact cells other than the epithelium. We therefore agree with the Reviewer that a thorough characterization of how cellular compartments are modified upon GW3965 administration is needed. Toward this, we performed scRNA-seq of epithelial and non-epithelial cells and spatial transcriptomics (visium, 10x genomics) of small intestinal tissues after 10 days exposure or not with GW3965 diet. This unbiased analysis showed comparable cell clusters (epithelial, stromal, and immune cells) and gene programs within the small intestine from GW3965 and standard diet-fed animals (Extended Data Fig 2d-g). Given the role of LXR in regulating inflammatory immune responses, we unbiasedly mapped “inflammatory response” genes (selected based on the Gene Ontology repository) to the immune cell clusters. Of note, despite efficient LXR activation (as seen by the induction of LXR target genes) (Extended Data Fig 2h), we did not record significant changes between standard and GW3965 diet-fed mice (Extended Data Fig 2i), suggesting no alteration in immune reactivity upon 10 days of diet. To assess the role of LXR activation more thoroughly in immune cells we crossed Vav1-Cre with LXR $\alpha\beta^{\text{flox/flox}}$ animals (to generate mice conditionally deficient of LXR in all immune cells). Unfortunately, we detected germline LXR deletion, resulting in whole-body LXR deficiency, including stromal and epithelial cells. We therefore decided not to use this mouse line for experiments as it was not reliable. Altogether, although these findings do not entirely exclude a potential contribution of LXR activation in immune cells to bolster regeneration, our data using organoids *in vitro* and Villin-Cre driven conditional KO *in vivo* show that LXR activation intrinsically in epithelial cells is sufficient to drive a pro-regenerative effect in the intestine in response to damage.
- 2) **“LXR stimulation might alter the epithelium already in steady state, thus preconditioning the organ to a better recovery before exposure to injury”**. We fully agree that, with our experimental protocol (i.e., 10 days of GW3965-diet administration followed by irradiation or concomitant GW3965 diet and DSS administration), we are not activating LXR simply during regeneration but also before/during injury. Indeed, SI tissue after 10 days of GW3965 diet showed significantly increased villus length (and to a minor extent crypt height) (Extended Data Fig 3a). This suggests that 10 days of preemptive LXR activation seems to change epithelial homeostasis and potentially precondition the tissue to better recovery after injury. While these data provide new relevant insights into changes in intestinal epithelium upon prophylactic LXR activation, they do not contradict our model proposing physiological LXR activation leading to a stronger pro-regenerative response that surfaces only upon injury. Support for our model stems from lack of a proliferative (i.e., BrdU incorporation) advantage in epithelial cells from mice fed with GW3965 diet compared to standard diet in the absence of injury (Extended Data Fig. 3b-c) compared to the epithelium following injury (Fig. 1g). Moreover, when SI crypts were extracted (after 10 days of GW3965- or standard-diet) to generate organoids (mimicking an injury-regenerative response *in vitro*) in the absence of any further GW3965 stimulation *in vitro*, GW3965 pre-

conditioned crypts showed increased budding compared to control counterparts (Extended Data Fig. 3d). These new data contribute to draw a more accurate model of the pro-regenerative mode of action of LXR activation in the intestinal epithelium and we thank the reviewer for their comment that led us to this new insight.

- 3) **“LXR should be activated following injury to fully understand its role during regeneration”**. This concern is very relevant as it paves the way to a potential therapeutic application of LXR stimulation. We have noticed that mice lose appetite and weight following whole body irradiation and therefore makes it difficult to administer the diet during the short length of irradiation experiments. Therefore, to address the reviewer comment, we instead administered standard or GW3965-diet_following DSS treatment (i.e., from d6 to d14, regenerative phase). Of note, this “therapeutic” setting proved to be beneficial as seen by increased crypt cell proliferation and colon length at day 14 in the LXR agonist-treated group. These new results not only hint at a potential therapeutic application of LXR agonist, but also strengthen the concept that LXR activation fuels regeneration rather than dampening damage. These new data are now included in Fig. 1h-i.

The link between EGFR activation downstream of GW3965 needs to be substantiated. All the evidence is so far circumstantial. The authors argue that elevated levels of endogenous expression of Areg promotes the growth promoting effects of GW3965. However, the pictures included in figure 2 are clearly not representative (based on the data in the graph), and it will be appropriate to assess this using a genetic approach. Also, although internalisation of EGFR is linked with active signaling, there appears to be ligand dependent effects with Areg being a low affinity ligand. To assess pathway activation the authors need to assess levels of active receptor (phosphorylation levels). Moreover, are there additional differences upon stimulation with ligand e.g. growth rates, differentiation markers?

Authors’ response: we agree with the reviewer that our data on EGFR activation downstream LXR activation should be further substantiated. Towards this, we performed and included in the revised manuscript the following experiments:

- 1) As suggested by the reviewer, we used a genetic approach to uphold our conclusions on Areg mediating LXR-dependent pro-regenerative effect. Organoids from Areg^{-/-} mice failed to display a LXR-induced regenerative advantage (i.e. increased crypt domain formation) compared to their WT littermates. These new data confirm our previous observations using anti-AREG blocking antibodies and substantiate the notion of epithelial amphiregulin as a critical pro-regenerative mediator downstream LXR (see new Fig. 2j)
- 2) To assess EGFR pathway activation, we stimulated organoids with DMSO or GW3965 and collected them at day 5 to analyze EGFR pathway activation by Western Blot. Despite comparable total EGFR and pEGFR (pEGFR^{Y1068}) levels, (Fig 1a for Reviewer 1), we observed enhanced phospho-ERK (a known downstream mediator of EGFR signaling pathway) in GW3965-treated compared to DMSO-treated organoids (Extended Data Fig. 4i). Of note, such effect was observed only when organoids were cultured in NR media, as addition of EGF (i.e., ENR media) compensated the difference between DMSO and GW3965 and led to an overall downregulation of phospho-ERK (Extended Data Fig. 4i).
- 3) To identify additional differences upon stimulation with LXR agonist and different EGFR ligands, we treated organoids with DMSO, GW3965, EGF (200ng/mL),

and AREG (50ng/mL) and assessed crypt domain formation and differentiation markers by qPCR. GW3965, EGF or AREG addition significantly increased organoids budding efficiency (as seen by the overall increase in organoids with more than 4 crypt domain) albeit LXR activation promoted it to a lesser extent compared to AREG and EGF. In line with the enhanced organoids growth, assessment of differentiation marker by qPCR showed that AREG or EGF stimulation proportionally increased the abundance of *Alpi* transcript (i.e., differentiated enterocytes) over stem or Paneth cell specific transcripts (usually more abundant in more immature or non-differentiated organoid cultures) (please see Fig 1b-d for Reviewer 1). Thus, although LXR activation favored organoid growth less efficiently than supraphysiological stimulation with EGFR ligands (EGF and AREG), all these ligands induced a similar epithelial differentiation pattern compared to DMSO-treated controls.

- 4) To improve our analysis of regenerative potential in organoids (and have more representative pictures), we are now providing a stratified quantification showing proportions of organoids with 0, 1, 2, 3 or ≥ 4 crypt domain/organoid (see Figs. 1 and 2 and all Extended Data Figs with organoids data).

Fig.1 for Reviewer-1

Fig. 1 for Reviewer 1: EGFR signaling and impact of EGFR ligands in intestinal organoids in the context of LXR activation. (a) Western blot quantification of total EGFR (normalized to vinculin) and pEGFR^{Y1068} (ratio of pEGFR/tEGFR after each normalized to their loading control vinculin) from mouse SI organoids cultured for 5 days with DMSO or GW3965 in NR medium. (b) qPCR analysis of *Abca1* expression in mouse SI organoids treated with DMSO, GW3965, EGF (200ng/mL) or AREG (50ng/mL) and cultured for 5 days in NR medium. (c) Quantification of the % of organoids with the indicated number of buds from SI organoids cultured for 5 days with DMSO, GW3965, EGF (200ng/mL) or AREG (50ng/mL) in NR medium. (d) qPCR analysis of IEC lineage markers (PC = Paneth Cell, CBC = crypt base columnar cells, GC = Goblet cell, EEC = Enteroendocrine cell, EC = Enterocyte) in mouse SI organoids treated with DMSO, GW3965, EGF (200ng/mL) or AREG (50ng/mL) and cultured for 5 days in NR medium. Data are representative of two (b-d) or three (a) independent experiments with 2-3 mice/group. Each dot represents one mouse. Significance was assessed by paired t-test (a). ns = not significant.

The authors argue for an autonomous effect in the epithelium upon GW3965, and perform a scRNAseq experiment to identify the source. However, it remains unclear why this experiment is performed in vitro and not in vivo. This would also enhance the strength of the conclusions related to cluster formation during regeneration, which is currently rather artificial given that these are based on organoids. Moreover, the authors argue based on the scRNAseq studies that *Areg* is upregulated by reserve stem cell population, this should be substantiated by in vivo studies e.g. colocalization studies of

e.g. Clu, Ly6a and Areg (e.g. Ayyaz et al., 2019, Nature; Yui et al., 2018, Cell Stem Cell). The latter could be done using e.g. RNAscope analysis.

Authors' response: We agree with the reviewer that performing scRNAseq on *in vitro* cultured organoids might lead to artificial conclusions. Our initial decision to perform scRNAseq on organoids was justified by the following rationale: having observed that LXR activation in IEC was sufficient to promote regeneration *in vivo*, we opted for a simplistic approach (i.e. organoids) to first validate the *in vivo* pro-regenerative effect and then identify a potential *direct and cell-autonomous/intrinsic* mechanism driving it. Nonetheless, the reviewer is right when stating that the conclusions were not fully backed by the data presented in the initial version of the manuscript. To this end, we have now included/modified the following:

- 1- The discussion of the clusters identified by scRNAseq in organoids has been substantially reduced, as drawing conclusions on reserve stem cell population based on *in vitro* findings could be misleading.
- 2- We still exploit the scRNAseq performed in organoids to guide us in identifying how direct LXR activation in IEC can drive regeneration in a cell autonomous manner. We believe that this system can, in some instances, provide a less noisy window to pinpoint relevant biological mechanisms that should nonetheless be validated *in vivo*. Towards this, after identifying *Areg* as a potential LXR-dependent pro-regenerative mediator, we:
 - a. Validated upregulation of *Areg* in sorted epithelial cells from small intestine of wild-type mice at 0-1-3 dpi and confirmed *Areg* upregulation in IEC (please see Fig 2 for Reviewer 1).
 - b. We attempted performing scRNAseq on IEC from irradiated STD or GW3965-fed mice. However, we did not succeed in obtaining good quality sequencing data (possibly due to the high degree of cell death upon irradiation). Therefore, we endeavored in a parallel approach and performed Visium spatial transcriptomics (ST) at 0 and 3 dpi in STD and GW3965 fed mice. Using this unbiased dataset, we confirmed *Areg* upregulation in the tissue in response to damage and LXR activation (Extended Data Fig.6 a-d). Of note, among the ST spots where *Areg* expression was detected, the spots in cluster 4 and 5 showed the highest and most significant upregulation in GW3965 diet- compared to STD diet-fed mice at 3 dpi. When mapping the ST cluster 4 and 5 on the tissue, we observed that they were localized at the base of the crypt, further corroborating our model of *Areg* upregulation in epithelial cells in the crypt region upon LXR activation (Extended data Fig. 6a-d).

Besides providing this new experimental evidence, we agree with the reviewer that the accuracy of our conclusions should be carefully assessed. While we do not exclude other potential pro-repair roles downstream LXR engagement in non-IEC cells, our data show a novel pathway of host adaptation to damage driven by LXR activation in IEC.

Fig.2 for Reviewer-1

Fig.2 for reviewer-1: Amphiregulin (Areg) expression is induced in intestinal epithelial cells following damage. qPCR analysis of *Areg* in FACS sorted EpCAM⁺ IECs from WT mouse SI collected at indicated days post-irradiation (dpi). Data are representative of two independent experiments with 4-6 mice/condition (each dot represents one mouse). Data are shown as mean ± SD and significance was assessed by one-way ANOVA with Tukey's post-test. **p<0.01

LXR mutant animals should be further characterized. The text currently states that they “did not display any significant phenotypic alteration at steady state (data not shown).” What does this mean? The authors provide 16 extended data figures and 4 main figures and leave no place for the characterization of the LXR mutant animals, and to what degree it has been analysed.

Authors’ response: We have now included the new Extended Data Fig. 3f-j, where we quantified SI and colon length, SI crypt and villus height and SI crypts BrdU+ cells (after a 2-hour pulse) in Villin-Cre:LXRab^{flox/flox} mice and littermate controls in steady state. These new data confirmed that LXR deficiency does not lead to significant phenotypic alterations in the intestinal epithelium in the absence of damage.

If CYP27a1 is the enzyme responsible for activation of LXR signaling, the authors should be able to rescue the phenotype by treatment with GW3965 to WT levels.

Authors’ response: We agree with the reviewer that the suggested experiment is critical to prove the CYP27A1-LXR axis. Our new data with *Cyp27a1*^{-/-} mice fed GW3965 diet in the context of irradiation-induced damage and tumorigenesis showed indeed that LXR activation was able to rescue the phenotype to WT levels (i.e., favoring regeneration and dampening tumorigenesis compared to standard diet-fed *Cyp27a1*^{-/-} mice). These new data have significantly improved our manuscript and are now included in Figure 3i and 4b.

It is currently unclear how the authors in the last part of the manuscript can distinguish between the level of damage and inflammation, which has been demonstrated to drive tumor development, and the proposed PPAR signaling. The authors will need to provide experimental evidence to support that the proposed model dependent on PPAR signaling is responsible for the observed effect. Currently, it remains unclear how the tumor development data links with the rest of the manuscript.

Authors’ response: We agree with the reviewer that the initially proposed model dependent on PPAR signaling to restrain tumorigenesis was not fully demonstrated by our results but rather by concomitant evidence from previously published studies². In an attempt to demonstrate whether PPAR α acted downstream LXR to dampen tumorigenesis, we imported PPAR α deficient animals, fed them with STD or GW3965 diet and administered AOM-DSS. As shown by our preliminary experiment (please see Fig 3 for Reviewer 1), we did not observe enhanced tumorigenesis in PPAR α deficient animals compared to their littermate controls (which can be due to less number of mice, different animal housing conditions and thus a potential co-contribution of microbial exposure to PPAR α -dependent tumor susceptibility etc.), but more importantly, LXR activation equally restricted tumor burden in both control (*Ppar α* ^{+/+}) and *Ppar α* ^{-/-} animals. Thus, our initially proposed LXR-PPAR α anti-tumorigenic axis was not backed by the new experimental evidence (please see Fig 3 for Reviewer 1).

We, therefore, re-interrogated our unbiased kinetic transcriptomic dataset (Fig. 4c) and focused our attention to module 5 (“Hematopoietic cell lineage” and “Intestinal Immune network for IgA production”). Unlike other modules, module 5 started to diverge and was upregulated in GW3965-fed mice after d22 onwards (Fig. 4c), mirroring the

Fig.3 for Reviewer-1

Figure 3: PPARA does not mediate LXR induced anti-tumorigenic effects. WT mice were fed with standard or GW3965 diet and injected with AOM followed by 3 cycles of DSS. Mice were sacrificed at day 70. Bar plot shows the quantification of tumor numbers. Each dot represents one mouse.

difference observed in overall tumor growth between the two groups (**Extended Data Fig. 10j-k**), suggesting the pathways represented in module 5 by d22 might drive the phenotypic differences observed at a later time point. Moreover, module 5 showed no differences in mice fed or not with GW3965 diet in the absence of AOM-DSS (considered day 0 in the Figure), arguing that these pathways are engaged upon tumorigenesis. To corroborate these findings, we performed Visium spatial transcriptomics at d22 and d43 of AOM-DSS treatment in STD and GW3965-fed mice (**Extended Data Fig 10l**). While cluster analysis showed detectable differences at d43 but not d22 (**Extended Data Fig 10l**), deconvolution analysis showed the appearance of a B cell signature in colonic tissues from GW3965-fed mice at d22 which is virtually absent in STD-fed animals (**Fig. 4d**). By day 43, the B cell and plasma cell signatures were enhanced in GW3965- compared to STD-fed animals (**Fig. 4d**), suggesting that LXR activation during tumorigenesis expanded B cells within the tissue as early as d22, before any macroscopic tumor was evident. Of note, the B cell signature was distributed in the shape of clusters (**Fig. 4d**), resembling tertiary lymphoid structures (TLS) that have been associated with improved prognosis in colorectal cancer (CRC) patients³. Such expansion of TLS upon LXR activation during tumorigenesis was confirmed by immunohistochemistry staining at d70 (**Fig. 4e**). To functionally demonstrate whether B cell expansion mediated LXR-dependent anti-tumor effect, we treated STD- and GW3965-fed mice with CD19 blocking antibodies over the entire duration of the experiment (**Extended Data Fig. 10m**). Blocking CD19 significantly worsened tumor progression in GW3965-fed mice, albeit not to the level of STD-fed mice, suggesting that B cell expansion partially drives LXR-dependent anti-tumor effect (**Fig. 4f**). Finally, to address whether this axis may be present in human CRC patients, we mapped the expression of B cell signature genes in the microarray dataset originally presented in Fig. 4, which includes 6 tumor subtypes and non-tumor (NT) controls. Analogous to LXR target gene ABCG1 and CYP27A1 expression, B cell genes were downregulated in CRC patients relative to NT controls (**new Extended Data Fig. 10n**).

Additional concerns:

Can organoids grown in NR supplemented with GW3965 be passaged as efficiently as cell cultured in ENR?

Author's response: Yes. We have plated organoids in NR+GW3965 or in ENR (without additional stimulation) and passaged them for 6-7 times. Pictures were taken after every

passage and organoids morphology and number look similar between the two conditions (please see Fig4 for Reviewer 1).

Extended data figure 3: Refers to ligand expression at the RNA level as EGFR signalling, however, this is not synonymous with EGFR signaling, but rather EGFR or rather ErbB ligand expression.

Authors' response: we have now replaced “EGFR signaling” with “EGFR ligands” in all Figures.

The data in extended data 7 is not very convincing. It is unclear how this is normalized. Moreover, how much variation is observed between experiments.

Authors' response: we apologize for the lack of clarity. To quantify AREG expression in the crypt regions, crypts were outlined manually (using QuPath software) and, using the automatic quantification tool, the mean DAB intensity and the crypt area were quantified for each crypt. Therefore, the quantification of AREG in the crypt region was calculated as the mean DAB intensity for each crypt normalized to its area. All these values calculated for each individual crypt were then averaged to get an average value per mouse (corresponding to an individual dot in the plot presented in the original Extended Data 7, and now in the new Extended Data Fig 6e-f). To quantify AREG expression in the villus region, the crypts (manually outlined as described above) were selected out and using the cell detection tool, the mean DAB intensity and the area of each individual cell were calculated. The mean DAB intensity for each cell was normalized to its area and these values were then used to calculate the overall DAB intensity in the villus region per mouse. Therefore, in the original plot each dot represented one mouse and the average of all the crypts or villus regions analyzed. We have now included another plot where each dots represents an individual crypt or villus to provide a better representation of the variability present between experiments and within each mouse.

Fig. 4 for Reviewer-1

Figure 4: SI organoids grown in NR+GW3965 can be passaged as efficiently as organoids grown in ENR medium. Representative pictures of primary and secondary SI organoids (from passage 2, 4 and 6) from WT mice cultured in either ENR-DMSO or NR-GW3965 condition. 5X images taken on d0 shows the seeding density and the 10X images on d4 after every passage show the organoid morphology. Experiment was performed from 2WT mice.

Referee #2 (Remarks to the Author):

In the manuscript Das, Parigi and coworkers explored how LXR activation impacts ISC proliferation and tumorigenesis. The authors propose Areg as the responsible mediator of improved regeneration after damage observed upon LXR activation. The authors suggest that LXR is induced during regeneration thanks to upregulation of Cyp27a1, which produces oxysterols that activate LXR. Deletion of Cyp27a1 led to more severe DSS induced colitis. Also, the authors report that LXR activation is protective against intestinal tumorigenesis, and they claim this is the result of PPAR α activation. They report that Cyp27a1^{-/-} mice showed higher tumorigenesis, and human tumors had lower level of Cyp27a1^{-/-}.

Although some of the findings are potentially interesting, many important aspects of intestinal metabolism and physiology have not been considered in the analysis. The reported results raise many questions, but unfortunately none of these have been rigorously addressed in the present work.

Authors' response: We are pleased to read that the reviewer found our work conceptually impactful. We also agree that some aspects of cholesterol metabolism were not addressed by our manuscript in the original version. We, therefore, added new experimental evidence following the reviewer's suggestions, which significantly improved the quality of the manuscript and accuracy of our conclusions. Please find below our point-by-point response.

Major comments

1) What happens in terms of proliferation of ISCs in response to LXR activation under basal conditions (not during damage recovery)?

Authors' response: We thank the reviewer for raising this comment, shared by the other colleagues too. Our *in vivo* treatment regimen includes 10 days of modified diet administration before inducing intestinal damage and, in this time window, LXR activation might precondition the tissue to a better recovery. As suggested by the reviewer, we assessed small intestine (SI) crypt proliferation at 0 days post-irradiation (dpi) after 10 days of LXR activation by BrdU incorporation. Of note, we did not observe any statistically significant difference compared to standard diet (STD)-fed mice (see new Extended Data Fig. 3b). To more specifically assess intestinal stem cell (ISCs) proliferation, we stained the SI tissue (upon 10 days of modified diet) with wheat germ agglutinin (WGA, to identify Paneth cells at the crypt base), Olfm4 (stem cell marker) and BrdU (after a 2-hour pulse). *Bona fide* proliferating crypt base columnar cells (CBC) were defined as Olfm4⁺ positioned below the uppermost WGA⁺ cell in the SI crypt. Similar to the results described above, we did not record enhanced proliferation upon LXR activation under steady state condition (Extended Data Fig. 3c). However, despite unaltered basal proliferation upon LXR activation, the tissue appeared primed to an enhanced regenerative potential, which phenotypically surfaced only in response to injury. Supporting this hypothesis, SI crypts from mice fed with GW3965 diet when plated *in vitro* in absence of further stimuli and tested for their regenerative potential (thus mimicking an injury) formed more crypt domain/organoids (new Extended Data Fig 3d). Altogether, these new data appreciably improved our understanding of LXR activation effect on intestinal physiology.

2) The authors claim that Abca1 and Abcg1 expression is increased after irradiation and DSS in response to LXR. However, they did not use intestinal specific LXR knock-out

mice to show that *Abca1* and *Abcg1* expression is decreased compared to control after irradiation and DSS. *Abca1* and *Abcg1* could be altered independent of LXR as these genes are known to be regulated by other factors.

Authors response: We appreciate the reviewer highlighting that *Abca1* and *Abcg1* are not solely LXR target genes. We used them as a proxy for LXR activation as they are the top upregulated genes in response to LXR activation (as seen also in our qPCR and RNAseq experiments when using GW3965). Nonetheless, it has been shown that unlike other LXR target genes (such as those related to fatty acid metabolism), genes related to cholesterol metabolism and efflux (e.g., *Abca1*, *Abcg1*) are not differentially expressed between WT and LXR deficient animals (despite being LXR target genes)⁴. The explanation behind such observation is that, in the absence of a strong ligand (e.g., synthetic activation with GW3965), expression of these genes is compensated by other factors in LXR-deficient settings. In line with these findings, when we measured expression of *Abca1* and *Abcg1* in crypts isolated from Villin-cre:LXR^{fl/fl} mice (LXR^{ΔIEC}) and littermate control mice after irradiation induced damage (i.e., 2-3 dpi), we did not observe downregulation of *Abca1* and *Abcg1* in LXR-deficient cells. Nevertheless, proving that *Abca1* and *Abcg1* are LXR-target genes, GW3965 administration led to an increase in both *Abca1* and *Abcg1* transcript levels in LXR-sufficient, but not LXR-deficient IECs (see Fig. 1 for Reviewer 2).

While the use of LXR-deficient mice did not allow us to establish *Abca1* and *Abcg1* as LXR target genes, our data did demonstrate that crypt cell proliferation, buds per organoids and *Areg* expression were LXR-dependent in the context of intestinal regeneration. We hope the reviewer agrees that our use of *Abca1* as a proxy for LXR activation in wild type mice led us to investigate LXR in intestinal regeneration, and our findings do not compromise our conclusion regarding the role of LXR in this process.

Fig.1 for Reviewer-2

Figure 1: Expression analysis of *Abca1* and *Abcg1* in the SI crypts isolated from control and VillinCre:LXR^{fl/fl} mice following irradiation. *Abca1* and *Abcg1* expression by qPCR from SI crypts from control or VillinCre:LXR^{fl/fl} (LXR^{ΔIEC}) mice exposed to 10Gy total body irradiation (TBI). Mice were fed with either standard or GW3965 diet for 10 days before exposed to TBI. Data are representative of 2-3 independent experiments with 4-9 mice/group (each dot represents one mouse). Data are shown as mean ± SD. Significance was assessed by one-way ANOVA with Tukey's post hoc test. **p<0.01, ****p<0.0001, ns= not significant.

3) It appears that LXR agonism is protective after irradiation and DSS; however, some of the results are difficult to understand. GW3965 fed mice have much less immune cell infiltration in the lamina propria after DSS. It is known that LXR signaling is abundant in macrophages, which could explain this finding. The paper does not determine whether the normal crypt-villus architecture is due to LXR's effect on suppressing inflammation. On the other hand, pro-inflammatory cytokine expression is not different in DSS-induced colitis on GW3965 diet, which indicates that LXR treated mice have just as much ongoing inflammation than the standard diet. This data on the cytokines does not match the absence of inflammation on histology in the LXR treated mice.

Authors' response: We agree with the Reviewer that LXR has a well-known anti-inflammatory role. To provide a more detailed analysis of the histological score, we

present the stratified score, which includes ulceration, hyperplasia, and severity scores (Extended Data Fig. 1c-d). Although no individual parameter showed significant differences between STD and GW3965-fed mice, the area of tissue affected was higher in STD-fed mice, leading to a worse cumulative score. Although this data suggests decreased inflammation in GW3965-fed mice, it is equally possible that the lower grade of inflammation observed at the end of the experiment is the result of accelerated repair. Pro-inflammatory cytokine quantification by qPCR showed no differences between standard and GW3965-fed mice (presented in the original version of the manuscript and now as Fig. 2 for Reviewer 2). To functionally investigate this further, we treated wild type mice with DSS and administered STD or GW3965 diet only during the repair phase (i.e., between d6 and d14)⁵. LXR activation during the repair phase was sufficient to promote accelerated tissue healing (see new Fig 1h), suggesting a pro-regenerative role for LXR in epithelial cells independent from its potential role in controlling inflammation. Our new data together with the *in vivo* experiment of irradiation in Villin-Cre:LXRab1/fl mice and *in vitro* organoid studies support the thesis that LXR activation drives regeneration.

Fig.2 for Reviewer-2

Figure 2: Transcriptional changes upon LXR activation *in vivo* in mice undergoing DSS model of damage-repair. qPCR analysis of pro- and anti-inflammatory cytokines in colonic biopsies at day14 post DSS. Data are representative of 2 independent experiments with 4-5 mice/group (each dot represents one mouse). Data are shown as mean \pm SEM. Significance was assessed by unpaired *t* test between diet groups. ns= not significant.

4) Many of these findings appear to be unrelated. The authors do not do a good job of tying the findings together and providing a clear mechanism to understand them. For example, the authors state that inhibition of EGFR abolishes the pro-regenerative effect of LXR activation in organoids, but does LXR inhibition change EGFR levels? In extended data Fig 3h, there is no change in EGFR with LXR treatment. The authors also do not show that Cyp27A1 deficiency leads to a decrease in PPAR α signaling in the setting of increased tumor size.

Authors' response: As the reviewer pointed out; EGFR transcript levels do not appear to change upon LXR stimulation. We have now confirmed by Western Blot that EGFR protein levels are not altered by LXR activation (Fig. 1a for Reviewer 1). We further tested whether LXR inhibition affected EGFR expression levels in Villin-Cre:LXRab^{fllox/fllox} organoids stimulated or not with GW3965. We found that LXR deficiency did not alter EGFR expression in comparison to Villin-Cre:LXRab^{fllox/fllox} organoids, regardless of GW3965 stimulation (Fig. 3 for Reviewer 2). These results suggest that LXR does not directly control EGFR levels but rather regulates its activation by increasing *Areg* levels, as supported by our previous findings. Further confirming these notion, we now showed

that LXR activation promotes ERK phosphorylation, a downstream mediator of EGFR receptor activation, in organoid cultures. Finally, to reconcile all these findings to pro-regenerative phenotype, we now included new data showing that LXR activation does not confer a growth advantage to *Areg*^{-/-} organoids compared to *Areg*-sufficient controls. As per the link between *Cyp27a1*, LXR and PPAR α , please refer to our response below to point 7 and 8 of the Reviewer's comments.

5) Cholesterol metabolism has been shown to be important for intestinal proliferation but the authors do not appear to have taken this into consideration. It is important to address whether known mechanisms linking lipid metabolism to proliferation could be relevant to the authors findings.

6) The authors focused solely on *Areg* to explain the improved intestinal regeneration promoted by LXR activation. The data demonstrate that GW3965 has an effect on EGFR signaling. However, is hard to believe that all the metabolic remodeling happening in response to LXR activation plays no role, especially given the extensive literature on links between metabolism and proliferation. The fact that the authors identify cholesterol metabolism genes as differentially regulated during regeneration (Fig 1A) further suggests cholesterol metabolism and homeostasis play an important role. In the authors' investigation of the role of LXR activation in tumorigenesis, their RNAseq data suggested modulation of cholesterol metabolism and PPAR signaling, yet the authors decided to only investigate PPAR targets.

Authors' response: we thank the Reviewer for raising these concerns as this allowed us to draw our conclusions more accurately. Over the points 5 and 6, the Reviewer expresses two main concerns, which we answer below:

Our initial analysis (shown in Fig. 1a) identifying conserved pathway that might play a role during intestinal regeneration showed "Cholesterol metabolism" as the fourth most enriched hit. As the reviewer pointed out, regulation of cholesterol levels has been shown to control IEC (and immune cells) proliferation and may explain our phenotype. While we fully agree that this option stands, our search for pathways that promote regeneration but tumorigenesis, prompted us to forswear the study of cholesterol metabolism as high cholesterol diet^{6,7} and accumulating cholesterol by knocking down *Lpcat3*¹ has been associated with promoting colorectal cancer.

Nonetheless, the reviewer's comment that "*LXR activation may favor regeneration not only by inducing *Areg* but also by regulating cholesterol metabolic remodeling*" remains very relevant. Indeed, all our unbiased transcriptomic analysis performed comparing GW3965-stimulated samples over controls always showed genes regulating cholesterol biosynthesis or metabolism as the most upregulated hit. While we decided to focus on the transcriptional regulation of regenerative mediators (such as

Areg) as a less explored avenue, this does not exclude the possibility that LXR promotes regeneration also via regulation of cholesterol metabolism. To explore this hypothesis, we treated LXR-sufficient and -deficient organoids with cholesterol and assessed their regenerative potential. Notably, cholesterol treatment conferred a growth advantage only if the epithelial cells were LXR-sufficient (see Extended Data Fig. 8a).

Overall, these data suggest that LXR activation (therapeutically or in response to damage) promotes regeneration via its role as a master regulator of cellular cholesterol remodeling and through transcriptional control of pro-regenerative mediators. As these new data expanded our view on the mechanisms underlying LXR regenerative potential, we now included these results in the manuscript and discuss them at line 231-243.

7) It is known that *Cyp27a1*^{-/-} mice have decreased formation of bile acids and excretion of fecal bile acids, so it is possible that this impacts cholesterol absorption in intestine. Have the authors considered this might play a role in the observed phenotype?

Authors' response: we thank the reviewer for raising this suggestion. In view of the abovementioned pleiotropic effects that *Cyp27a1* deletion has on bile acid and cholesterol metabolism, to strengthen our conclusions on the *Cyp27a1*-LXR axis we now included two new experiments in Fig. 3i and Fig. 4b. In irradiation-induced injury and tumorigenesis we have now shown that LXR activation (by means of GW3965 diet) can rescue the defects (i.e., impaired regeneration and enhanced tumorigenesis) observed in *Cyp27a1*^{-/-} animals to the WT level.

8) The authors should test what happens if *Cyp27a1*^{-/-} mice are treated with PPAR α agonist. Is this protective? This will support the existence of a *Cyp27a1*-LXR-PPAR link, which is otherwise not demonstrated by current results.

Authors' response: We appreciate the reviewer's comments regarding our proposed model for LXR-mediated anti-tumor effects. In an attempt to validate our model we found that PPAR α -deficient mice were equally susceptible to AOM-DSS as their WT littermates, contradicting previous studies to which we initially referred. In addition, LXR activation equally attenuated tumor formation in both PPAR α -deficient and control mice, suggesting that PPAR α does not act downstream of LXR to impair oncogenesis (see Fig. 3 for Reviewer 1). In light of unexpected findings when attempting to functionally link PPAR α and LXR during tumorigenesis, we have developed a new model.

First, we re-interrogated our kinetic transcriptomic dataset obtained over the course of tumor formation in STD and GW3965-fed mice and found that module 5 ("Hematopoietic cell lineage" and "Intestinal immune network for production of IgA") started to diverge and was upregulated in GW3965-fed mice after d22 onwards (Fig. 4c), mirroring the difference observed in overall tumor growth between the two groups (Extended Data Fig. 10j-k). Second, we conducted Visium spatial transcriptomic and immunohistochemistry experiments, which also confirmed increased B cell and plasma cell signature in GW3965-fed mice predominantly at d43 and d70 (compared to d22, please see Fig. 4d-e). B cells accumulating in the tissue of GW3965-fed mice undergoing tumorigenesis clustered in structures resembling tertiary lymphoid structures (TLS), previously correlated with better prognosis in colorectal cancer patients³. To functionally link LXR and B cell expansion in the control of tumor development, we treated STD and GW3965-fed mice undergoing AOM-DSS protocol with anti-CD19 antibodies. This treatment partially but significantly increased tumor burden in GW3965-fed mice, suggesting that B cell expansion/activation downstream of LXR activation partially dampens tumor progression (Fig. 4f). Overall, we believe that these new findings significantly improve the quality of our manuscript.

Referee #3 (Remarks to the Author):

In the manuscript Das et al. the authors describe how LXR signaling regulates intestinal regeneration and tumorigenesis. By using different injury models (i.e., irradiation and dextran sodium sulfate (DSS)) on intestinal organoids. This setup in combination with scRNA seq generated different transcriptomic landscapes. Based on their analysis, several genes (i.e., Abca1, Apoc3, Apoa1, and Apoa4) are regulated by LXR. To assess whether LXR increases regeneration, Das and colleagues supplement an LXR agonist in the diets of mice. Both in vivo and in vitro studies demonstrate increased regeneration, which is driven by the intestinal niche cells. The regenerative response is mainly driven in an EGFR-pathway dependent manner, amphiregulin (Areg) seems responsible for this effect in intestinal stem cells.

In addition, the authors found that the LXR ligand-producing enzyme CYP27A1 was upregulated in the niche of the damaged intestinal crypts. Here, perturbations studies demonstrated either an increase or decrease of tumorigenesis upon LXR activation or inhibition, respectively.

All together the authors hypothesize that LXR activation could function as a rheostat. Where LXR regulates regeneration and preserves homeostasis and prevents neoplastic transformation. Further evidence comes from LXR activation in two different mouse tumor models where the activation results in restrained cancer progression.

In general, the manuscript is written well, but some parts miss depth and clearer explanation, moreover it is not really tailored for a broad audience and the structure is not always linear. Moreover, they have not been using state-of-the-art techniques (see Major points) to present better how significant their study and hypothesis is. The use of the different organs (small intestine and colon) and the different injury methods is unclear (DSS and irradiation). We recommend paying some extra attention in the text to why you use the different strategies in addition to our major comments. Finally the conclusion on the role of LXR as rheostat are not really justified by the data.

Authors response: We thank the Reviewer for raising some very important points in their assessment. We intend to follow their suggestions and to partially reformat our manuscript to provide a clearer message and avoid confusion.

Major points:

- Could you untangle Figure 1A even more:

DSS irradiation
small intestine

colon

Figure 1A could serve as supplemental. Will the authors find back the same results by using the representation? Here, it stresses the difference between the two organs small intestine and colon. Especially, in the mouse intestinal track there is regional specificity, which is rather important (Beumer et al., 2020 Cell and Gebert et al., 2019 Cell Reports).

Authors' response: We apologize for any confusion caused by our previous description of the methodology used to build Fig 1A. To clarify, we first performed differential gene expression analysis between 0dpi and 3dpi in small intestinal crypts and between d0 and d14 DSS treatment in bulk colonic tissue RNA sequencing. This led to the identification of 668 regeneration-specific genes in the small intestine (based on the irradiation injury model) and 4019 regeneration-specific genes in the colon (based on the DSS colitis injury model). These two sets of genes were then overlapped to identify the 245 genes displayed in the Venn diagram of Fig. 1a. We acknowledge the regional

heterogeneity of the murine gastrointestinal tract, which we have previously described in a separate publication (Parigi SM, et al, *Nat Communications*, 2022). However, the purpose of Fig 1A was to highlight a gene signature that is conserved across different organs (i.e., small intestine and colon) and different injury models (i.e., irradiation and DSS colitis). While we recognize that there may be organ- and injury-specific responses to damage, we present this as an example of a conserved response to intestinal injury. We have revised the text (at line 84-91) to provide greater clarity and to avoid any potential confusion for the reader.

In addition, irradiation and DSS treatment are different injury methods and result in a different response. In general, it would be important to study the same parts of the mouse, even though DSS is mainly affecting the mouse colon. Nevertheless, the authors could make use of the ileal part of the DSS treated mouse and compare it to the ileal part of the whole-body irradiated mouse.

This will make their findings more concise and maybe they will also find a more profound effect for a specific region of the intestine. Also did the author have similar results in colon organoids? At this moment the colon data in the manuscript is relatively little. Probably more *in vitro* experiments and analysis on the colon would be necessary.

Authors' response We concur with the Reviewer's concerns regarding the confusion that might stem from the use of different injury methods (i.e. irradiation and DSS) and the analysis of different organs (i.e. small intestine and colon). We thank the Reviewer for suggesting two very informative experiments to partially solve this issue:

1_ "The analysis of ileal sample after DSS exposure". We have now included in the revised manuscript new results showing enhanced regenerative potential of organoids generated from ileal crypts of mice exposed to DSS + GW3965 diet, without any further stimulation *in vitro* (Fig. 1j). These new data not only help clarifying the effect of LXR on two distinct types of injury by analyzing the same tissue, but also provide functional evidence of the pro-regenerative role of LXR.

2_ "Corroborating our results on the colonic tissue by adding *in vitro* data". We have now included new results in new Extended Data Fig 8d showing that the LXR-mediated induction of pro-regenerative mediators extends to colonic organoids.

- Ayyaz et al., 2019 demonstrated the role of the reverse stem (RS) cell after damage and the signaling cascade initiated by YAP to repopulate the crypt niche. By perturbing or activating YAP, AREG and EREG should also go down or up (Sera et al., 2019 Nature). First it is important for the author to describe properly the literature on intestinal regeneration and the role of YAP. Second some staining and perturbation of YAP would be quite important. Third checking the RS stem cells and comparing the scRNA clusters also with other data would be important. Finally, the authors should elaborate on how YAP, CYP27A1, and AREG behave on a transcriptomic timeline during organoid development from stem a non-stem cells. And maybe compare by using different datasets combined with a time course for GW3965 (pre)treated organoids.

Authors' response: We thank the Reviewer for raising several important points on the biology of reserve stem (RS) cells and YAP in IEC. As they suggested, we have adjusted our manuscript to accommodate a more detailed description on the role of YAP pathway in intestinal regeneration (line 225). To functionally assess whether YAP plays a role in our system, we cultured secondary organoids of SI crypts from Villin-CreERT2+ Yap^{ff} Taz^{ff} stimulated with GW3965 or DMSO. YAP ablation obtained by tamoxifen treatment over 4 days of culture led to dramatic organoids atrophy and loss of the

regenerative architecture regardless of LXR activation treatment (see Fig. 1 for Reviewer 3). These results suggest that YAP is necessary to allow organoids growth and as such, abrogation of this pathways overcomes any pro-regenerative effect observed when activating LXR. However, to more physiologically assess whether YAP is mediating LXR pro-regenerative program, we treated secondary organoids from CreERT2+ Yap^{fl/fl} Taz^{fl/fl} with GW3965 or DMSO for 8 days followed by short term YAP deletion by tamoxifen treatment for the last 24h. This treatment regimen led to efficient YAP deletion and significant downregulation of YAP target genes such as *Ilf3* and *Msln*, as seen by qPCR (Extended Data Fig. 7). However, it was not sufficient to abrogate GW3965-induced increase in *Abca1* and *Areg* transcription, suggesting that LXR drives its regenerative program independently from YAP activation (Extended Data Fig. 7). Finally, regarding the transcriptional expression pattern of *Cyp27a1*, *Yap*, *Areg* and LXR target genes over the course of organoids development, we are now providing these results in Fig. 2 for Reviewer 3 (reanalysis of the data from Serra et al., 2019, Nature). As it can be appreciated by this analysis, while *Yap* is relatively highly expressed homogenously throughout the course of organoids development, *Cyp27a1* (LXR ligand producing enzyme), LXR target gene (*Abca1*) and *Areg* follow the same pattern of upregulation during the first days of culture followed by a slight downregulation (or a plateau in some instances).

Fig.2 for Reviewer-3

Organoid kinetics reanalysis from (Serra et al., Nature, 2019)

Figure 2: Kinetic expression of *Cyp27a1*, *Abca1*, *Areg* and *Yap1* during different days of SI organoids culture. Reanalysis of RNAseq datasets from previously published study (Serra et al., 2019, Nature) to determine the kinetic expression pattern of *Cyp27a1*, *Abca1*, *Areg* and *Yap1*.

o Thinking about this transcriptomic timeline could the authors explain/comment on why in Figure 1C and 3C the levels of *Abca1* and *Cyp27a1* are not going back to a homeostatic level after stopping initial damage. We see a downwards trend in Figure 3C but it takes rather long and for Figure 1C there is constant high level.

Authors' response: We thank the reviewer for raising this very interesting observation. One could argue that, despite a macroscopic return to homeostasis (e.g., weight gain after DSS), the tissue is not completely healed (at least at a molecular and histological level) at the time points that we chose to end our experiments (i.e., d14 for DSS experiments). This might explain why *Abca1* and *Cyp27a1* do not return to homeostatic level. To prove this hypothesis, we have exposed mice to DSS-induced injury and collected the ileal crypts for qPCR at a later time point (i.e., d28), when the tissue is expected to be fully healed. Both *Abca1* and *Cyp27a1* expression levels returned to baseline at this time point, suggesting that this adapted response to damage is transcriptionally transient and subsides once the tissue has histologically healed. We have now included these new data in Fig. 1d and Fig. 3d. In the irradiation setting, however, it was unfeasible to perform such experiment as, without bone marrow reconstitution, the mice would die shortly after d5-6 and thus it was not possible to assess a later time point.

o In line with the previous comment, we find it difficult to understand that both LXR-target genes of interest, *Abca1* and *Abcg1*, do not show any upregulation during induced damage. This happens without specifically activating LXR by feeding GW39656 in neither the small intestine nor the colon as shown in Extended Fig1? This contradicts heavily with Figure 1B-C.

Authors' response: While we agree with the Reviewer that *Abcg1* levels do not display the same behavior as *Abca1* (i.e., upregulation upon injury), it is a misunderstanding that the data in the old Extended Data Fig. 1 (and now shown in Fig. 3 for Reviewer 3) contradicted the ones shown in Fig. 1c. Indeed, for *Abca1*, the data presented in the two figures are exactly the same. While in Fig. 1c we are showing *Abca1* levels upon irradiation in standard diet fed mice, in Fig. 3 for Reviewer 3, we are showing *Abca1* levels upon irradiation in both standard and GW3965 fed mice. However, the plotted values of standard diet mice are the same in both figures. Of note, the old Extended Data Fig1 has been now removed from the manuscript as it was replaced by a more

detailed analysis by scRNAseq in the new Extended Data Fig. 2. Thank you for your feedback, which helped us to improve the clarity of our presentation.

Fig.3 for Reviewer-3

Figure 3: Expression analysis of LXR target genes *Abca1* and *Abcg1* in SI crypts following irradiation induced damage-repair. qPCR analysis of *Abca1* and *Abcg1* in SI crypts at indicated days post irradiation (dpi). Data are representative of 3 independent experiments with 6 mice/condition (each dot represents one mouse). Data are shown as boxes with mean \pm min to max. Significance was assessed by unpaired *t* test between diet groups. *** $p < 0.001$, **** $p < 0.0001$.

• For the Figures 2b-d, 3e, 4i and Extended Data Figure 2b, 7c-j it would be nice to see spatial transcriptomics. Even in vivo spatial transcriptomics would make things clearer in their natural microenvironment for the different timepoints.

Authors response: As the Reviewer pointed out, spatial transcriptomics (ST) does indeed provide a more informative topographical characterization of our systems. In the revised version of the manuscript, we have included two new experiments of Visium 10x ST at 0 and 3dpi and at d0, d22 and d43 of AOM-DSS in both STD and GW3965-fed mice. These new datasets allowed us to:

- Validate and map on the tissue the overexpression of *Abca1* and *Areg* upon LXR activation and in response to damage (as seen in Fig. 1b and Extended Data Fig 6a-d).
- Locate the strongest *Areg* upregulation at the base of the crypts when mice were irradiated and fed GW3965 diet (Extended Data Fig 6a-d).
- Identify the expansion of tertiary lymphoid structures enriched in B cell signatures in the colonic tissue of GW3965-fed mice during tumorigenesis (Fig. 4d).

On a minor note, if we understood the comment above correctly, the Reviewer suggests performing ST on *in vitro* cultured organoids (which were presented in the original Figure 2b-d). While we agree that this analysis might provide new insights, we are afraid that this technology is not optimal to visualize small clusters of cells (like organoids) due to the limitations of the state-of-the-art unbiased ST not allowing single cell resolution.

• Organoid quantifications are not powerful. Think in terms of seeding density and Matrigel-dome (i.e., volume). The average per mouse is not specific enough. Therefore, one could think of whole well overviews and organoid count per well. Crypt domain/organoid is still a good measure but should be done at the last step of the previous quantifications. Maybe also think about a dead-cell marker.

Authors' response: We understand the reviewer's concern. However, while variability is undoubtedly a confounding factor in organoid quantification, the same number of

crypts were seeded in all our experiments with GW3965 or anti-AREG experiments. Moreover, all quantification were done blindly to maximize the reliability of our results. Nonetheless, we thank the Reviewer for suggesting a more thorough analysis of our organoids culture experiments. In the revised manuscript we have now added a stratified quantification showing the percentage of organoids with 0, 1, 2, 3 or more or equal to 4 crypt domains in each condition in addition to our original quantification of average crypt domain/organoid. This analysis revealed, even more robustly, that GW3965 stimulation favors the relative accumulation of highly regenerative organoids (i.e., with equal to or more than 4 crypt domain) at the expenses of poorly regenerative ones (i.e. with 0 or 1 crypt domain), (see Figs. 1 and 2 and all Extended Data Figs with organoids data). Moreover, we are now providing the quantification of organoid count/well of DMSO or GW3965 stimulated organoids in NR media (please see Fig.4 for Reviewer 3), where we did not note significant differences between the conditions.

Fig.4 for Reviewer-3

Figure 4: LXR activation does not influence SI organoid numbers grown under NR or ENR medium. Quantification of organoid numbers established from WT SI crypts and grown under NR or ENR medium with or without GW3965 treatment. Data are representative of 4 independent experiments with 4 mice/condition (each dot represents one organoid well). Data are shown as boxes with mean \pm SD. Significance was assessed one-way ANOVA with Tukey's post-test. ns= not significant.

- In the lines 62-63 is the hypothesis about different tissues. This is highlighted for the intestinal track and mainly small intestine. However, it would make this study broader by demonstrating that LXR signaling regulates also tissue regeneration in for example liver, lung, or skin.

Authors' response: We thank the reviewer for raising this fascinating suggestion, which indeed could broaden the relevance of our findings. To address this point, we performed organoids generated from salivary gland cells⁸. Briefly, salivary glands were extracted from wild type mice, dissociated into single cell suspension and plated for organoid cultures following the protocol described in⁸. After 7 days of culture in regular base media, organoids were split and plated for secondary culture with simultaneous DMSO or GW3965 stimulation. 3 days post-secondary culture, DAPT was added to the culture medium and organoids were cultured for another 1-3 days and were analyzed for organoids size and gene expression. Remarkably, GW3965 stimulation of salivary glands organoids resulted in enhanced growth and expression of LXR target genes as well as *Areg* compared to DMSO treated organoids (Extended Data Fig. 8e). These results suggest that LXR activation drives a more generalized regenerative program which is conserved in organs outside the gastrointestinal tract.

- The scRNAseq clusters are really not clear and not interpretable in line of already published datasets. For example, cluster 0 has enterocytes and regenerative cells? And the author then mention "potential involvement in intestinal regeneration and enterocyte differentiation". This part is not clear and having the scRNA sequencing in the middle also disrupt the flow as most of the other data is qPCR.

- In figure 1 the authors only look at few genes (Areg, Ereg, EGFR). Having the scRNAseq why the authors don't start with a more comprehensive analysis?

Authors' response: We apologize for the confusion and thank the reviewer for their suggestions over the two abovementioned points. We have now rephrased the text to avoid misinterpretation of our scRNAseq results. When annotating the unbiased clusters that we identified in our organoid scRNAseq, we tried to stay as true as possible to the gene signatures we observed. However, we have now removed sentences that could lead to inaccurate conclusions. We have also reorganized the order of the results to avoid flow disruption. Conceptually, the qPCR data are described before the scRNAseq as the former are done on organoids grown in ENR media and the latter on organoids grown in NR media (which we started using after noticing enhanced EGFR signaling in GW3965 treated organoids). Finally, as per the suggestion of using the scRNAseq for a more comprehensive analysis, we have now run unbiased differential gene expression analysis and identified *Areg* as the only canonical niche factor significantly upregulated in GW3965 (see Extended Data Fig 5e).

- Did the authors try to add Areg or Ereg to the organoids or the diet?

Authors' response: Yes, in Extended Data Fig. 6h-i, we have made use of the rAREG and a-AREG and observed a pro-regenerative effect of rAREG.

Minor points

- The bridge towards the human data is promising. It would be even more powerful if one could show, in addition to the expression data in Figure 3m and n, an example of a UCs staining with these markers (e.g., protein and spatial transcriptomics).

Authors' response: we agree with the reviewer that this additional analysis on human material might significantly improve the translational relevance of our manuscript. Unfortunately, due to lack of access to such material, we were unable to perform this experiment.

- Is there an explanation for the cytoskeletal remodeling in Extended data Fig 14.d?

Authors' response: we apologize for the potential misunderstanding, but we believe that cytoskeletal remodeling is not mentioned in the original Extended Data Figure 14d. We, therefore, kindly ask the Reviewer if they could clarify this point.

- Figure 4j where were the non-tumor (NT) biopsies taken?

Authors' response: in the original Figure 4j (now Extended Data Fig. 10n), we have repurposed a deposited dataset from Marisa L. et al., PLoS Med 2013. In that study, normal tissue biopsies were collected from non-tumor tissue adjacent to tumors (which were collected from both proximal and distal colon)

- When comparing the overlap between the two previously published scRNAseq datasets, general fat digestion and absorption comes up as the top hit of DEG instead of the LXR-regulated cholesterol metabolism. Why did the authors choose to not follow these hits and instead focus on the LXR pathway?

Authors' response: we decided to not focus on the first hit, given the extended body of literature that is already available on fat metabolism and regeneration. For instance, several studies have already link high fat diet consumption with enhanced IEC proliferative potential. After noticing LXR signature in the cholesterol metabolism hit, we were intrigued about the possibility of one nuclear receptor converging this pathway into a transcriptional program regulating regeneration.

- In figure 3c, Cyp27a1 seems to get upregulated only after the DSS treatment stops (day 8). This is contrary to the previous result, where LXR target gene Abca1 already starts to be upregulated after 5 days of DSS treatment (Fig 1c) even though it should be downstream of Cyp27a1. Can the authors comment on this delay?

Authors' response: as it can be appreciated from the graph in Fig. 1d, *Abca1* expression is significantly upregulated starting from day 10, while it is not at day 5. Therefore, *Abca1* expression kinetic follows *Cyp27a1* expression kinetic. In addition, the data originally shown in Figure 1b are generated from ileal crypts, while the graph in Figure 1c is based on an RNAseq experiment from colonic biopsies. The difference in the starting material used to extract RNA (ileum vs colon and crypts vs whole tissue biopsy) might account for slight differences in the kinetics too.

- There seems to be a problem with Figures hiding text. Line 79, 111, 195 and 221 seem to be hidden by the text.

Authors' response: we have reformatted the manuscript and fixed those issues.

- Ext. Data Fig 10 and 14 have a low resolution.

Authors' response: We have replaced those images with new ones with higher resolution.

References:

1. Wang, B. *et al.* Phospholipid Remodeling and Cholesterol Availability Regulate Intestinal Stemness and Tumorigenesis. *Cell Stem Cell* **22**, 206-220.e4 (2018).
2. Luo, Y. *et al.* Intestinal PPAR α Protects Against Colon Carcinogenesis via Regulation of Methyltransferases DNMT1 and PRMT6. *Gastroenterology* **157**, 744-759.e4 (2019).
3. Sautès-Fridman, C., Petitprez, F., Calderaro, J. & Fridman, W. H. Tertiary lymphoid structures in the era of cancer immunotherapy. *Nat Rev Cancer* **19**, 307–325 (2019).
4. Bideyan, L. *et al.* Integrative analysis reveals multiple modes of LXR transcriptional regulation in liver. *Proc Natl Acad Sci U S A* **119**, (2022).
5. Czarnewski, P. *et al.* Conserved transcriptomic profile between mouse and human colitis allows unsupervised patient stratification. *Nat Commun* **10**, 2892 (2019).
6. Wang, C. *et al.* Cholesterol Enhances Colorectal Cancer Progression via ROS Elevation and MAPK Signaling Pathway Activation. *Cell Physiol Biochem* **42**, 729–742 (2017).
7. Wu, C., Wang, M. & Shi, H. Cholesterol Promotes Colorectal Cancer Growth by Activating the PI3K/AKT Pathway. *J Oncol* **2022**, 1515416 (2022).
8. Yoon, Y.-J. *et al.* Salivary gland organoid culture maintains distinct glandular properties of murine and human major salivary glands. *Nat Commun* **13**, 3291 (2022).

Reviewer Reports on the First Revision:

Referee #1 (Remarks to the Author):

The authors have carefully addressed the major points raised from the first submission. This has substantially improved the manuscript. The manuscript provides for the first time clear insight into how modulation of specific signaling pathways can ameliorate tissue regeneration. Importantly, the authors provide mechanistic insight into how this is orchestrated at the molecular level using appropriate cellular and genetic tools. I believe that the findings have the potential to impact how patients suffering from intestinal ulcerations are currently being treated.

I only have a few concerns, which need to be addressed:

- 1) Line 121-124, also 148-150, as well as throughout the manuscript: What do the authors mean when they state that the number of buds equal higher regenerative potential? The buds form as a consequence of symmetry breakage (re Serra et al., 2019, Nature), and the same paper in fact argue that this represents the end of the regenerative phase of organoid growth. It has never been shown that the number of buds represent higher regenerative potential. If the authors argue for an increased regenerative potential, one expects to see an enhanced plating efficiency upon seeding cells as single cells. It is essential that the entire protocol for organoid cultures is included in the materials and methods including how organoids are replated. This is currently missing. Given that the authors put a lot of emphasis on the organoid morphology, it will be appropriate to include a temporal analysis of at least a couple of the experiments starting from single cells e.g. showing the effect of GW3965 over time. Alternatively, the effect is simply reached by enhancing proliferation.
- 2) Line 178-181: If the authors do not observed increased levels of pEGFR (data for reviewer), it is unclear how they can argue that the effect is mediated via Egfr. The experiment using an EGFR inhibitor simply demonstrate that EGFR is required for organoid growth. Have the authors considered assessing other ErbB family members? Or they should tone down their claim related to Egfr involvement.
- 3) Line 183-192: If the authors argue that GW3965 can substitute for EGF in the medium and thereby maintain the regenerative potential of organoids, they have to use longer-term experiments to assess stem cell potential/regenerative potential by doing replating experiments. Simply looking at the number of buds per organoid is insufficient and does not address this question.
- 4) Extended figure 5c – this is the only element in the figure that has the clusters in the reverse order. At first glance this can be confusing, and I recommend that the authors align this with the other elements.
- 5) The Areg analysis is focused on the epithelium (e.g. extended data figure 5). Is GW3965 inducing Areg expression in other cell types in the intestine (fibroblasts immune cells etc)?

6) Line 237-238 extending into 248-252: Here the authors state that similar effects were observed in organoids derived from the colon, yet the only test was the expression of Areg. The statement in the text is not reflecting the experiment performed by the authors. Also, under the utilised cell culture conditions, there is only minor budding from colonic organoids and the provided pictures (Extended Data Fig. 8d) clearly does not show enhanced budding. Similarly they go on to make general statements based on experiments using salivary gland organoids. If both Areg and Abca1 are target genes of Lxr it is not surprising that these are upregulated upon GW3965 stimulation. To make statements related to the general effects on tissue regeneration relevant regeneration experiments need to be completed.

7) The data related to tumorigenesis distracts from the main story and the data related to B-cell involvement in the observed phenotype is very weak. It is even included in the graphical abstracts, and one can question whether the minor effects of anti-CD19 treatment really supports the authors hypothesis.

Referee #2 (Remarks to the Author):

The manuscript is improved and I appreciate the authors' efforts.

Remaining concerns:

It was not really explained if the effect of Cyp27a1 deletion on bile acids and cholesterol absorption has an impact on the phenotype?

It is also not clear to me why during tumorigenesis there is lower LXR activation (Cyp27a1 goes down, and as a consequence, there is less Cd19 and B cells accumulation). It would have been interesting to explore why this happens, e.g. instead of comparing only AOM-DSS standard diet vs GW diet in fig 4c, also having healthy mice w and w/o treatment and see what happens there. It is not explained how the LXR response is blunted in the cancer model.

Referee #3 (Remarks to the Author):

In the revised manuscript, Das et al. have significantly expanded their work, which is impressive. The updated text is well written and has improved interpretability of the data, leading to a clearer message. The majority of comments from all reviewers are addressed adequately. The manuscript still lacks some breath, However, we therefore suggest that the addition of a few more straightforward experiments that could strengthen the support the broad applicability of the findings.

Major point 1: We have concerns regarding the quantification of regeneration based solely on the number of crypts. Especially in the SI, crypts emerge from differentiating Paneth Cells which induce a ISC niche. This process is typically not attributed to regeneration itself, but symmetry breaking after the regenerative phase (i.e., the YAP1-high phase) ended (Gregorieff et al., 2015, Serra et al., 2019). We propose that instead of using this method, the outgrowth rate from single cells should be measured in addition (i.e., directly from the treated and non-treated mice or organoids including: Fig4 for Rev3, Fig1J, Fig3I, FigExtd 6G-I, 8D, and E). While we acknowledge that the authors have mentioned starting from an equal amount of freshly isolated crypts, we believe that incorporating the outgrowth rate from single cells into the existing data would provide an even clearer picture of the regenerative process. On a similar note, in FigExtd 8D, where only AREG is shown, the data may benefit from additional information to describe regeneration (and the already existing data), such as organoid area or outgrowth efficiency.

We are not suggesting that the authors conduct another scRNA-seq experiment for the colon. However, we do recommend including additional markers, aside from AREG in FigExtd 6F to add more cell-type specificity (such as LYZ, LGR5/OLFM4, YAP1) – see also major point 2.

Major point 2: Regarding Fig1B, while we appreciate the effort to set up ST, we find the chosen representation as a swiss-roll lacking in providing a better understanding of which cells/region this effect originates in. To address this, we suggest exchanging 1B with a method similar to that used by Holloway et al. in 2020 (PMID: 33278341) with FISH to provide the reader with an initial understanding of spatial specificity. However, we would like to acknowledge the authors' excellent contribution with Ext6 A-D, which has increased our understanding of cell type specificity. However, we would also like to know more about the spatial resolution (i.e., high resolution not quantified per definition) along the crypt-villus axis.

Finally, we would like to express our appreciation to the authors for demonstrating the YAP "independency" in their study. Additionally, we would like to thank the authors for incorporating a different model system (i.e., salivary glands), to stress the broader applicability of the mechanism postulated. Moreover, we commend the authors for providing more in-depth analysis output from the scRNA-seq data.

Minor point(s):

- Fig.1E-F: Whilst figure F has quantification of Colitis score, figure 1E has only representative images, with no quantification.

Author Rebuttals to First Revision:

We truly thank the reviewers for revising our manuscript and suggesting experiments that significantly improved the quality of our story. We are happy to report that we have successfully addressed most of the major concerns raised by the reviewers. In particular, in this final round of revision, two major issues were raised:

1) Demonstrating **LXR regenerative potential in organoids culture** by performing single-cell replating assays (a concern that was shared by both reviewers 1 and 3). We have now included a significant amount of new data that address this point. Our longitudinal analysis of organoids growth upon single-cell re-seeding proves LXR pro-regenerative function in both small intestinal and colonic epithelial cells.

2) Strengthening the results on the **cellular mechanism of LXR anti-tumor effect**. This comment was suggested by reviewer 1 given the smaller effect size of rescued tumorigenesis upon B cell inhibition in LXR-agonist treated mice (included in our previous submission). We have now broadened our analysis to other arms of the adaptive immune system, as suggested by our kinetic RNAseq analysis and given the known role for T cells in dampening tumor progression. Our results show that concomitant inhibition of B and CD8+ T cells in GW3965-treated mice rescues tumor progression to levels comparable to control mice. These results suggest that, in the tumor context, LXR favors expansion of adaptive immune cells (as seen by increased formation of tertiary lymphoid structures in the tissue), which in turn dampens tumor progression.

Below, we are outlining a more detailed point-by-point response to the reviewer's comments.

Referee #1 (Remarks to the Author):

The authors have carefully addressed the major points raised from the first submission. This has substantially improved the manuscript. The manuscript provides for the first time clear insight into how modulation of specific signaling pathways can ameliorate tissue regeneration. Importantly, the authors provide mechanistic insight into how this is orchestrated at the molecular level using appropriate cellular and genetic tools. I believe that the findings have the potential to impact how patients suffering from intestinal ulcerations are currently being treated.

We are very thankful to the reviewer for these praising remarks and for finding our manuscript impactful and clinically relevant. We agree that addressing the few remaining concerns listed below will improve our message further and thus we thank the reviewer for suggesting these experiments.

I only have a few concerns, which need to be addressed:

1) Line 121-124, also 148-150, as well as throughout the manuscript: What do the authors mean when they state that the number of buds equal higher regenerative potential? The buds form as a consequence of symmetry breakage (re Serra et al., 2019, Nature), and the same paper in fact argue that this represents the end of the regenerative phase of organoid growth. It has never been shown that the number of buds represent higher regenerative potential. If the authors argue for an increased regenerative potential, one expects to see an enhanced plating efficiency upon seeding cells as single cells. It is essential that the entire protocol for organoid cultures is included in the materials and methods including how organoids are replated. This is currently missing. Given that the authors put a lot of emphasis on the organoid morphology, it will be appropriate to include a temporal analysis of at least a couple of the experiments starting from single cells e.g. showing the effect of GW3965 over time. Alternatively, the effect is simply reached by enhancing proliferation.

Author's Response: We agree with the reviewer that crypt budding *in vitro* emerges as a result of symmetry breaking events and thus has been linked to the end of the regenerative phase. Yet, other groups have used this parameter as a synonym of the regenerative capacity of organoids (see, among others, Pentinimikko N et al., Nature 2019¹; Pentinimikko N et al., Science Advances, 2022²; He G-W et al., Cell Stem Cell 2022³). Regardless of these discrepancies, the experiments suggested by the reviewer are instrumental to unequivocally prove the regenerative potential of LXR on intestinal epithelial cells. Towards this, we are now providing the following results in the revised version of the manuscript:

- **Plating efficiency upon single cell re-seeding of SI epithelial cells:** We have cultured primary organoids in ENR media (as a reference) and in NR media +/- GW3965. After 5-7 days, organoids were dissociated and 10.000 live intestinal epithelial cells were re-seeded for secondary organoid culture using the same media condition as in their primary culture (see Extended data Fig. 4g and Fig. 2d-f). After four days, plating efficiency (%) was calculated as: number of organoids/10.000*100. As seen in Fig. 2d-e and Supplementary Video 1a-c, LXR stimulation significantly increased the plating efficiency compared to DMSO treated organoids consistent with the notion that LXR promotes regenerative potential. Further, the increased buds/organoid and % of budding organoids upon LXR stimulation were retained in secondary organoid cultures (Fig. 2d, f and Supplementary Video 1a-c). These results have also been included in the revised version of the manuscript (Extended data Fig. 4g, Fig. 2d-f and Supplementary Video 1a-c) and at lines 187-193.
- **Temporal analysis of organoids growth upon LXR stimulation:** Upon single cell re-seeding (as described above), organoids were monitored longitudinally using Incucyte Imaging and the total organoids area was measured over time. GW3965 treatment resulted in increased organoids growth, which starts diverging from around day 3 and maintained throughout (see Fig. 1 for reviewer 1).

- **Clarification on organoid plating/re-seeding protocol:** A detailed description of the organoids plating and splitting condition has been added to material and methods section of the revised manuscript (Supplementary Text line 201-249) and illustrated as a cartoon in Extended data Fig. 4g.

2) Line 178-181: If the authors do not observed increased levels of pEGFR (data for reviewer), it is unclear how they can argue that the effect is mediated via Egfr. The

experiment using an EGFR inhibitor simply demonstrate that EGFR is required for organoid growth. Have the authors considered assessing other ErbB family members? Or they should tone down their claim related to Egfr involvement.

Author's Response: We agree with the reviewer that not observing higher pEGFR level might imply no involvement of EGFR signaling downstream AREG. However, the kinetics of EGFR phosphorylation is highly time-sensitive (Stanoev A et al., Cell Syst, 2018⁴; Capuani F et al., Nat Commun, 2015⁵; Bakker J et al., J. Cell. Sci, 2017⁶). Since in our experiments we are not adding directly the ligand (i.e. AREG) but rather activating a pathway upstream to AREG (i.e. LXR), it is hard to determine the most appropriate time window to measure pEGFR. Therefore, we concur with the reviewer that given the experimental data presented in the manuscript, it is appropriate to tone down the claims related to EGFR involvement in the manuscript. Please find the revised conclusions at lines 182-195 of the revised manuscript.

3) Line 183-192: If the authors argue that GW3965 can substitute for EGF in the medium and thereby maintain the regenerative potential of organoids, they have to use longer-term experiments to assess stem cell potential/regenerative potential by doing replating experiments. Simply looking at the number of buds per organoid is insufficient and does not address this question.

Author's Response: We apologize with the reviewer for the misunderstanding. We do not intend to posit that GW3965 can substitute EGF in organoids media, as we do not provide evidences for that in the manuscript. However, since both AREG and EGF are known to act via EGFR, we reasoned that adding exogenous EGF might mask the pro-regenerative effect driven by LXR induced AREG. Indeed, in our organoids experiments, GW3965 stimulation promoted regeneration more robustly in NR conditions, compared to ENR. The ENR condition in the manuscript is provided as a reference (ENR media being the standard culture media for small intestinal organoids). We have now rephrased the manuscript (line 182-187) to avoid potential misunderstandings.

4) Extended figure 5c – this is the only element in the figure that has the clusters in the reverse order. At first glance this can be confusing, and I recommend that the authors align this with the other elements.

Author's Response: We apologize for this typo; we have modified the graph to make it consistent with the other data in the manuscript.

5) The Areg analysis is focused on the epithelium (e.g. extended data figure 5). Is GW3965 inducing Areg expression in other cell types in the intestine (fibroblasts immune cells etc)?

Author's Response: To test whether LXR promotes AREG in cell types other than the epithelium, we performed the following experiments:

- **Fibroblast:** Fibroblasts from small intestine were established following a previously published protocol⁷. Once established, cells were stimulated for 72h with 1µM GW3965 and *Areg* expression was measured by qPCR. As shown in Fig. 2a-c for reviewer 1, while LXR stimulation efficiently induced *Abca1* expression, it failed to promote *Areg* expression, suggesting that LXR activation in fibroblast does not regulate *Areg* expression.

- **Immune cells:** Given the challenges of maintaining primary immune cells in culture over time, we administered standard or GW3965-diet to wild type mice and assessed AREG production by intracellular cytokine staining upon PMA-Ionomycin (PMA/IONO) stimulation. As expected, AREG was expressed by a large proportion of CD4⁺ T cells and innate lymphoid cells (ILCs). While LXR stimulation induces AREG in CD4⁺ T cells, it did not induce AREG production by ILCs (see **Fig. 2d-f for reviewer 1**). While we find this data interesting and we can not rule out the potential involvement of LXR in inducing Areg in other cell types, it does not affect our conclusions regarding its role in the intestinal epithelium. However, due to our current limited understanding of the role of LXR induced CD4⁺ T cell-derived AREG in intestinal regeneration, we chose not to include this data in the main text and will focus on it in future investigations.

6) Line 237-238 extending into 248-252: Here the authors state that similar effects very observed in organoids derived from the colon, yet the only test the expression of Areg. The statement in the text is not reflecting the experiment performed by the authors. Also, under the utilised cell culture conditions, there is only minor budding from colonic organoids and the provided pictures (Extended Data Fig. 8d) clearly does not show enhanced budding. Similar they go on to make general statements based on experiments using salivary gland organoids. If both Areg and *Abca1* are target genes of *Lxr* it is not surprising that these are upregulated upon GW3965 stimulation. To make statements related to the general effects on tissue regeneration relevant regeneration experiments need to be completed.

Author's Response: We agree with the reviewer that more functional experiments are required to broaden our conclusions on the pro-regenerative effect of LXR to other organs. Towards this, we are now including the following results:

- **Colon:** we have repeated colonic organoids cultures, this time assessing the regeneration potential by splitting and re-seeding single cells for secondary organoid culture (please see **Extended Data 8e-h and and Supplementary Video 2a-b**). We observed that LXR stimulation significantly boosted organoid plating efficiency and total organoid area growth over time, thus supporting the pro-regenerative role of LXR stimulation on colonic epithelial cells. These results are now included in the revised manuscript (**Extended Data 8e-h and and Supplementary Video 2a-b**) and at lines 250-254.
- **Salivary glands:** We agree with the reviewer that simply measuring *Areg* expression upon LXR stimulation of salivary glands organoids does not imply heightened regeneration. Nonetheless, our results (provided in the previous version of the manuscript now **Extended Data Fig. 8i**) showing increased salivary gland organoids area upon GW3965 stimulation argue for LXR pro-regenerative role. We are now complementing these results by providing evidences of enhanced salivary gland regeneration upon LXR activation *in vivo* (**Extended data Fig. 8j**). Briefly, we administered standard or GW3965-containing diet to WT mice followed by focal irradiation of the neck region targeting salivary glands. Using salivary gland duct area as a proxy for regenerative growth, an unbiased histological evaluation at 14 days post-irradiation (dpi) revealed that mice fed a GW3965 diet had a larger ductal area compared to those fed a standard diet. These data are now included in the revised manuscript (**Extended Data Fig. 8j**) and at lines 257-260.

7) The data related to tumorigenesis distracts from the main story and the data related to B-cell involvement in the observed phenotype is very weak. It is even included in the graphical abstracts, and one can question whether the minor effects of anti-CD19 treatment really supports the authors hypothesis.

Author's Response: Since the main concept of our manuscript is to demonstrate that LXR unlinks regeneration and tumorigenesis, and therefore may represent a safer alternative to treat patients with intestinal disorders in which mucosal healing need to be achieved. We believe that this part constitutes an important piece of the main story and thus should not be excluded from the manuscript or removed from the graphical abstract. We sincerely hope the reviewer agrees with our approach.

We agree with the reviewer that inhibiting B cells led to a weak phenotype. We thus have broadened our analysis to other arms of the adaptive immune system that might play a role in this context as suggested by our kinetic RNAseq analysis (i.e. module 5 showing robust hematopoietic lineage response in GW3965 treated mice) and by the expansion of tertiary lymphoid structures upon LXR activation. We are now including new data in this rebuttal (**lines 343-352**) showing that concomitant inhibition of B cells and CD8+ T cells has a larger effect size on rescuing LXR-mediated anti-tumor effect (please see **Fig. 4f and Extended Data Fig. 11a-b**). These data are in line with recent reports showing a cooperative effect of B and T cells in imparting anti-tumor responses in multiple tumors including colorectal cancer⁸⁻¹¹, and therefore enable us to conclude that the LXR-mediated anti-tumor effect requires the adaptive immune system.

Referee #2 (Remarks to the Author):

The manuscript is improved and I appreciate the authors' efforts.

We thank the reviewer for his suggestions in the first submission and for acknowledging the improvement and the quality of our manuscript.

Remaining concerns:

It was not really explained is if the effect of Cyp27a1 deletion on bile acids and cholesterol absorption has an impact on the phenotype?

Author's Response:

Answer-1

We agree with the reviewer that whole body Cyp27a1 knockout (KO) mice have defective bile acid synthesis, impacting intestinal cholesterol absorption (Choudhuri S et al., Drug Metab Dispos, 2021¹²), which could influence the observed phenotype. However, the rescue of impaired regeneration in Cyp27a1 KO mice with GW3965 suggests that bile acids are not required for CYP27A1-LXR mediated intestinal regeneration. Despite this, our results do not rule out the possibility that the impaired regeneration in Cyp27a1 KO mice might be affected by altered bile acid and cholesterol absorption. A more systematic study involving conditional deletion of Cyp27a1 in a tissue-specific manner, with and without bile acid supplementation, is necessary to fully understand the role of Cyp27a1 in bile acid and cholesterol absorption and its effect on intestinal regeneration. This will be a future focus in our lab. Given the limitations of our current findings, we have acknowledged these caveats in the revised manuscript. Please refer to lines 287-289 for further details.

It is also not clear to me why during tumorigenesis there is lower LXR activation (Cyp27a1 goes down, and as a consequence, there is less Cd19 and B cells accumulation). It would have been interesting to explore why this happens, e.g. instead of comparing only AOM-DSS standard diet vs GW diet in fig 4c, also having healthy mice w and w/o treatment and see what happens there. It is not explained how the LXR response is blunted in the cancer model.

Author's Response:

Answer-1

The reviewer raises an interesting point regarding the endogenous expression pattern of the CYP27A1-LXR axis during the course of AOM-DSS tumorigenesis by comparing the kinetic datasets only in STD diet-fed mice. This is to explore whether the downregulation of all the players involved, such as the correlative downregulation of the CYP27A1-LXR axis in human CRC compared to non-tumor control (now Extended Data Fig. 11c), is in agreement with the development of tumors.

To test this hypothesis, we mined our kinetics datasets of only STD diet-fed mice and analyzed the expression of Cyp27a1, LXRs (*Nr1h2* and *Nr1h3*), and LXR target genes

(*Abca1* and *Abcg1*) longitudinally during the course of AOM-DSS tumorigenesis. However, unlike that of human tumor datasets, we did not observe significant downregulation of the genes analyzed except for *Nr1h3* (LXR α) (see Fig. 1 for reviewer 2). A potential caveat is that mouse tumor datasets include both tumor and non-tumor parts in the biopsy, unlike human tumor biopsies primarily composed of tumor cells.

On the contrary, we observed a causal upregulation of CYP27A1 and *Abca1* (LXR activation) in the context of tissue damage. Understanding this mechanism is a crucial area for future investigation. Once we understand the mechanism by which CYP27A1 and/or *Abca1* (LXR activation) are upregulated in the context of regeneration, we will test them in the context of tumorigenesis, rather than relying on pharmacological LXR activation with GW3965. While this is a future direction actively pursued in the laboratory, we believe that it is beyond the scope of the current manuscript.

Referee #3 (Remarks to the Author):

In the revised manuscript, Das et al. have significantly expanded their work, which is impressive. The updated text is well written and has improved interpretability of the data, leading to a clearer message. The majority of comments from all reviewers are addressed adequately. The manuscript still lacks some breath, However, we therefore suggest that the addition of a few more straightforward experiments that could strengthen the support the broad applicability of the findings.

We thank the reviewer for the laudatory remarks, and we are pleased that they found our manuscript improved and both conceptually and translationally relevant. We also thank them for suggesting some few key experiments to solidify our conclusions.

Major point 1: We have concerns regarding the quantification of regeneration based solely on the number of crypts. Especially in the SI, crypts emerge from differentiating Paneth Cells which induce a ISC niche. This process is typically not attributed to regeneration itself, but symmetry breaking after the regenerative phase (i.e., the YAP1-high phase) ended (Gregorieff et al., 2015, Serra et al., 2019). We propose that instead of using this method, the outgrowth rate from single cells should be measured in addition (i.e., directly from the treated and non-treated mice or organoids including: Fig4 for Rev3, Fig1J, Fig3I, FigExtd 6G-I, 8D, and E). While we acknowledge that the authors have mentioned starting from an equal amount of freshly isolated crypts, we believe that incorporating the outgrowth rate from single cells into the existing data would provide an even clearer picture of the regenerative process.

On a similar note, in FigExtd 8D, where only AREG is shown, the data may benefit from

additional information to describe regeneration (and the already existing data), such as organoid area or outgrowth efficiency. We are not suggesting that the authors conduct another scRNA-seq experiment for the colon.

We thank the reviewer for suggesting these experiments to strengthen our conclusions. As outlined in detail in response to reviewer 1 (who suggested a similar approach, please see Point 1 and 6 in response to reviewer 1), we have now included the following results in the revised version of the manuscript:

- **Plating efficiency upon single cell re-seeding:** We have cultured secondary organoids starting from the same number of single cells obtained from primary organoids cultured in the presence (or not) of GW3965. As shown in **Extended data Fig. 4g and Fig. 2d-f and Supplementary Video 1a-c** (lines 187-193) of the revised manuscript, LXR activation resulted in significantly higher plating efficiency (**Fig. 2d-e**), buds/organoid (**Fig. 2d, f**), % of budding organoids (**Fig. 2d, f**) and increased organoids area over time (**Fig. 1a for reviewer 1**) upon single cell re-seeding, further confirming LXR pro-regenerative effect.
- **Regenerative readout for colonic organoids:** we have performed the same experiment (i.e. single cell re-seeding for secondary organoid culture) this time starting from primary colonic organoids. Similarly to what observed in the small intestine, LXR activation resulted in enhanced plating efficiency and organoids growth over time, suggesting that its pro-regenerative role is conserved across tissues (**Extended data Fig. 8d-h and Supplementary Video 2a-b**). These data are included in the revised manuscript at **lines 250-254**

However, we do recommend including additional markers, aside from AREG in FigEtxd 6F to add more cell-type specificity (such as LYZ, LGR5/OLFM4, YAP1) – see also major point 2.

Major point 2: Regarding Fig1B, while we appreciate the effort to set up ST, we find the chosen representation as a swiss-roll lacking in providing a better understanding of which cells/region this effect originates in. To address this, we suggest exchanging 1B with a method similar to that used by Holloway et al. in 2020 (PMID: 33278341) with FISH to provide the reader with an initial understanding of spatial specificity. However, we would like to acknowledge the authors' excellent contribution with Ext6 A-D, which has increased our understanding of cell type specificity. However, we would also like to know more about the spatial resolution (i.e., high resolution not quantified per definition) along the crypt-villus axis.

We thank the reviewer for acknowledging our efforts to provide spatial coordinates for LXR-mediated intestinal pro-regenerative responses. As they recommended, we have now performed RNA insitu hybridization (RNAscope) for *Areg*, *Abca1* and *Lgr5* on intestinal tissues from STD and GW3965-fed mice at 3 dpi showing increased *Abca1* and *Areg* expression in the crypts of GW3965-treated mice (**Extended data Fig. 6g**). Intriguingly, aside from detecting *Areg*⁺ cells in the crypt, we also observed *Areg*⁺ cells at the crypt-villus junction, some of which co-expressed *Lgr5* (**Extended data Fig. 6g**). These results are now included in the revised manuscript (**Extended Data Fig. 6g**) and added at lines 216-220.

Regarding Fig. 1b, we agree with the reviewer that while the provided ST dataset shows overall increase in *Abca1* upon irradiation, does not provide sufficient spatial resolution along the crypt-villus axis. We now have performed RNA insitu hybridization (RNA scope) for *Abca1* and *Lgr5* on intestinal tissues from STD diet fed mice at 0 and 3 dpi. High resolution images spanning crypt-villus axis are provided in **Fig. 1b**. Analysis of crypt-villus axis at high resolution (**Fig. 1b**), we observed that epithelial cells in the villi expressed higher *Abca1*

compared to the crypts in steady state. However, upon irradiation, we observed more number of crypt epithelial cells expressing Abca1 compared to the crypts in steady state (Fig. 1b). These results are now included in the revised manuscript (Fig. 1b).

Finally, we would like to express our appreciation to the authors for demonstrating the YAP "independency" in their study. Additionally, we would like to thank the authors for incorporating a different model system (i.e., salivary glands), to stress the broader applicability of the mechanism postulated. Moreover, we commend the authors for providing more in-depth analysis output from the scRNA-seq data.

We are truly gratified to read this congratulatory note. We want once more to express our gratitude to all the reviewers and the Editorial board as we firmly believe that this revision process significantly improved the quality of our manuscript and strengthened our conclusions.

Minor point(s):

- Fig. 1E-F: Whilst figure F has quantification of Colitis score, figure 1E has only representative images, with no quantification.

We thank the reviewer for pointing this out. In our revised manuscript, we have provided an artificial intelligence-based histological quantification at 5dpi showing enhanced crypt area and crypt:villus area ratio in GW3965-fed mice over standard diet-fed controls (Fig. 1e). These results are now included in the revised manuscript replacing the previous Fig. 1e.

1. Pentinmikko, N. *et al.* Notum produced by Paneth cells attenuates regeneration of aged intestinal epithelium. *Nature* **571**, 398–402 (2019).
2. Pentinmikko, N. *et al.* Cellular shape reinforces niche to stem cell signaling in the small intestine. *Sci. Adv.* **8**, eabm1847 (2022).
3. He, G.-W. *et al.* Optimized human intestinal organoid model reveals interleukin-22-dependency of paneth cell formation. *Cell Stem Cell* **29**, 1333-1345.e6 (2022).
4. Stanoev, A. *et al.* Interdependence between EGFR and Phosphatases Spatially Established by Vesicular Dynamics Generates a Growth Factor Sensing and Responding Network. *Cell Systems* **7**, 295-309.e11 (2018).
5. Capuani, F. *et al.* Quantitative analysis reveals how EGFR activation and downregulation are coupled in normal but not in cancer cells. *Nat Commun* **6**, 7999 (2015).
6. Bakker, J., Spits, M., Neefjes, J. & Berlin, I. The EGFR odyssey – from activation to destruction in space and time. *Journal of Cell Science* jcs.209197 (2017) doi:10.1242/jcs.209197.
7. Frede, A. *et al.* B cell expansion hinders the stroma-epithelium regenerative cross talk during mucosal healing. *Immunity* **55**, 2336-2351.e12 (2022).
8. Bod, L. *et al.* B-cell-specific checkpoint molecules that regulate anti-tumour immunity. *Nature* (2023) doi:10.1038/s41586-023-06231-0.
9. Wang, Q. *et al.* Tertiary lymphoid structures predict survival and response to neoadjuvant therapy in locally advanced rectal cancer. *npj Precis. Onc.* **8**, 61 (2024).
10. Zhang, Y. *et al.* Tertiary lymphoid structural heterogeneity determines tumour immunity and prospects for clinical application. *Mol Cancer* **23**, 75 (2024).
11. Di Caro, G. *et al.* Occurrence of Tertiary Lymphoid Tissue Is Associated with T-Cell Infiltration and Predicts Better Prognosis in Early-Stage Colorectal Cancers. *Clinical Cancer Research* **20**, 2147–2158 (2014).
12. Choudhuri, S. & Klaassen, C. D. Molecular Regulation of Bile Acid Homeostasis. *Drug Metab Dispos* **50**, 425–455 (2022).

Reviewer Reports on the Second Revision:

Referees' comments:

Referee #1 (Remarks to the Author):

The authors have addressed all of my concerns.

Referee #2 (Remarks to the Author):

The authors have addressed my concerns and I now recommend publication.

Referee #3 (Remarks to the Author):

The authors have effectively addressed our queries. Key strengths include the validity of their methodology and data, as well as the clarity and quality of presentation. They have appropriately utilized statistics and addressed uncertainties. Additionally, the references adequately credit previous work. Overall, the abstract, introduction, and conclusions are clear and contextually appropriate. It can be noted that all these aspects have been thoroughly checked and found satisfactory.